# Hydrothermal weakening and slope instability at Vulcano (Italy) analyzed using drones and in-situ strength measurements

Benjamin F. De Jarnatt [1,2] ✉, Thomas R. Walter[1,2], Michael J. Heap [3,4], Daniel Müller [1] & Antonino Fabio Pisciotta[5]

Instability at volcanic edifices poses significant hazards, yet the processes driving rock weakening, particularly on steep, eroding flanks, remain poorly understood due to limited accessibility. Hydrothermal alteration is a key factor in weakening volcanic rocks, contributing to edifice destabilization and flank instability. La Fossa cone (Vulcano, Italy) provides an ideal setting for this study, with accessible hydrothermal alteration at the crater rim and similar alteration along inaccessible flanks that have a recent history of mass wasting. Here, we developed an integrated methodology combining drone photogrammetry with in situ Schmidt hammer testing to derive an empirical alteration-to-strength relationship for the crater rim and applied this knowledge to alteration sites on inaccessible flanks. An alteration map was generated using a Principal Component Analysis (PCA) to aid our classifications. This map was used to transpose over 1000 Schmidt hammer measurements (R-values ranging from 10.5 to 82), creating a thematic strength-alteration map. Results indicate a ~50% reduction in relative rock strength correlating with areas of degassing and hydrothermal activity, which coincides with past mass-wasting events. This integrated approach offers a transferable workflow for assessing volcanic slope instability, with direct applications to hazard monitoring and early warning systems.

Volcanic flank instability and subsequent mass-wasting pose considerable hazards at both active and dormant volcanoes worldwide. Large-scale mass-wasting events, such as sector collapses that generate volcanic debris avalanches, can have far-reaching consequences and potentially devastating effects on local communities and infrastructure. However, despite their high impact, such large-scale events are relatively rare, and our ability to forecast them remains limited. Documented observations are confined to only a few well-studied cases[1,2]. Volcano flank failure may occur suddenly and catastrophically, generating far-traveling deposits and even tsunami hazards, or evolve gradually, as a slow-moving mass-wasting process, such as creep or a rock slide that can be closely monitored over an extended period[3,4]. These failure styles reflect the wide spectrum of landslide behaviors documented in geologic studies. Another key factor is the scale of flank instability, ranging from small mass-wasting events (a few m³) to giant landslides (>1000 km³),

with the scale inversely proportional to the occurrence rate[3]. The process of a volcanic flank becoming unstable can manifest from a range of processes, including flank over-steepening[4–6], magma emplacement[7], stress changes[8], weakened basement rock[9,10], shifts in pore pressure[7,11–13], heightened seismic activity[4,14], basement spreading[15], and hydrothermal alteration[16–18]. While a single factor may trigger instability, it often results from a combination of these processes acting together[4,19]. Among these processes, hydrothermal alteration is gaining increasing attention for its persistent role in weakening volcanic flanks.

Hydrothermal alteration provides insight into both the timing and location of failure, yet its effects often occur in steep, hazardous, or otherwise inaccessible areas, making it difficult to monitor[20]. As a result, the link between hydrothermal alteration and mass-wasting is usually established in the aftermath, based on hydrothermally altered material within debris

[1]GFZ Helmholtz Centre for Geosciences, Potsdam, Germany. [2]Institute of Geosciences, University of Potsdam, Potsdam, Germany. [3]Institut Terre et Environnement de Strasbourg, CNRS, UMR 7063, Université de Strasbourg, Strasbourg, France. [4]Institut Universitaire de France (IUF), Paris, France. [5]Istituto Nazionale di Geofisica e Vulcanologia (INGV), Palermo, Italy. ✉e-mail: bdejarna@gfz.de

deposits[17]. Evidence of this relationship has been observed in past mass wasting events, including those at La Soufrière de Guadeloupe (France), Las Casitas (Nicaragua), Mount Hood and Mount Rainier (USA), Santiaguito (Guatemala), Merapi (Indonesia), Volcán de Colima (Mexico), and elsewhere[21–24].

For example, Mount Rainier's edifice contains hydrothermally altered material that has contributed to past major sector collapse events, notably the Osceola Mudflow, which generated a ~3.8 km² lahar that traveled over ~113 km more than 5000 years ago[23,25]. In contrast, Merapi exhibits smaller-scale, more frequent dome collapses, where zones of hydrothermal alteration reduce dome stability and facilitate gravitational dome collapse and pyroclastic flows[21,26]. Deposits from both Mt. Rainier and Merapi showed evidence of acid-sulfate alteration, highlighting the pervasive role of hydrothermal systems in driving instability. These studies demonstrate not only the global relevance of hydrothermal weakening but also the persistent challenge it poses for volcanic hazard assessment, specifically, the difficulty of identifying altered, unstable material before failure occurs.

While hydrothermal alteration is most often associated with rock weakening, certain processes, such as silicification, can instead increase rock strength by reducing porosity and permeability. This occurs through the infilling of fractures and pore spaces with silica-rich minerals, leading to increased cohesion and stiffness[27,28]. Nevertheless, strengthening as a result of alteration has also been found to contribute to instability by facilitating pore pressure buildup and reducing permeability, ultimately promoting

deformation or failure[12]. These contrasting effects underscore the complexity of hydrothermal alteration and the importance of understanding how different alteration processes influence slope stability.

Consequently, there is a pressing need to develop methods for identifying areas susceptible to alteration-driven flank failure. With its active hydrothermal system and frequent landslides, the La Fossa cone of Vulcano Island (Italy) (Fig. 1) serves as an ideal natural laboratory for developing and testing these methods, ultimately improving forecasts for larger, more hazardous volcanic collapses. Although the landslides at La Fossa are orders of magnitude smaller than sector collapses, their frequency and accessibility offer an opportunity to develop a workflow that can later be scaled to assess volcanic flanks of greater size and destructive potential. These flank failures at La Fossa are thought to be driven, in part, by hydrothermal alteration, making it an ideal site for identifying altered regions prone to future flank instability[29].

Hydrothermal alteration changes the composition, mineralogy, and color of rocks, which is why it can often be recognized by the naked eye[30]. The effects of hydrothermal alteration in volcanic settings are widely acknowledged for their impact on the physicochemical and petrophysical properties of the associated rocks[12,26,31–33]. Hydrothermal alteration in volcanic environments typically involves the circulation of aggressive, acidic, and hot (>200 °C) hydrothermal fluids, facilitating fluid-rock interactions that lead to mineral dissolution, replacement, and precipitation. Consequently, this process alters the chemical and physical composition of the

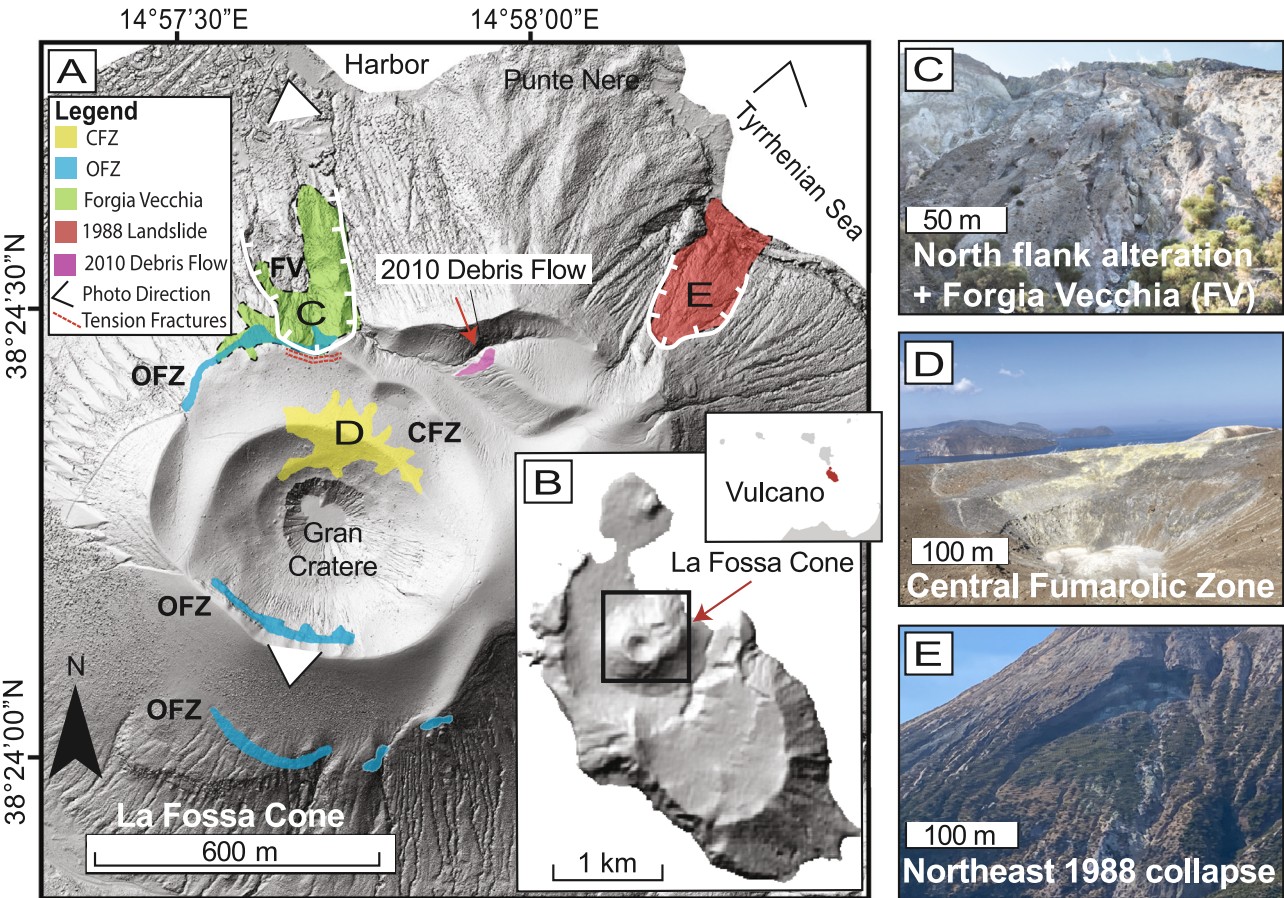

**Fig. 1 | La Fossa cone and highlighted regions of interest. A** The shaded-relief map overview shows the nested crater regions, with the deepest crater being Gran Cratere. Regions highlighted are the flank collapse area on the north flank, including Forgia Vecchia (FV) (shaded in green, labeled "C"), landslide scarp, tension cracks, the area of intense alteration, Central Fumarolic Zone (CFZ; shaded in yellow, labeled "D"), the Outer Fumarolic Zone (OFZ) (shaded in blue), the collapsed slope in the northeast (1988 landslide, shaded red, labeled "E"), and the small and most recent

landslide (2010 debris flow, shaded in pink). The triangles represent the camera's direction for the photos taken in (**C–E**). **B** Shaded-relief map of Vulcano Island, with La Fossa cone outlined by a black box. The inset shows the location of Vulcano (highlighted in red) within the Aeolian Archipelago, north of Sicily. **C** Photograph of the northern flank, taken from the base of La Fossa facing S-SE. **D** Photograph of the fumarole field, taken from the southern crater rim facing the N-NE. **E** Photograph of the 1988 landslide scar, taken offshore facing S-SW.

rock from its original state[34,35], creating "rotten rocks". Previous studies have demonstrated that hydrothermal alteration can affect the porosity, permeability, and strength of volcanic rocks, thereby compromising the gravitational stability of the volcanic structure[12,26,31,33]. Specifically, the relationship between hydrothermal alteration and mechanical weakening has been well established, with strength closely linked to increased porosity and alteration intensity. Heap et al.[17] and Darmawan et al.[26] both observed uniaxial compressive strength (UCS) weakening in hydrothermally altered volcanic rocks. Although conducted at different volcanoes, their work both demonstrated that strength loss is strongly correlated with increasing porosity, a greater proportion of secondary minerals, isotopic shifts, and rock discoloration. Consequently, these findings highlight the need to monitor hydrothermally altered edifices; however, limited and hazardous access to high-risk areas remains a major challenge for effective hazard assessment.

Typically, regions undergoing hydrothermal alteration are found near craters and along the flanks of volcanic edifices, where fluid ascent is controlled by stress and fracture networks[36]. However, it is not always visible at the surface, particularly in unbreached volcanoes, where hydrothermal alteration remains confined at depth. In such cases, detecting subsurface alteration is especially challenging due to limited surface expression and potential coverage by younger deposits. Where exposed, altered regions often lie in steep, unstable, or actively degassing areas that are hazardous or inaccessible. As a result, evaluating their mechanical properties and identifying areas prone to failure, particularly in such complex terrain, remains a challenge. Previous works, therefore, applied remote sensing technologies, including multispectral and hyperspectral imaging[37], optical and infrared drone remote sensing[30], and airborne electromagnetic and magnetic measurements[38] to map areas containing hydrothermal deposits. While these methods remain valuable for monitoring volcanoes and identifying hydrothermal alteration, they rarely provide insight into the mechanical behavior of rocks on hydrothermally altered flanks. In this study, we address this limitation by combining high-resolution drone photogrammetry and collecting extensive field strength measurements to quantify hydrothermally altered rocks and further explore the possibility of accessing otherwise inaccessible areas. Using La Fossa cone as our focus site, we extrapolated our data to regions of the volcano that are not directly accessible.

Here, we integrated thousands of close-range drone images to create a high-resolution map, enabling a pixel-wise analysis of color variations linked to hydrothermal alteration and facilitating surface classification. Additionally, over 1000 in situ relative strength measurements were collected using a Schmidt hammer, providing a second dataset for assessing rock mechanical properties (see "Methods"). Combining these datasets, we established an empirical relationship between increasing alteration and decreasing rock strength at the La Fossa cone, allowing us to understand instability hazards at the outer flanks, particularly in the northern and northeastern regions. This approach overcomes the previous limitations of traditional rock mechanical assessments, which often rely on laboratory experiments, and provides a scalable framework for understanding how hydrothermal alteration influences volcanic flank instability.

## Study area

The La Fossa volcano, located on Vulcano Island, within the Aeolian Archipelago, represents the southernmost volcanic structure of this volcanic arc situated in the Tyrrhenian Sea, off the northern coast of Sicily, Italy (Fig. 1B). The archipelago lies above a subduction zone related to the convergence of the African and European plates, characterized by the northwestward subduction of the Ionian lithospheric slab beneath the Tyrrhenian lithosphere, where subduction-related stresses and hydrothermal activity have contributed to flank instability across Aeolian Arc volcanoes[39,40]. The complex NNW–SSE fault system further influences the structural and volcanic development of the region and plays a role in the recent tectonic activity and deformation processes observed specifically at Vulcano[41,42]. La Fossa volcano is a composite volcanic edifice characterized by a pyroclastic cone approximately 400 m high with a basal diameter of about

1200 m at sea level. It is situated within La Fossa Caldera, a depression formed by past eruptive and collapse events that have shaped its current morphology[41,43–45]. The most recent eruption occurred between 1888 and 1890, becoming the archetype of "Vulcanian" eruptions, characterized by dense lapilli-tuffs and volcanic bombs[46], establishing La Fossa cone's present-day structure. Prominent morphological elements include the Forgia Vecchia on the outer northern flank and the Punte Nere promontory (Fig. 1A), indicative of a complex volcanic history shaped by eruptive and erosive processes[47]. The summit hosts prominent nested crater structures, the most recent of which is the Gran Cratere (Fig. 1A). This crater contains abundant high-temperature fumaroles[48,49], hereafter referred to as the Central Fumarolic Zone (CFZ) and Outer Fumarolic Zone (OFZ).

La Fossa has undergone frequent periods of anomalous fumarolic degassing, notably in 1979–1981, 1985–1986, 1988–1991, 1996, 2004–2005, 2009, and most recently since September 2021[50–52]. These periods are characterized by elevated fumarole gas concentrations of $CO_2$, $N_2$, and He relative to $H_2O$, suggesting increased magmatic influences[50,53,54]. Additional indicators of past unrest include fumarole temperatures reaching as high as 650 °C in 1991[54], as well as outgassing fluxes that increased by one order of magnitude from 1983 to 1995[55], along with heightened diffuse degassing and thermal output from the flanks[50]. The 2021 unrest was driven by a rapid, large-scale injection of magmatic gas and heat into the shallow hydrothermal reservoir[52], leading to boiling, overpressurization, and fracturing. This led to massive vapor-driven degassing, resulting in measurable crater uplift and increased seismicity[44].

Aside from its changing fumarole fields and alteration zones, over the past 35 years, La Fossa cone has experienced geomorphological changes due to landslides that occurred during both intense and quiescent fumarolic activity. One of the most notable recent events took place in 1988 (Fig. 1A, E) when approximately 200,000 m³ of material from La Fossa's northeastern flank slid into the sea, generating a small tsunami[29]. Several triggering mechanisms have been proposed to account for the 1988 landslide. Previous works identified hydrothermally altered surfaces at the main scarp and upper landslide sections, along with reduced fluid conductivity and increased baseline water content. Additionally, the transformation of volcanic minerals into hydrothermally altered phases was also observed, likely modifying the material's mechanical and hydraulic properties[29]. These findings suggest that hydrothermal alteration may also have played a key role in this morphological event[29]. Past works at La Fossa have also mapped zones undergoing hydrothermal alteration using geochemical[30], thermal[56], and remote sensing approaches[30,57], establishing the location's suitability for examining how hydrothermal processes contribute to volcanic flank instability.

Further signs of instability are evident along the northern flank, particularly in the Forgia Vecchia regions (Fig. 1A, C), which exhibit active deformation, as indicated by geodetic surveys[58]. This region is continuously monitored using geophysical methods (e.g., GPS and seismic stations) due to its ongoing morphological slope changes[58], sub-parallel tension cracks near the slope edge[59] (Fig. 1A), and proximity to the village of Vulcano and the nearby harbor (Fig. 1A). Additionally, in 2010, another landslide occurred along the northeastern external rim of La Fossa cone, resulting in a small debris flow not far from the other landslide areas (Fig. 1A). This event occurred during calm conditions, as indicated by stable and seismic monitoring data[60], and exposed hydrothermally altered pyroclastics at the surface[29]. The recent 1988 and 2010 landslides at La Fossa, combined with persistent degassing and widespread hydrothermal alteration, offer a unique opportunity to investigate the controls on volcanic slope stability and alteration-related weakening. Here, we combine high-resolution drone imagery with in situ strength testing (see "Methods") in accessible regions to infer relative rock strength across the otherwise inaccessible steep flanks.

## Results
### Orthomosaic analysis and hydrothermal alteration mapping
The true-color orthomosaic map generated from drone (unmanned aerial vehicle (UAV)) imagery (Fig. 2A) covers an area of approximately 3.74 km²

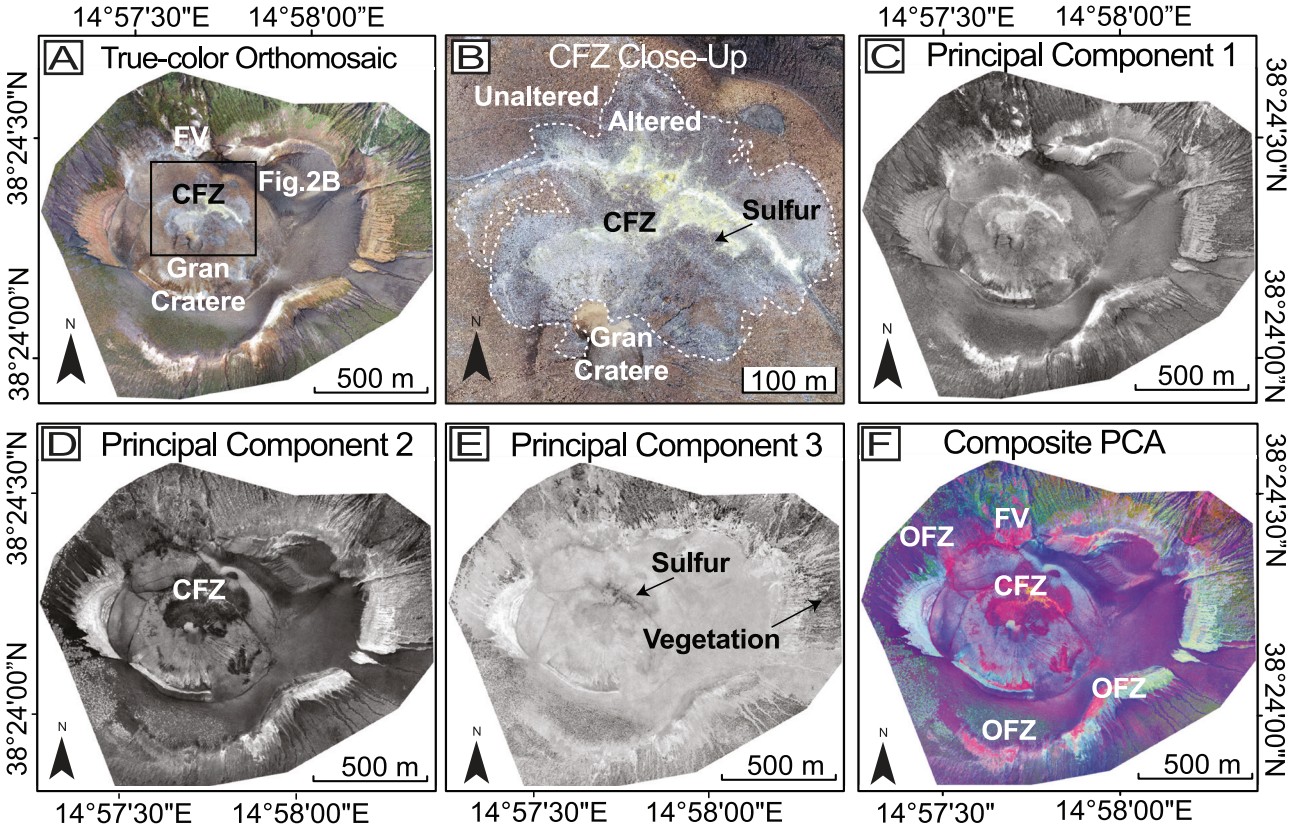

**Fig. 2 | Orthomosaic and principal components (PCs). A** True-color orthomosaic of the study area, providing a high-resolution reference. **B** Close-up view of the CFZ with a dashed line representing the contact between the altered and unaltered surfaces. **C** PC1 highlights the most variance within the dataset; **D** PC2 emphasizes the second most variation in the data and highlights color variance, notably at the CFZ; **E** PC3 contains the least amount of variance in the data and highlights the vegetation and sulfur deposits in dark gray; **F** composite PCA image enhancing visualization between all principal components, and effectively highlights the CFZ and OFZ and other altered regions in red-to-pink colors.

## Table 1 | Principal component analysis eigenvalue statistics from all single PCs

| Principal component (PC) # | Eigenvalues | % Eigenvalues | Accumulation of eigenvalues |
|---|---|---|---|
| PC1 | 6305.10683 | 95.6591 | 95.6591 |
| PC2 | 181.13775 | 2.7482 | 98.4073 |
| PC3 | 104.97993 | 1.5927 | 100.0000 |

The table shows the statistical summary of the eigenvalues, percentage of variance explained, and cumulative variance for PC1, PC2, and PC3.

with a resolution of 7.6 × 7.6 cm, providing a comprehensive view of the La Fossa cone (see "Methods": *UAV-remote sensing*). This map also includes less accessible and less explored regions, such as the 1988 landslide on the northeasternmost edge of the island and the steepest areas along the north flank and Forgia Vecchia (Fig. 1A, C, E). The true-color orthomosaic (Fig. 2A) reveals regions indicative of hydrothermal alteration, characterized by a heterogeneous distribution of pale gray to yellowish and white colors, as suggested by previous studies[49]. This discoloration, particularly along the summit and crater rims, resembles typical hydrothermal alteration patterns observed at the summits of other active volcanoes. At La Fossa, most of the alteration is concentrated along the upper rim of the Gran Cratere and outer, older crater rims. Some alteration sites are extensive, stretching over 425 m long and 145 m wide. Interspersed sulfur deposits concentrated within the CFZ were identified based on yellow coloring in the true-color orthomosaic, confirmed through field observations and consistent with previous studies[26] (Figs. 1D and 2A, B). The alteration spreads inward from the CFZ on the northern crater crest and the inner crater wall, terminating to the south at the crater floor. In all other directions, the brightly-colored altered material sharply transitions into reddish-brown colored material, marking the contact between altered and unaltered regions (Fig. 2B).

Additionally, bleached material is observed along the northern outer flank, in the Forgia Vecchia region, where the altered material fans westward and descends toward the village of Vulcano (Figs. 1C and 2A). Other altered areas are also observed on the southern flanks and within the boundaries of the 1988 landslide areas. These regions, characterized by hydrothermal deposits and alteration, are often challenging to map accurately using true-color data alone, highlighting the need for a more effective feature extraction method. Consequently, we applied a statistical technique (principal component analysis (PCA); see "Methods": *Principal component analysis*) to reduce data dimensionality and isolate spectral variations related to hydrothermal deposits, thereby enhancing our mapping efforts.

Our PCA resulted in three principal components (PCs) (PC1, PC2, and PC3), which account for approximately 95.6%, 2.7%, and 1.6% of the total variance in the data, respectively (Table 1). PC1 captures overall surface brightness and highlights the OFZ and also alteration on the outer flanks; however, it lacks the necessary color change detection for outlining the CFZ (Fig. 2C). PC2 highlights the CFZ as dark gray, interpreted as altered material, and clearly distinguishes it from the surrounding lighter gray, unaltered surfaces based on color transitions (Fig. 2D). However, PC2 fails to highlight altered regions on the outer flanks. PC3 accounts for the smallest portion of the total variance and highlights the features associated with vegetated regions along the lower outer flanks and localized sulfur deposits within the main fumarole field (Fig. 2E). These features highlighted in each PC are expressed at different grayscale values. Some regions are emphasized by higher (brighter) values (e.g., PC1), while others, such as the CFZ in PC2 and vegetation and sulfur in PC3, are more prominent at lower (darker)

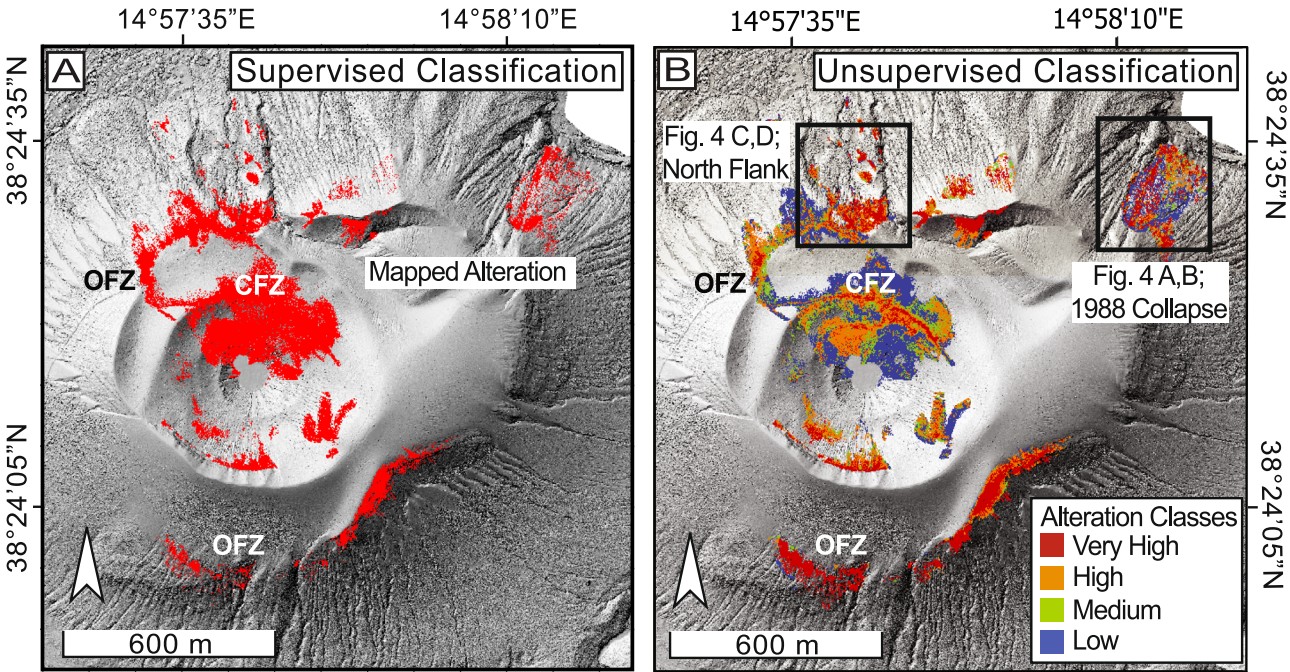

**Fig. 3 | Hydrothermal alteration maps. A** Supervised classification, **B** unsupervised classification, with black squares representing areas shown in detail in Fig. 4A–D.

values. While each component captures useful surface characteristics, none were sufficient for alteration mapping across the entire region when used individually. Therefore, we excluded the individual PCs from single-component analysis and instead used a composite of all three to support classification and interpretation.

By combining these components, we developed a false-color composite raster by stacking PC1, PC2, and PC3 into the red, green, and blue channels, respectively (Fig. 2F). Each PC contributed distinct visual information to the composite, and their combination highlighted distinct features associated with alteration. The resulting red-to-pink hues effectively delineate the CFZ and highlight features that are less discernible in the true-color orthomosaic. The red-to-pink regions include the CFZ and OFZ regions (Fig. 2E), Forgia Vecchia, the 1988 landslide, and the sections affected by the 2010 landslide. These regions reflect silicic to advanced argillic alteration, consistent with previous observed patterns of hydrothermal alteration zonation at Vulcano[49,61]. The blue to blue-purple tones correspond to unaltered regions, while green hues emphasize varying vegetation in the lower-flank, and yellow hues highlight sulfur deposition within the CFZ. This differentiation between altered, unaltered, and vegetated surfaces results from the PCA's ability to enhance spectral variations associated with surface alteration and bleaching, thereby maximizing variance between surface types, and allowing hydrothermally altered areas to be separated from other materials[62]. Consequently, regions highlighted in red-to-pink within the composite raster correspond to hydrothermally altered surfaces, making the PC composite raster a suitable candidate for feature extraction.

**Alteration classes**

Based on our PCA composite (Fig. 2E), we applied a supervised classification using the Maximum Likelihood algorithm in ArcGIS Pro to delineate and highlight all regions of hydrothermal alteration (see "Methods": *Supervised and unsupervised classification of alteration zones)*. We selected 50 representative training sites within the red-to-pink hues of the PCA composite that coincided with the CFZ, a well-documented area of hydrothermal activity where elevated acid gas fluxes promote chemical leaching and surface bleaching[30,63]. This approach identified ~0.15 km² of altered surface, including both expected areas at our training sites and the CFZ, as well as several unrecognized or inaccessible regions along the northern, northeastern, southern, and eastern outer flanks (Fig. 3A).

Additionally, the method highlighted locations associated with past mass-wasting events, notably those in 1988 and 2010 (Fig. 3B).

Narrowing the study area to a defined alteration zone enabled a subclass refinement using an unsupervised clustering approach. This highlighted region corresponds to areas of advanced argillic to acid-sulfate alteration, as previously identified at La Fossa[57]. Our unsupervised classification was performed in ArcGIS Pro (see "Methods": *Supervised and unsupervised classification of alteration zones)*, which segmented our alteration zone into four subclasses: Low, Medium, High, and Very High, based on the intensity of discoloration or bleaching. These subclasses reflect the increasing levels of hydrothermal alteration intensity, as inferred from the progressively brighter bleached surfaces across the region. The number of classes was predefined, and thresholds were derived from the statistical clustering of RGB pixel values. The resulting mean RGB values for each subclass were (147, 150, 155) for Low, (167, 170, 173) for Medium, (182, 186, 186) for High, and (219, 220, 216) for Very High, demonstrating a progressive shift toward brighter, whiter surfaces with increasing alteration.

These subclasses are heterogeneously distributed across the La Fossa cone, with several regions containing all four subclasses (Fig. 3B). The surface areas covered by each subclass are approximately 0.0473 km² (Low), 0.0187 km² (Medium), 0.0463 km² (High), and 0.0371 km² (Very High). The CFZ exhibits the most significant variation in alteration distribution and contains the largest area of Low-grade alteration (Fig. 3B in blue). The Low subclass nearly encircles the CFZ and is juxtaposed with all other alteration types. These spatial patterns align with findings from Müller et al.[30], who mapped hydrothermal alteration at La Fossa and validated their classifications using geochemical and mineralogical analyses. Their study showed a systematic increase in sulfur content and a decrease in $Fe_2O_3$ across alteration gradients, further supporting the interpretation that our subclasses reflect meaningful variations in alteration intensity.

While the High alteration category is predominantly concentrated within the CFZ, clusters are also observed along the flanks (Fig. 3B in orange). In these flank regions, areas are often classified as High, with adjacent zones categorized as Very High. The Very High category, representing the most intensely altered rocks, is observed within the CFZ and is predominantly found along the steepest sections of the northern and

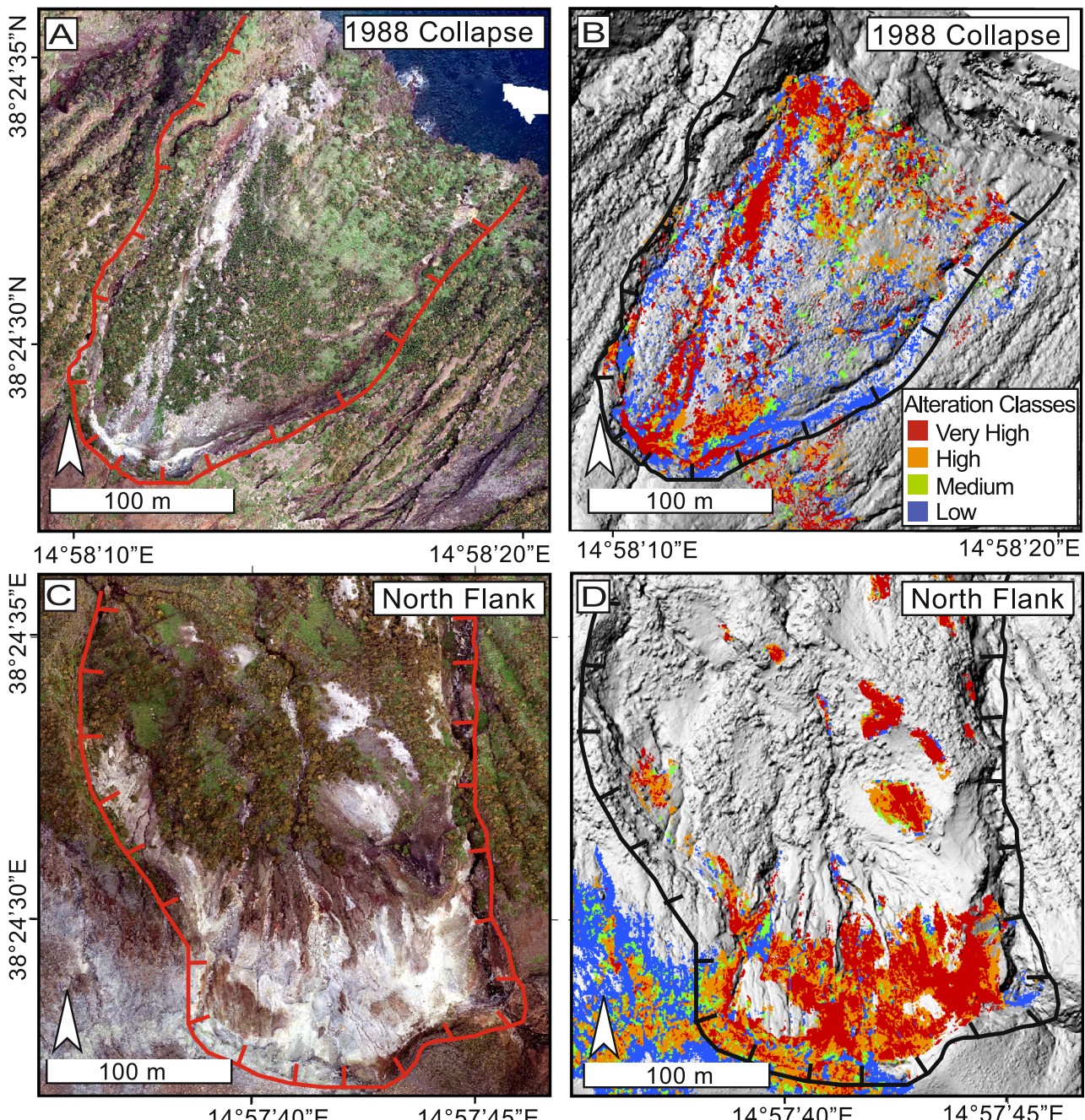

**Fig. 4 | Close-up views of selected flank regions.** The images focus on regions with a history of mass wasting and hydrothermally altered surfaces. **A** A true-color close-up of the 1988 landslide scar. **B** Close-up of the 1988 landslide with alteration classifications. **C** True-color close-up of the north flank (landslide scarp modified from Madonia et al.[29]). **D** Close-up of the north flank with alteration classifications.

southern flanks and in the 2010 and 1988 landslide regions (Figs. 3B and 4A–D in red). At the 1988 landslide site, a distinct region of Very High alteration extends in a north-northwest direction, closely following a gully. This may, in part, reflect the downslope transport of altered material from higher elevations. However, similarly altered surfaces also extend beyond the gully, suggesting that this feature likely reflects both in situ and remobilized altered material. These subregional classifications, derived from an unsupervised color-based analysis, effectively highlight the extent of hydrothermal alteration (see Supplementary Video 1 for visual reference). The well-defined transitions between alteration types serve as a framework for integrating and transposing Schmidt hammer rebound values measured in the field, contributing to refining our thematic map of relative rock strength and stability.

### Alteration classification and Schmidt rebound value correlation

Schmidt hammer measurements provide an $R$ value, representing the rebound recorded upon impact, and correlate empirically with UCS, a commonly used metric for rock strength, through material-specific conversion curves[64–67]. For simplicity, we refer to the $R$ value as the "Schmidt rebound value" hereafter. Since no UCS conversion was applied for this study, our Schmidt rebound values served as a proxy to assess relative rock strength. Our Schmidt hammer campaign comprised approximately 1100 measurements collected along multiple transects[68] (Fig. 5A–E). The primary focus was to assess Schmidt rebound values across various alteration intensities and environments (see "Methods": *In situ rock hardness assessment*). The largest transect, oriented north-south, extended toward the northern flank of Forgia Vecchia. This transect encompassed a range of

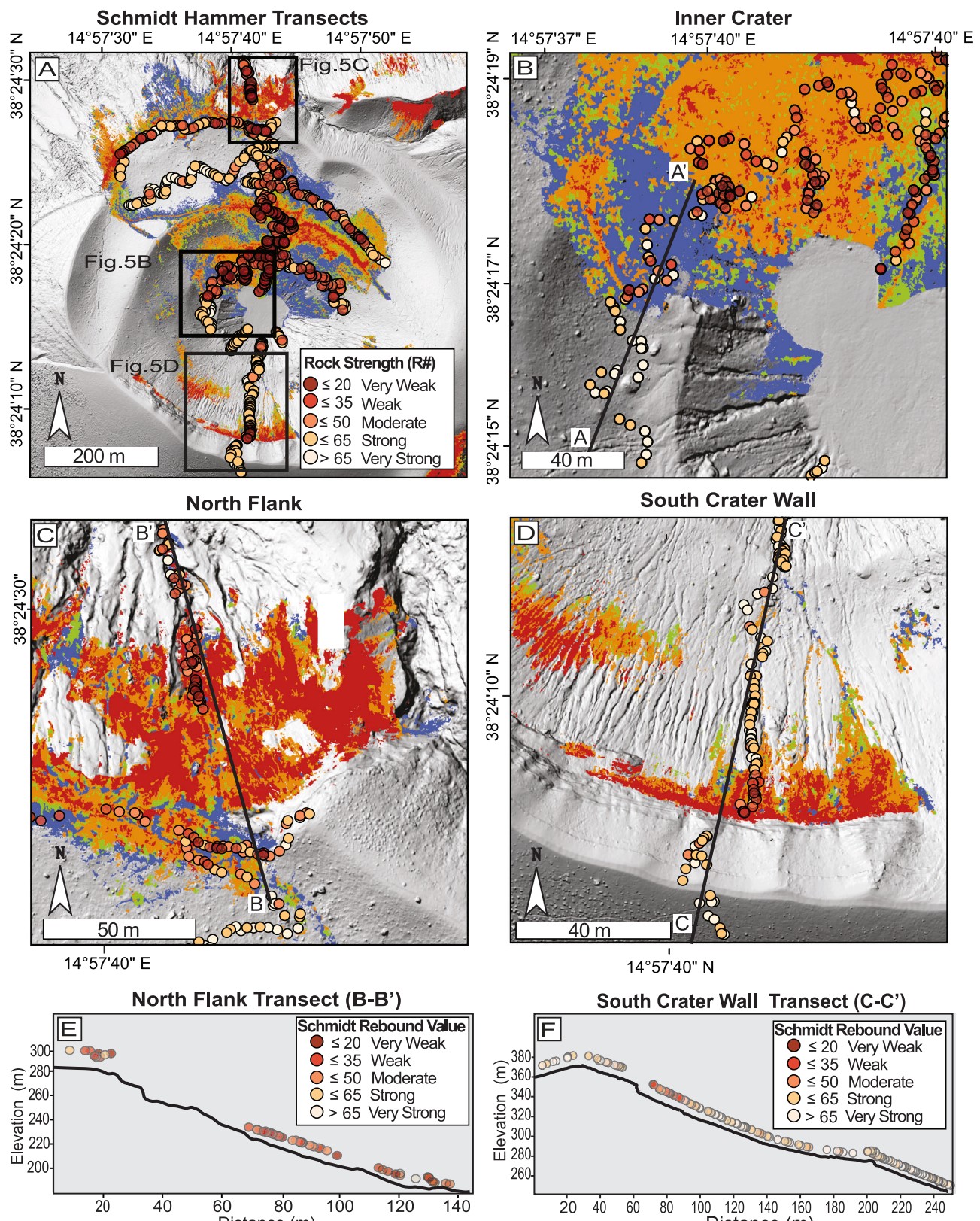

**Fig. 5 | Schmidt hammer measurements across alteration classes. A** Distribution of all Schmidt hammer measurement sites across the field area, black boxes correspond to close-up views in (**B**–**D**). **B** Close-up of a sub-section of the CFZ representing the sharp transition from high to low Schmidt values as the A-A' transect crosses into altered territories (see also Supplementary Fig. 1). **C** Close-up of measurement locations on the north flank, with the black line indicating the elevation profile shown in (**D**). **D** Measurement locations on the south flank, with the black line marking the elevation profile shown in (**E**). **E** Elevation profile of the north flank (black line) with overlying Schmidt rebound value locations. Schmidt measurements are color-coded to indicate relative rock strength: Very Strong, Strong, Moderate, Weak, and Very Weak. **F** Elevation profile of the south flank (black line) with corresponding Schmidt hammer measurements, similarly color-coded by relative rock strengths.

geological features, including the outermost flanks, inner crater walls, the fumarole field, and, in part, the rim of the collapsing northern flank, providing extensive surface coverage. In addition to the north-south transect, three east-west transects were conducted. The northernmost east-west transect measured rocks along or near the edge of the northern flank. The middle transect captured the largest amount of Unaltered material and traversed the northern section of the CFZ, including rocks classified as Low-grade altered material. The smallest, innermost transect measured rocks containing the highest concentration of altered material and the largest amount of High-grade altered material. A focused subset of this transect, denoted A-A' (Fig. 5B), highlights a sharp transition in Schmidt rebound values (see Supplementary Fig. 1 for corresponding transect values). As the transect progresses from southwest to northeast, relative rock strength drops abruptly as the transect crosses the contact of the CFZ. While this pattern is clearly seen specifically at this location, other transects do not exhibit the same abrupt change in relative rock strength, but instead show a more heterogeneous distribution of values across mapped alteration. Nonetheless, the pattern shown demonstrates a clear spatial correlation between mapped alteration zones and in situ mechanical weakening, reinforcing the utility of our combined mapping and field measurement efforts.

The main north-south transect proved valuable, capturing relative rock strength across some of the flank's steepest and most altered regions. For instance, the southern inner crater wall, corresponding to the C-C' elevation profile (Fig. 5D), exhibited a distinct gradient in Schmidt rebound values along its transect (Fig. 5F, E). Rocks at the unaltered southern rim were among the strongest measured in the field, with Schmidt rebound values in the Very Strong ($R > 65$) to Strong ($R \leq 65$) range. As the transect progressed northward toward the steepest section of the inner crater wall, it intersected surfaces mapped as Very High and High. This area coincided with a decline in relative rock strength, showing of Weak ($R \leq 35$) and Very Weak ($R \leq 20$) Schmidt hammer values (Fig. 5E). Further downslope, relative rock strength increased again as the material transitioned back into Unaltered material (Fig. 5F).

A similar pattern was observed on the outer northern flank, where the weakest rocks were concentrated in the steepest sections (Fig. 5C). Along the B-B' elevation profile, near the edge of the landslide scar, Schmidt rebound values predominantly fell within the Moderate ($R \leq 50$) to Very Weak ($R \leq 20$) range, indicating decreased relative rock strength at the scarp crest. As the B-B' profile progresses northward toward the lower portion of Forgia Vecchia and, within the mapped alteration, most rocks fell into the Very Weak ($\leq 20$) to Weak ($\leq 35$) categories. However, at the lowest elevations, Schmidt rebound values exhibited greater heterogeneity (Strong to Very Weak), likely due to increased variation in alteration intensity and localized geological influences, particularly mass wasting, which transports rocks with differing alteration types downslope (Fig. 5E). Rocks classified as High and Very High alteration grades are primarily located in the CFZ (Fig. 5A, B), where many of the weakest rocks documented during the campaign were found. These alteration grades were also the dominant surface type on the northern and southern flanks; however, their spatial extent was more limited compared to the CFZ, resulting in fewer measurements from those areas. Nonetheless, measurements taken near the sites of earlier flank failures consistently recorded low Schmidt rebound values, indicating a concentration of some of the softest and weakest rocks in our field area.

Overall, the average Schmidt rebound values measured in the field ranged from 10.5 to over 82, showing high variability in relative rock strength across the study area. This variation is observed across all surface classifications, from Low to Very High alteration grades throughout the field. Despite the widespread variability in Schmidt rebound values, our summary statistics (mean, quartiles, and 95th percentile, and bootstrapped 95% confidence intervals (CIs)) reveal a distinct pattern: Schmidt rebound values decrease systematically with increasing alteration intensity. Unaltered material, marked in gray (Fig. 6), accounts for 461 locations ($n = 461$), or 44% of our dataset, and consistently exhibits the highest rebound values, averaging 60.1 with a median of 63 [95% CI: 62.00–65.00], representing the strongest rocks in our field area. Although altered surfaces collectively account for a comparable number of measurements, individual alteration grades are less represented and exhibit progressively lower rebound values. Specifically, the Low ($n = 216$), Medium ($n = 80$), High ($n = 203$), and Very High ($n = 78$) grades have average Schmidt rebound values of 42.8, 36.2, 34.8, and 31.1, respectively. Very High alteration exhibits the lowest values, with a median of 27.0 [95% CI: 23.75–30], representing the weakest material observed in the field. This decrease in Schmidt rebound values is consistent across all interquartile ranges for every surface type.

The 95% CI illustrates statistically distinct group medians between Unaltered and all altered categories, with no interval overlap[69]. In contrast, overlapping intervals among Medium, High, and Very High alteration classes indicate greater dispersion and reduced statistical separability within these groups that contain a higher degree of alteration. Our Kruskal–Wallis test confirms that differences in Schmidt hammer values across alteration classes are statistically significant ($p < 0.0001$). Post hoc pairwise comparisons with Bonferroni correction reveal that Unaltered rocks significantly differ from all alteration categories, and that Low alteration is also significantly different from both High and Very High classes ($p < 0.001$) (see "Methods": *In situ rock hardness assessment*). However, Medium, High, and Very High groups are not statistically separable, consistent with increasing overlap in rebound distributions at greater alteration intensities (Fig. 6).

Overall, this represents a 48.25% decrease in relative rock strength from Unaltered areas to those exhibiting the highest degree of alteration, as indicated by their level of bleaching. We note that the same Very High alteration class is found not only along the northern outer flanks and inner walls of Gran Cratere but also on the southern outer flanks of the edifice, which may represent zones of potential flank failure. The violin plot (Fig. 6) provides a statistical summary of the Schmidt rebound values with associated alteration classifications, illustrating the decreasing relative rock strength trend with increasing intensity. The Very Weak rebound values are spatially associated with High to Very High alteration, including both regions of past mass-wasting and intact slopes showing signs of surface weakening. This spatial alignment corresponds to a localized zone of reduced strength consistent with mapped alteration intensity.

## Discussion

Volcanoes are traditionally governed by phases of construction and destruction[70], during which topographic growth and slope steepness lead to structural instability and mass-wasting events. However, increasing evidence suggests volcanic instability can occur during quiescent periods, particularly during degassing phases. This phenomenon has been observed in several volcanic systems, including Soufrière Hills volcano (Montserrat, Eastern Caribbean)[71], Tutupaca (Peru)[70], Casita (Nicaragua)[72], and Mt. Rainier (USA)[73], where destabilization has been linked to prolonged hydrothermal activity and alteration processes.

Hydrothermal alteration is a widespread process that affects volcanoes across diverse geological settings and volcanic types and is detectable by various methods[74]. Our field-based Schmidt hammer measurements at La Fossa (Fig. 6) demonstrate a clear and statistically significant decline in relative rock strength with increasing alteration intensity, with an approximate 50% reduction in rebound values between Unaltered and Very High alteration classes. This trend is statistically supported by both Kruskal–Wallis and post hoc test results: Unaltered rock significantly differs from all altered categories ($p < 0.0001$), and Low alteration differs from High and Very High ($p < 0.001$). In contrast, differences between Medium, High, and Very High classifications are not statistically significant, suggesting increasing overlap at the higher alteration intensities.

The consistent, non-overlapping 95% CIs between Unaltered and altered classes further reinforce the strength of this interpretation. For example, Unaltered material has a median rebound value of 63 with a 95% CI of [62.00, 65.00], while the Very High alteration has a median of 27 and a CI of [23.75, 30.00], with no overlap between these intervals. A 95% CI provides a range, defined by lower and upper bounds, within which the true population median is expected to fall with 95% confidence, assuming random sampling. Because our Schmidt rebound values in this study are widely

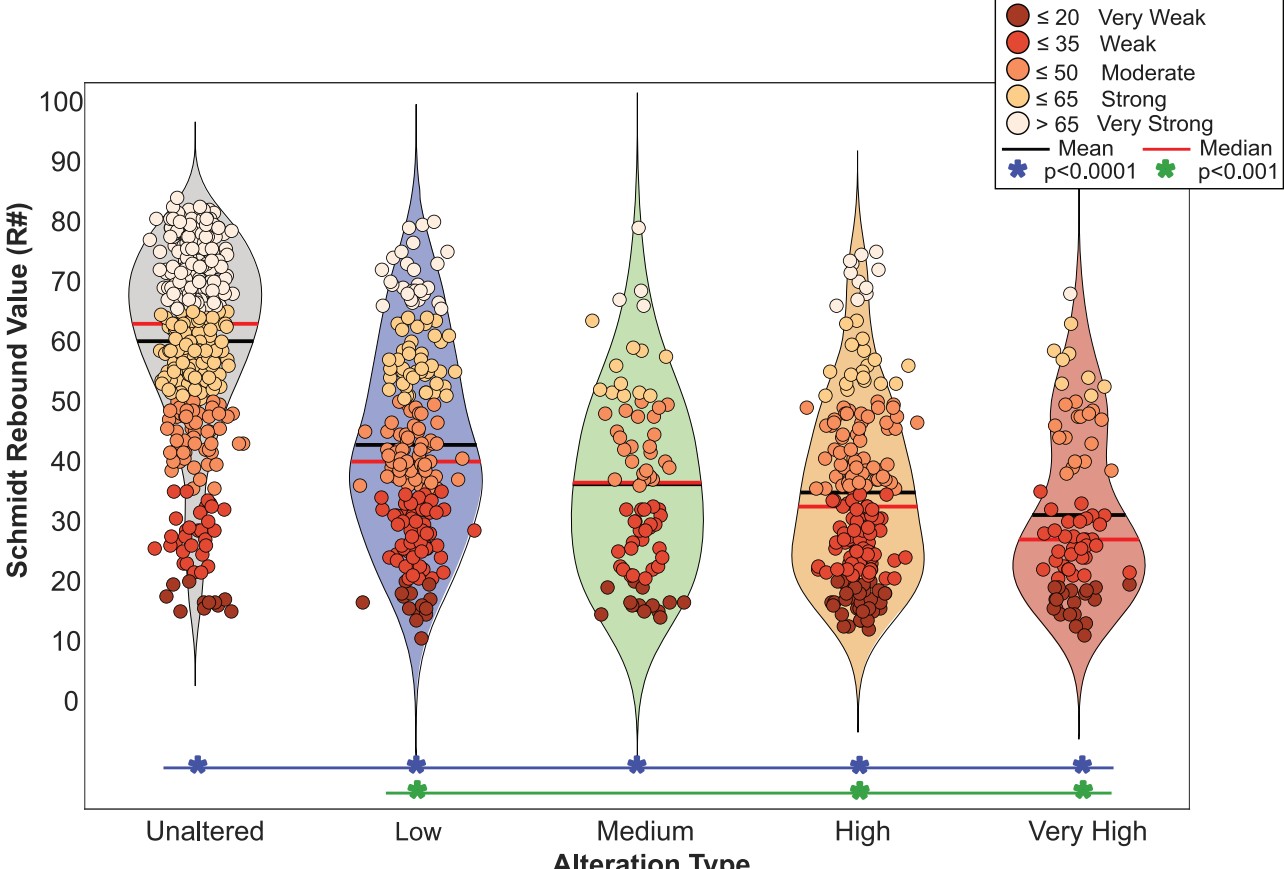

**Fig. 6 | Schmidt rebound value vs. alteration types.** The violin plot illustrates the Schmidt hammer rebound values distribution across different alteration types. The red line on each violin represents the median, while the black line indicates the mean. Data points ($n = 1038$) are color-coded by Schmidt rebound value, and the color of each violin corresponds to the alteration grades in Figs. 3–5. The width of each violin reflects the density of measurements. To assess statistical significance, we applied a non-parametric Kruskal–Wallis test, followed by Bonferroni-corrected post hoc pairwise comparisons. Blue line and asterisks indicate pairwise comparisons significant at $p < 0.0001$, and green lines and asterisks denote comparisons with $p < 0.001$.

distributed, bootstrapped CIs offer a reliable approach for meaningful comparison between groups.

Although these values provide relative rock strength rather than absolute strength, the observed ~50% reduction in strength aligns well with past laboratory-based studies, which observed a similar trend in rock strength weakening of hydrothermally altered volcanic rocks. For example, at La Soufrière de Guadeloupe (Eastern Caribbean), two independent studies reported a decrease in rock strength with increasing alteration: Heap et al.[17] measured UCS values decreasing from 140–60 MPa to less than 10 MPa, and Poganj et al.[75] found average values of ~82 MPa for pristine rock and ~6.5 MPa for extremely altered rock, showing that rock strength decreases systematically with the increasing Alteration Grade Index, a visual metric incorporating discoloration. Similarly, at Merapi (Indonesia), Darmawan et al.[26] reported a decrease in UCS values from ~132 MPa in fresh dome rocks to ~11 MPa in fully altered rocks, correlated with an increase in porosity, an increase in $\delta^{18}O$, and, most relevant for this study, an increase in rock discoloration. These parallels support the use of the Schmidt hammer rebound values as a practical, field-based proxy for assessing alteration-driven rock strength weakening.

While our interpretation assumes that low Schmidt hammer rebound values reflect pre-collapse weakening, we acknowledge other alternative mechanisms, such as post-failure weathering or clast transport, that could influence our measurements. Although we are confident that the reduced Schmidt rebound values result from hydrothermal alteration, we cannot entirely rule out the contribution of weathering in rocks exposed at the surface for prolonged periods. However, the consistent association of low

Schmidt values with mapped High or Very High alteration zones supports a pre-collapse origin.

Consistent with Poganj et al.[75], our data (Fig. 6) underscores the large heterogeneity in rock strength resulting from ongoing hydrothermal alteration, which is a key factor in slope stability and collapse. This spatial variability complicates slope-stability modeling and emphasizes the need for high-resolution, spatially continuous techniques. Our classification and strength maps reveal diffuse heterogeneity across the edifice, reinforcing the importance of spatially resolved approaches for characterizing volcanic instability. Although our Schmidt hammer measurements are not UCS-calibrated, they effectively capture spatial patterns of mechanical weakening. This strengthens our approach and highlights the Schmidt hammer as a powerful tool to streamline future fieldwork by guiding sampling strategies and prioritizing zones for a detailed mechanical analysis. Future work could build on this foundation by incorporating targeted sampling at mapped alteration sites to validate the classification scheme and further constrain the links between mineralogy, porosity, and mechanical weakening. Our integrated UAV photogrammetry, PCA, and classification technique proved especially valuable in steep, hazardous terrain, where direct access and assessment are limited. This capability is critical in volcanic environments, where hydrothermal alteration, a key driver of flank instability, often occurs in inaccessible areas. As such, our workflow reinforces the role of remote sensing techniques as effective, practical tools for identifying hydrothermal activity and alteration zones. These tools enable detailed, spatially comprehensive assessments of hydrothermally altered regions, supporting subsurface inferences that can enhance hazard assessment and monitoring

**Fig. 7 |** Conceptual illustration of slope instability processes associated with hydrothermal alteration and zonation, highlighting how shallow processes (e.g., landslides or flank collapses) can remove overlying material and expose alteration at the surface.

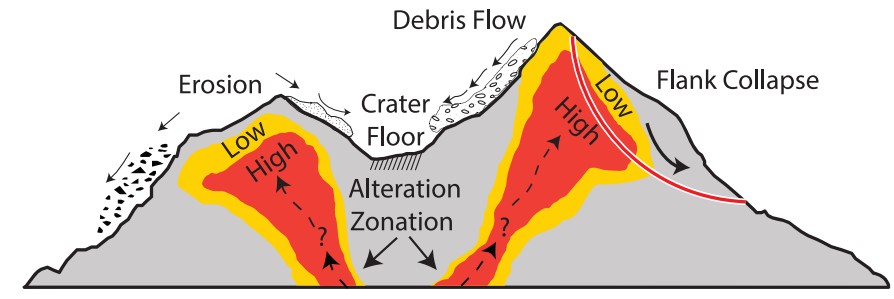

efforts. While porosity and mineralogical data are often used to complement mechanical interpretations, such measurements were beyond the scope of this field-based study. Nonetheless, the observed trend weakening correlates with visible discoloration and mapped alteration zones, which closely align with works at other volcanoes that included porosity data, providing independent support for the interpretation of hydrothermal-induced rock weakening[22,26,75].

This study focuses on Vulcano Island as a case study, demonstrating the potential of integrating field-based rock strength investigations with remote sensing techniques. By combining datasets from both approaches, we present a framework for assessing volcanic stability in regions otherwise inaccessible for direct sampling, allowing for a complete and comprehensive dataset. While this workflow is broadly applicable, it has operational limits. The workflow effectiveness depends on surface exposure, image resolution, and spectral distinguishability of alteration intensity. These limitations may reduce classification accuracy, particularly in shadowed or densely vegetated areas.

## Implications

This study demonstrates the effectiveness of integrating UAV photogrammetry, PCA, and Schmidt hammer testing to map and assess hydrothermal alteration and its impact on rock strength. In some cases, a landslide or collapse event can excavate and expose deeper, previously buried zones of hydrothermal alteration (Fig. 7). Our findings emphasize the impact of hydrothermal alteration on reducing slope stability and increased erosion, reinforcing its significance for volcanic hazard assessment. The method used here successfully identified and mapped otherwise inaccessible regions of alteration along the steep flanks of the La Fossa cone, offering a broader understanding of the structural weakening associated with volcanic alteration processes. Using the Schmidt hammer, we developed a thematic map illustrating spatial variations in rock strength across classified alteration units. This application parallels other techniques, such as exposure-age dating, where the relative exposure age is inferred by measuring the relative rock strength, as weaker rocks typically indicate prolonged exposure to weathering processes[76].

Beyond La Fossa, our findings support improved volcanic flank instability monitoring, particularly in steep, inaccessible regions where direct field access is challenging. Our method enhances hazard assessment and mitigation by enabling detailed flank monitoring by identifying areas of reduced rock strength. While this study focused on La Fossa, the approach has strong potential for broader applications to other volcanoes, particularly those with restricted direct field access. For example, at Merapi (Indonesia), previous studies were limited to sample collection from the crater rim and flanks due to the inaccessibility of the crater dome, which was the primary region of deformation[26,77]. Similarly, at Volcán de Colima, rock samples were collected and analyzed only after a mass-wasting event, with sampling restricted to hydrothermally altered deposits within the debris-flow track rather than in situ sampling[24]. Because direct access to these locations is often limited, drone-based photogrammetry offers a practical alternative for continuous monitoring, allowing for more frequent data collection while minimizing risks.

Interestingly, once the relationship between alteration class, as determined by drones, and relative rock strength, as determined by the Schmidt hammer, has been established, a time series of data can be generated. For example, suppose the violin diagram relationship is established for the dome region of Merapi or Colima; then, while using drones and the PCA classification approach, this relationship can be extended through time. In that case, one can continue monitoring the changes in apparent strength and slope stability of the evolving volcano by using repeated drone surveys. In addition, as suggested at Merapi, areas of alteration and mechanical weakness may be covered by younger volcanic deposits, indicating that mass-wasting may occur even in the absence of visible alteration. Once calibrated to rock strength values, the remote sensing approach allows us to monitor the evolution of a growing lava dome and its hydrothermal alteration, with substantial implications for dome deformation and pyroclastic density flow generation[78]. While our workflow is not a real-time monitoring tool in the strictest sense, it enables high-frequency, repeated assessments of alteration and flank conditions. More broadly, it lays the foundation for a workflow that, once refined, could serve as a near-real-time proxy for mechanical weakening in evolving volcanic settings, offering a valuable, low-risk complement to traditional monitoring techniques, particularly where access is restricted.

Additionally, the techniques used in this study can be integrated with other volcanic mapping and monitoring methods, such as hyperspectral alteration mapping. For instance, previous work on Mount Ruapehu (New Zealand), aeromagnetic surveys, hyperspectral hydrothermal mapping, and laboratory-based rock mechanical analysis were used to model potential rock failures. Incorporating our method could enhance such studies by providing additional inputs for stability modeling, further refining, and fine-tuning model hazard assessments[37]. Moreover, this approach enables the identification of structurally weakened areas that can guide future sampling strategies, where access permits for detailed strength analysis, including laboratory testing such as UCS. Applying our approach in other active volcanic regions could enhance hazard assessments and monitoring strategies, providing critical insights into hydrothermal alteration in volcanic edifice destabilization.

## Conclusions

Hydrothermal alteration across the La Fossa cone is spatially heterogeneous in both extent and intensity, with the most pronounced alteration occurring along the crater rim near the fumarole field and on parts of the cone's flanks. By integrating UAV-based photogrammetry, PCA, and supervised and unsupervised classification techniques, this study effectively mapped alteration patterns and delineated classes based on discoloration intensity linked to rock bleaching. Our PCA-based approach, particularly using a composite PC raster, proved more effective than single-component extractions, allowing for more accurate detection of alteration zones within the central fumarole field and along the flanks. While single PC feature extraction (e.g., PC2) can effectively highlight alterations in smaller, well-defined areas, it was less effective across the broader and more diverse terrain of La Fossa, where misclassification of unaltered regions was more frequent. The composite raster approach

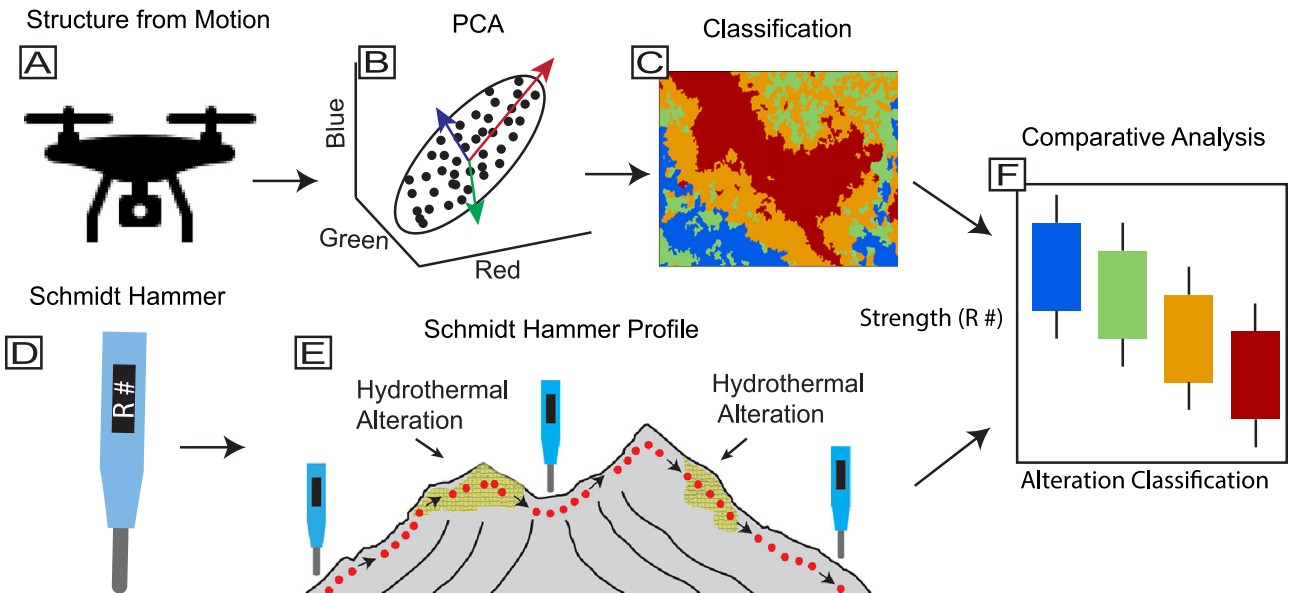

**Fig. 8 | A simplified sequential methodological workflow was used in this study.**
**A** Drone photogrammetry data acquisition and Structure-from-Motion (SfM) processing. **B** Principal component analysis (PCA) for feature extraction. **C** Classification and mapping of hydrothermal alteration sites. **D** Schmidt hammer data acquisition, where *R* values represent the rebound measurements recorded by the instrument. **E** Illustration of a Schmidt hammer profile across La Fossa cone, with altered regions highlighted in yellow. **F** Box plot comparing rock strength across different alteration classes.

overcame these limitations, allowing for a clearer distinction between altered and unaltered surfaces.

In conjunction with in situ Schmidt hammer measurements, our classification of altered zones revealed a consistent inverse relationship between alteration intensity and rock strength. Unaltered rocks exhibited the highest Schmidt rebound values, while rocks classified under Very High alteration grades showed a nearly 50% reduction in relative strength. Despite the variability in our Schmidt hammer measurements (10.5–82 R), a large dataset and holistic analysis approach showed that relative rock strength decreased as a function of alteration intensity. Regions such as the base and lower portions of the northern flank showed higher-than-average variability. These anomalies could reflect the downslope transport of weaker material due to mass movement processes.

Our classification of altered rocks aligns with strongly eroded slopes and notably with known historic mass-wasting events, such as the 1988 landslide and the Forgia Vecchia region. This correlation reinforces previous works demonstrating how hydrothermal alteration contributes to structural weakening and potential flank failure. Notably, Very High alteration regions correlate with the lowest rock strength values, suggesting heightened susceptibility to future slope failure.

Beyond La Fossa, this study highlights the broader implications of hydrothermal alteration as a key driver of volcanic instability. Our integrated remote sensing and geomechanical methodology offers a scalable approach to hazard assessment, particularly in inaccessible or hazardous terrains for direct fieldwork. While the applicability of our approach is constrained by operational factors such as surface exposure, image resolution, and spectral distinctiveness within the mapped alteration, the correlation between mapped alteration classes and in situ rock strength underscores the robustness of this method as a proxy for identifying mechanically weakened regions. The absence of subsurface constraints (e.g., geophysical or petrological data) limits our ability to assess depth-dependent weakening; however, this limitation reflects a common challenge in many volcanic settings, where only surface outcrops are available. Within this context, our protocol remains well-suited for field-based hazard evaluation. By enabling the direct detection of structurally weak zones, this framework overcomes many limitations of traditional geotechnical methods and can be applied to other volcanic systems worldwide, providing critical insights into

the role of hydrothermal alteration in the long-term stability of volcanic edifices.

## Methods

Here, we integrate classification results from close-range photogrammetric data with in situ Schmidt hammer measurements. The combined analysis facilitates a detailed investigation of regions of hydrothermal alteration and their corresponding physical attributes by mapping and categorizing the altered areas and evaluating their respective rock strengths. Fig. 8 presents a simplified workflow; the full method is described in detail below.

### UAV-remote sensing

Recent advancements in UAVs have greatly improved volcanic landform and hazard mapping by enabling high-resolution, cost-effective, and frequent data acquisitions[79]. While satellite and airborne remote sensing remain essential for large-scale studies, UAVs provide greater flexibility, higher spatial and temporal resolution, and real-time data collection, making them particularly effective for monitoring localized areas[80,81]. Integrating UAV technology with Structure from Motion (SfM) techniques has further improved aerial triangulation and high-resolution orthomosaic generation, enabling precise characterization of volcanic surfaces[82].

Traditionally, image triangulation methods required manual calculations of camera positions and 3D points, which limited the accuracy and efficiency of UAV-based mapping[82]. However, advancements in computer vision, onboard GNSS receivers, and SfM algorithms now enable automatic feature recognition and matching across multiple images. These features are tracked throughout the dataset, allowing for precise camera position estimation and 3D scene reconstruction. A non-linear least squares minimization technique iteratively refines the reconstruction, optimizing spatial accuracy[81,83]. The SfM algorithm then aligns these features and stitches together individual images, resulting in a high-resolution orthomosaic[81]. These advancements facilitate the creation of a highly accurate, georeferenced orthomosaic with minimal manual intervention, providing a reliable dataset for further analysis.

For this study, we selected drone imagery from a previous survey conducted over La Fossa in 2019[49], which we reprocessed for this study[68].

This particular survey was conducted during a period of reduced degassing and minimal cloud cover, providing optimal conditions for high-quality image acquisition. The survey was conducted using a DJI Phantom 4 Pro quadcopter equipped with a gimbal-stabilized 20-megapixel camera (Fig. 8A), flown at ~150 m above ground level in a grid pattern to achieve complete coverage of the study area. Pre-processing steps were applied to filter and remove images affected by excessive gas plumes, incorrect flight heights, and visible horizons, minimizing alignment errors. The remaining images were then processed using the SfM software Agisoft Metashape (Version 1.5.2.7838) to create a 3D model, digital elevation model, and orthomosaic (Fig. 8A).

### Principal component analysis (PCA)

Despite these advancements in volcanic mapping and monitoring, challenges remain in photogrammetric analysis. One key issue is the homogeneous appearance of volcanic surfaces, often dominated by dark to grayish tones, making color anomaly detection difficult. Therefore, the limitations of true-color imagery in distinguishing alteration zones necessitated the application of a more advanced feature extraction technique. To overcome this limitation, we used PCA, a statistical method widely applied in remote sensing to reduce data redundancy while preserving spectral variation[84]. PCA transforms the original RGB data from our true-color orthomosaic into eigenvectors, known as PCs (Fig. 8B), which are linear combinations of the original RGB data weighted by transformation coefficients[85]. These coefficients are derived from the covariance matrix of the original RGB data. Thus, PCA generates an image representing a linear combination of the original data and PCs on a pixel-by-pixel basis throughout the image[84]. Unlike the true-color orthomosaic, the PCA composite highlights spectral variability instead of natural colors, making it a valuable tool for detecting hydrothermal alteration zones. The PCA technique is well established in remote sensing for detecting and mapping hydrothermal alteration[86]. The technique is widely applied in the volcanology community to delineate fumarolic deposits and geothermal features[30,49], classify hydrothermally altered materials[87], and identify structurally weakened areas influenced by hydrothermal processes[26]. These applications highlight PCA's effectiveness in analyzing regions affected by hydrothermal alteration. Using ArcGIS Pro (v3.3.0), we conducted PCA on the SfM-generated true-color orthomosaic, assigning three PCs to decorrelate the RGB bands. This analysis produced three raster layers (PC1, PC2, and PC3), each representing different levels of variance within the dataset (Fig. 8B). To further enhance the contrast between altered and unaltered areas, we generated a PCA composite raster, with the red band representing the highest spectral variance, followed by green and blue.

### Supervised and unsupervised classification of alteration zones

Following our PCA, we applied a supervised classification using the Maximum Likelihood algorithm in ArcGIS Pro to differentiate between altered and unaltered surfaces. Approximately 100 training areas were selected, with 50 placed within PCA-highlighted red-to-pink regions representing altered surfaces and 50 outside of the region representing unaltered surfaces. These training areas were manually drawn as polygons of varying size and shape, placed both inside and outside of the CFZ. Polygons were selected to capture the range of surface variability visible in the PCA composite, particularly the differences in color. The number of pixels per training site was not standardized, as the focus was on representing spectral diversity rather than a uniform size.

These training areas served as spectral reference inputs for the Maximum Likelihood classifier, allowing each pixel to be assigned to the most spectrally similar category[84,85]. Our justification for training area selection is based on the CFZ, a well-documented region of persistent fumarolic activity and hydrothermal alteration. Previous studies have mapped this region as an area of silicic and advanced argillic alteration using hyperspectral data[57], identified thermal anomalies associated with persistent fumarolic activity[56], and similar mapping methods as ours using drone-PCA methods, combined with gas flux measurements[30]. This distribution reflects a well-established hydrothermal zonation pattern, with the CFZ representing the

acidic, silica-rich region surrounded by an advanced argillic halo (OFZ)[88]. This cross-validation from independent datasets supports the CFZ as a reliable and validated spectral endmember for a supervised classification. Therefore, this region is well-suited for our training area site selection for our supervised classification of hydrothermal alteration. This method effectively differentiates hydrothermally altered rocks from unaltered surfaces while excluding vegetated areas, thus refining the study area for further sub-classification of altered surfaces.

To investigate variations within the altered zone itself, we applied an unsupervised classification, complementing the binary output of the supervised approach. A key challenge in mapping alteration zones is the heterogeneous distribution of alteration intensity, which makes it challenging to distinguish contacts between different alteration types based solely on color. This issue is evident in the true-color orthomosaic of La Fossa, where varying degrees of rock bleaching create a complex, non-uniform pattern rather than a clear transition between regions. To better capture these variations, we applied an unsupervised classification to the true-color (RGB) orthomosaic, confining the analysis to the previously classified altered region by masking out the unaltered regions and focusing only on altered material.

Here, we used the Iso Cluster Unsupervised Classification tool in ArcGIS to subdivide the alteration zone into four predefined classes, each representing a distinct intensity of alteration based on surface color characteristics (Fig. 8C). This tool groups pixels into classes by identifying natural clusters within the RGB values[84], employing two complementary algorithms: the Iterative Self-Organizing Data Analysis (ISODATA) algorithm, which iteratively refines clusters by adjusting their centers, and the Maximum Likelihood algorithm, which statistically assigns pixels based on probability distributions to enhance classification accuracy. Rather than applying an evenly spaced division of digital values (e.g., equal-interval binning), the classification process detects and separates naturally occurring clusters within the RGB color space. To maintain interpretability, we used a four-class system, consistent with common practice in unsupervised image classification, where large numbers of spectral clusters are often grouped into a smaller number of meaningful classes to reduce complexity[89]. As a result, the four classes reflect inherent spectral similarities associated with varying degrees of hydrothermal alteration, as expressed through surface color variations such as bleaching intensity.

Using this approach, we classified the alteration into four intensity levels: Very High, High, Moderate, and Low. These intensity classes were defined according to representative RGB color profiles, where Low alteration corresponds to darker tones with RGB values approaching (0, 0, 0), and Very High alteration represents highly bleached materials with RGB values near (255, 255, 255). We interpret these classes as representing increasing degrees of hydrothermal alteration, consistent with previous studies linking bleaching to mineralogical alteration developed in volcanic settings[21]. By systematically differentiating alteration intensity, the classification provides a detailed and accurate representation of surface variations, enhancing our ability to analyze hydrothermal alteration. To further quantify these variations, a transect-based approach was conducted to compare relative rock strength across classified alteration intensities. By systematically measuring surface rocks across altered and unaltered regions, this method provided additional insight into how hydrothermal alteration affects rock integrity.

### In situ rock hardness assessment (Schmidt hammer)

Hydrothermal alteration is well recognized for its strong influence on rock physical properties, including compressive strength. While studies have investigated these effects in controlled laboratory conditions, sample sizes typically remain limited (fewer than 20)[17]. This constraint is partly due to the research emphasis on inherited rock strength characteristics rather than their spatial distribution. Additionally, determining mechanical properties is time-consuming and costly[90], making transporting hundreds of rocks to the laboratory impracticable. As a result, broader geophysical patterns and regional variability of hydrothermal alteration remained less explored.

**Table 2 | Summary statistics of Schmidt hammer values across alteration grades**

| Data summary | Unaltered | Low | Medium | High | Very high |
|---|---|---|---|---|---|
| Mean | 60.1 | 42.8 | 36.2 | 34.8 | 31.1 |
| Median | 63 | 40 | 36.5 | 32.5 | 27 |
| 1st quartile | 52.9 | 31 | 23.3 | 22.5 | 19.5 |
| 3rd quartile | 71.1 | 55 | 47.8 | 45.5 | 43 |
| 95th percentile | 79.5 | 71.4 | 64.8 | 66.4 | 57.6 |
| 95% confidence interval | 62.00, 65.00 | 37.64, 42.75 | 30.00, 39.00 | 28.50, 36.00 | 23.75, 30.00 |

Reported values include the mean, median, first quartile (Q1), third quartile (Q3), 95th percentile, and bootstrapped 95% confidence intervals for the median Schmidt hammer "*R*" values. Classifications correspond to those shown in Figs. 4–6.

Our approach explores the capabilities of the Schmidt hammer (also known as the rebound or impact hammer), a device initially designed for testing the hardness of concrete[91] (Fig. 8D). We used a Proceq Silver Schmidt Type N hammer, calibrated prior to the field campaign using a manufacturer-supplied anvil specific to this model to ensure measurement accuracy. More recent research has expanded the use of the Schmidt hammer to assess rock integrity through surface rebound hardness[64,65]. The instrument's portability and ability to perform rapid in situ measurements make it a well-suited tool for direct field studies[66]. These characteristics enable a large number of measurements over a large region, enabling the analysis of the spatial variability in rock strength.

The fundamental operation of the Schmidt hammer involves an internal spring-loaded mass that, when pressed orthogonally against the surface, is released onto a plunger with a known energy. The resulting rebound height of the hammer yields a numerical value known as the *R* value, which serves as an indicator of surface hardness and, consequently, the material's compressive strength[64,65,67,91,92]. These *R* values can be converted to UCS values (e.g., MPa, N/mm²) using established empirical conversion curves[64,67]. However, developing specific conversion curves for each sampling scenario can be costly and time-consuming. While we could have applied empirical relationships from published UCS conversion curves, these are typically developed for unaltered lithologies and would not accurately represent the rocks at Vulcano. Alternatively, the direct comparison of rebound values provides an efficient means of estimating relative strength across different lithologies, allowing for a large number of measurements without requiring site-specific calibration. As a result, the direct use of *R* values is widely applied for relative field measurements of surface hardness[76].

Measurement techniques should be adapted to the specific research scope and field conditions to ensure meaningful results[76]. While standardized protocols in engineering geology, such as those established by the International Society of Rock Mechanics and Rock Engineering (ISRM) and the American Society for Testing and Materials, typically recommend taking a minimum number of measurements and reporting an average *R* value per location, this approach may not be suitable for all environments. In areas with limited exposed rock surfaces, a higher number of Schmidt hammer measurements may be necessary to obtain representative data, whereas in regions with abundant rock outcrops, fewer measurements may be required[76]. Sampling strategies should thus be tailored to the local geological conditions of the study area rather than strictly adhering to generalized approach protocols.

Previous works have also shown that repeated hammering may induce artificial weakening, resulting in lower Schmidt hammer measurements[93] or causing compaction to the skeletal structure of the sample, leading to higher values compared to the first measurement[64]. Therefore, recording only the first Schmidt hammer impact as the reported relative strength value for that particular sample is recommended[76]. Given the considerations mentioned, we apply the Schmidt hammer using the single-shot technique, taken across hundreds of rocks implemented through several transects that traverse unaltered and altered surface environments (Fig. 8E).

The transects were strategically oriented to target flanks affected by past mass-wasting events across multiple surface types, facilitating comparative analysis (Fig. 8F). Additionally, three smaller east-to-west transects were conducted: one measuring rocks paralleling the northern flank edge and the other two intersecting various alteration surfaces within the CFZ. Each Schmidt rebound value was recorded alongside its corresponding GPS coordinates and later incorporated into our thematic map[68]. We took approximately 1100 measurements, but reported 1038. The remaining measurements registered a "NaN" (Not a Number) read-out due to either the hammer not striking the rock orthogonally, the rock fracturing, or the sample surfaces being too soft for the device to register a value; therefore, they were excluded from our analysis.

To test whether surface bleaching intensity corresponds to rock strength, Schmidt hammer measurements were compared across the four alteration classes defined by our unsupervised classification. This approach allowed us to explore the spatial relationship between surface alteration intensity and in situ rock strength. To facilitate interpretation and comparison, we grouped Schmidt hammer rebound values into five relative strength categories based on *R* values: Very Weak ($\leq20$), Weak ($\leq35$), Moderate ($\leq50$), Strong ($\leq65$), and Very Strong ($\geq65$). These categories are not UCS-calibrated but reflect rebound value ranges commonly used for relative surface strength assessments[76]. Given the range of alterations, intensity, and absence of site-specific UCS testing, this classification provides a consistent framework for internal comparison for spatial trends. To assess statistical significance in our Schmidt hammer rebound values across alteration classes, we applied a non-parametric Kruskal–Wallis test, followed by Bonferroni-corrected post hoc pairwise comparison. Additionally, we calculated bootstrapped 95% CIs for median rebound values for each class using 1000 resampling iterations. This approach provided robust estimates of group-level trends under non-normal distributions. Full statistical outputs are provided in Table 2 and Supplementary Table 1.

## Limitations

One limitation of this study is that both Schmidt hammer measurements and drone-based photogrammetry provide information restricted to the volcano's surface, offering no direct insight into subsurface conditions. As a result, our interpretations of hydrothermal alteration are confined to surface exposures, and we, therefore, infer that similar alteration patterns extend at depth. While this limitation is acknowledged, it also reflects a practical reality at many active volcanic systems where borehole sampling is not feasible. Our surface-based approach is therefore intentionally designed to utilize accessible outcrops to develop a rapid, non-invasive approach for assessing volcanic instability. Future studies that integrate remote sensing, rock property measurements, and subsurface geophysical data, such as electrical resistivity tomography[94], would provide a more holistic assessment approach, but we stress that geophysical measurements are not possible at particularly hazardous volcanoes. This limitation reinforces the utility of our surface-based method, which is designed as a rapid, field-applicable tool for assessing hydrothermal weakening and slope instability where subsurface data cannot be readily acquired. Additional limitations are related to the spatial accuracy and environmental constraints of the data. The 2019 UAV survey lacked RTK (Real-Time Kinematics) or PPK (Post-Processed Kinematics) GNSS corrections, resulting in relative camera positioning errors that may exceed 1 m[95] and horizontal positioning uncertainties of up to 10 m[80]. Despite these minor spatial inaccuracies within our spatial precision, it did not compromise our ability to effectively map the field area, flanks, and associated regions of alteration. Environmental factors also influenced our data quality. Vegetation cover and surface erosion, particularly in the lower-flank regions, obscured key surface features, complicating classification efforts and likely contributing to an underrepresentation of

altered surfaces in those areas. For instance, in the 1988 landslide region, alteration and vegetation were mapped as scattered, irregular patches along the landslide footprint. As a result, the extent of alteration in this region may be larger than what is represented in our dataset.

Rock strength assessments were similarly affected by terrain. On steep slopes, particularly the northern flank, erosion and downslope transport of altered rocks introduced significant variability in our Schmidt hammer rebound values. Mixing in situ and displaced material complicates our interpretations of in-place relative rock strength. While Schmidt rebound values correlate with UCS[64] and have been used in previous studies[76,93], our method offers only relative assurance of rock strength. The absence of laboratory-based UCS testing limits the quantitative calibration of our strength estimations for use in slope stability modeling[96]. Nonetheless, the extensive dataset enabled effective mapping of strength trends and identification of mechanically weak zones for future sample targeting. Although we did not perform a comprehensive lithological analysis at each measurement site, an effort that would reduce, rather than maximize, the number of measurements collected, we mitigated this limitation by relying on established geological maps, which indicate that the La Fossa cone edifice is predominantly composed of pyroclastic deposits[43,97–100]. In an attempt to limit lithological variability, Schmidt hammer measurements were conducted, where possible, on exposed pyroclastic rock surfaces. This approach helped to ensure that the observed strength variations primarily reflect differences in alteration intensity, while acknowledging that hydrothermal overprinting can often obscure the original rock type.

Lastly, in our classification approach, since the PCA is an unsupervised dimensionality reduction technique that transforms data into single PCs, the subsequent analysis relies on visual interpretations. Consistent with past works[26,30,49], PC1 correlated with surface brightness, PC2 highlighted the CFZ, and PC3 emphasized sulfur deposits, suggesting that the classifications are not artifacts and can be replicated. While the single PC feature extraction approach worked well in localized regions, single PCs proved less reliable when applied across a more extensive and diverse field area and misclassified unaltered surfaces. Therefore, the composite raster approach proved more effective in distinguishing the contacts between altered and unaltered surfaces. Consequently, this method was chosen for our subsequent classification process.

Conceptually, we interpret the exposure of highly altered rocks at flank failure sites as being associated with Low Schmidt rebound values, with the latter dataset only available where field access was safe. In this regard, soft rock can trigger landslides, making the accurate identification of these rocks crucial. However, some of these exposures may result from the removal of overburden during failure, raising the possibility that alteration becomes visible only post-collapse. For this reason, we have closely examined areas and found alteration where erosional gullies are most pronounced (e.g., along the southern crater wall), in the regions surrounding the collapse amphitheaters in the north, or on the steep cliffs of the south-facing flank, all of which could potentially lead to future flank instabilities, as described above.

## Data availability

The data for this study and processing steps will be published on the Zenodo data repository (https://doi.org/10.5281/zenodo.17348856, De Jarnatt et al.[68]).

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

## Acknowledgements

All authors thank GFZ-Potsdam for financial support in the framework of the focus site "Etna and Southern Italy". The authors thank INGV Palermo for resources and logistical support in the field and the Geomorphology section (4.6) at GFZ-Potsdam for using their Schmidt hammer(s). The authors also thank Julia Nikutta for her assistance in the field. B.F.D., T.R.W., and M.J.H. are supported by a European Research Council Synergy Grant (ERC-ROTTnROCK-101118491). M.J.H. is also supported by ANR grant MYGALE ("Modelling the physical and chemical Gradients of hydrothermal alteration for warning systems of flank collapse at explosive volcanoes"; ANR-21-CE49-0010) and acknowledges support from the Institut Universitaire de France (IUF).

## Author contributions

B.F.D. led this paper's work, field campaign, analysis, and writing of this paper. T.R.W. supervised the project and contributed to the fieldwork and analysis. M.J.H. was the co-supervisor and contributed to the analysis and co-supervision. D.M. contributed to the fieldwork, data, and analysis; A.F.P. contributed to the fieldwork. All co-authors contributed to the writing of the manuscript.

## Funding

## Competing interests

The authors declare no competing interests.
