## [Transparent Peer Review file · Communications Earth & Environment]

Hydrothermal weakening and slope instability at Vulcano (Italy) analyzed using drones and in-situ strength measurements

Corresponding Author: Mr Benjamin De Jarnatt

Version 0:

Decision Letter:

Dear Mr De Jarnatt,

Your manuscript titled "Hydrothermal weakening and slope instability at Vulcano (Italy) analyzed using drones and in-situ strength measurements" has now been seen by 3 reviewers, whose comments are appended below. You will see that they find your work of some potential interest. However, they have raised quite substantial concerns that must be addressed. In light of these comments, we cannot accept the manuscript for publication, but would be interested in considering a revised version that fully addresses these serious concerns.

We hope you will find the reviewers' comments useful as you decide how to proceed. Should additional work allow you to

- address these criticisms (that is, either to incorporate the suggestions or provide a compelling argument why the point made by the reviewer is not valid or relevant to the editorial threshold as outlined below)

AND

- meet our editorial thresholds as outlined below,

then we would be happy to look at a revised manuscript.

In the following, we list our requirements for publication.

***Present novel and fully supported insight into the assessment of volcanic slope instability using drones and in-situ measurements

***Clarify the research gap, outline your method and all parameters used, justify the choice of the supervised classification algorithm, resolve statistical interpretation, and quantitatively compare your approach, referencing similar applications.

***Expand the discussion of your findings, addressing generality and uniqueness, and demonstrate that all your data and analysis fully support your claims.

If you choose to take up this option, please either highlight all changes in the manuscript text file, or provide a list of the changes to the manuscript with your responses to the reviewers.

When resubmitting, please provide a point-by-point response to the reviewers' comments. Please submit your responses as a separate file, distinct from your cover letter where you can add responses to the Editors' comments that you do not want to be made available to the reviewers. Word files are preferred. We recommend that any figures, tables or graphs that are included in the response to reviewers are also included in the main article or Supplementary Information.

If the revision process takes significantly longer than three months, we will be happy to reconsider your paper at a later date, as long as nothing similar has been accepted for publication at Communications Earth & Environment or published elsewhere in the meantime.

Please use the following link to submit your revised manuscript, point-by-point response to the reviewers' comments with a list of your changes to the manuscript text (which should be in a separate document to any cover letter), a tracked-changes version of the manuscript (as a PDF file) and any completed checklist:

Link Redacted

Please do not hesitate to contact us if you have any questions or would like to discuss the required revisions further. Thank you for the opportunity to review your work.

Best regards,

Teng Wang, PhD
Editorial Board Member
Communications Earth & Environment
0000-0003-3729-0139

Martina Grecequet, PhD
Senior Editor
Communications Earth & Environment

EDITORIAL POLICIES AND FORMAT

If you decide to resubmit your paper, please ensure that your manuscript complies with our editorial policies and complete and upload the checklist below as a Related Manuscript file type with the revised article:

- Behavioural and social science
- Ecological, evolutionary & environmental sciences
- Life sciences

For your information, you can find some guidance regarding format requirements summarized on the following checklist: (<https://www.nature.com/documents/commsj-phys-style-formatting-checklist-article.pdf>) and formatting guide (<https://www.nature.com/documents/commsj-phys-style-formatting-guide-accept.pdf>).

REVIEWER COMMENTS:

Reviewer #1 (Remarks to the Author):

Manuscript number: COMMSJ-25-1704

Hydrothermal weakening and slope instability at Vulcano (Italy) analyzed using drones and in-situ strength measurements

In order to determine the correlations with alteration types in the Vulcano Islands and assess the degree of stability, this research uses an integrated strategy using in situ Schmidt hammer and UAV-based photogrammetry.

These findings support earlier research on possible failure. For hydrothermalized massifs, a new classification is suggested.

I've made an effort to evaluate this manuscript and value its scientific content. The subsurface conditions and limitations have been considered, implying future studies.

It gives me great pleasure to suggest that the manuscript be published.

Reviewer #2 (Remarks to the Author):

Dear authors

I have evaluated the manuscript title: "Hydrothermal weakening and slope instability at Vulcano (Italy) analyzed using drones and in-situ strength measurements" prepared for publication in *Communications Earth & Environment* (COMMSERV-25-1704). The manuscript presents a contribution to the assessment of volcanic slope instability using drone-based photogrammetry and in-situ Schmidt hammer measurements. While the study is methodologically solid and potentially impactful, several key aspects require clarification and deeper analysis. In particular, the discussion lacks sufficient critical depth and comparative evaluation with existing literature. The authors should better justify methodological choices (e.g., classification parameters) and quantitatively compare their approach, ideally using benchmark metrics, or referencing similar applications at other volcanic sites. Furthermore, the conclusions are too general and need to be better grounded in the data, with clearer articulation of limitations and applicability conditions.

Kind regards
Antonio Pola

The following are my main observations and recommendations:

- 1. Abstract:** The abstract is generally well structured, but its clarity could be improved by more explicitly stating the research gap and the novelty of the proposed approach. While the methodology is briefly mentioned, it omits the crucial use of PCA-based remote sensing, which underpins the classification of hydrothermal alteration. The reported ~50% reduction in rock strength is compelling but would be more informative if contextualized with specific Schmidt rebound values or strength ranges. Finally, the abstract would benefit from a stronger concluding sentence that highlights the broader implications of the method for volcanic hazard monitoring and early-warning systems.
- 2. Introduction:** The introduction is informative but lacks focus and a clear statement of the research gap. It reviews many contributing factors to volcanic flank collapse yet delays in identifying what is specifically missing in the literature. The introduction cites numerous case studies, but their integration feels list-like. While these examples illustrate the global relevance of hydrothermal weakening, they are not clearly tied to the central research question. Additionally, while several relevant examples of volcanoes affected by hydrothermal weakening are cited, the transition to the case study at La Fossa is abrupt and would benefit from a stronger rationale for its selection.
- 3. Study area:** The section is detailed and provides solid geologic, tectonic, and eruptive context. However, it reads more like an extended descriptive summary than a targeted justification for the research. While the tectonic and eruptive background is relevant, the discussion could be condensed to emphasize the aspects most pertinent to instability and hydrothermal alterations specifically, geomorphological changes, landslide history, and degassing patterns. The 1988 and 2010 mass-wasting events are important, but their connection to hydrothermal weakening should be more explicitly framed to reinforce the rationale for site selection.
- 4. Results:** The Results section presents a rich and well-organized dataset. However, the section tends to remain descriptive and would benefit from more quantitative depth. While the observed ~50% reduction in Schmidt hammer values across alteration classes is compelling, statistical analyses (e.g., confidence intervals, significance tests, or effect size metrics) are missing and would strengthen the interpretation. Additionally, the results rely heavily on visual interpretation of figures, with limited engagement with the supplemental tables, particularly Table 2, which contains key summary statistics. Explicitly referencing these tables and integrating their content into the narrative would improve data transparency and traceability. Finally, the spatial patterns of weakening are discussed in qualitative terms but could be enhanced by linking them more systematically to mapped alteration zones and collapse-prone sectors, reinforcing the hazard implications of the study.
- 5. Discussions:** The discussion lacks the critical comparative depth and analytical rigor expected for a study addressing volcanic slope instability. While it reinforces the link between hydrothermal alteration and mechanical weakening, it does so primarily through site-specific correlations. The ~50% reduction in Schmidt hammer values is not compared with similar findings at other volcanoes, limiting the reader's ability to assess the generality or uniqueness of the observations. Moreover, the inference that low rebound values reflect pre-collapse weakening is not critically examined considering alternative explanations, such as post-failure exposure or clast transport. The discussion would also benefit from the inclusion of additional physical properties (such as porosity) which are often reported alongside Schmidt hammer values and would provide a more complete mechanical characterization of altered rocks. Lastly, while the workflow is presented as broadly applicable, the discussion fails to clearly define its operational limits, decision thresholds, or how it could be integrated into real-time hazard monitoring. A more systematic and comparative approach would significantly enhance the interpretive power and relevance of the study.
- 6. Conclusions:** The conclusions need to be more direct and grounded in the evidence presented. Key statements (such as the method being scalable or transferable) are too broad and not supported by comparative benchmarks or clearly defined conditions. The section also overlooks important limitations discussed earlier, such as the lack of subsurface data. For a study aiming at high-impact relevance, the conclusions must convey precise, evidence-based takeaways and clearly define the scope and limits of applicability.
- 7. Methods:** The Methods section is detailed and technically solid, but its placement after the Results disrupts the logical flow and hinders reproducibility. Moreover, while the PCA and classification workflows are described in depth, the rationale behind parameter choices (e.g., number of classes, selection of training zones) could be better justified with references or validation metrics. The use of both supervised and unsupervised classification is appropriate, but the transition between them is abrupt and would benefit from clearer explanation of their complementary roles. The Schmidt hammer protocol is

well detailed, including limitations and the decision to use single-shot measurements. Nonetheless, the absence of cross-validation with UCS laboratory data, although acknowledged, remains a limitation for quantitative modeling. The authors should: clarify the logic and validation behind classification choices; and discuss whether any calibration or comparison with UCS data is possible to support the strength estimates.

8. Figures: Please, take care of the following aspects. 1) Labels in composite figure should not differ in order, size, style, and position. 2) include the direction in which the photo was taken, if possible, include a graphic scale. 3) The range and scale of the composite figures should be the same to avoid any confusion; a different scale is presented in figure 2; consequently, visual comparison of the graphs could be difficult. 4) it must be ensured that when composite figures are scaled, the labels are not deformed on any of their axes.

Reviewer #3 (Remarks to the Author):

Review- Report to the authors

Title: Hydrothermal weakening and slope instability at Vulcano (Italy) analyzed using drones and in-situ strength measurements

Authors: Benjamin F. De Jarnatt; Thomas R. Walter; Michael J. Heap; Daniel Mueller; Antonino Fabio Pisciotta

This study addresses a very interesting topic focused on monitoring hydrothermal alteration zones to identify potential collapses. The study area is La Fossa volcano, on Vulcano Island (Aeolian Archipelago, Italy). The work is based on the generation of orthomosaics using unmanned aerial vehicles and the Schmidt hammer method to evaluate rock strength in relation to hydrothermal alteration, with the goal of producing thematic maps. The authors performed an interpolation of the results, as most of the hydrothermal alteration zones are largely inaccessible. They found that the results indicate a 50% reduction in rock strength, which they associate with areas of degassing and hydrothermal activity, and which also coincide with zones affected by mass-wasting events.

I believe this is a very novel and interesting research; however, I think there are some methodological issues that the authors could consider to improve their work.

It would be important to highlight somewhere in the text that hydrothermal alteration can also lead to a strengthening of the rocks, for example, through silicification.

Please move the Methods section before the Results. Many questions related to the methodology arise while reading the results.

Previous works related to hydrothermal alteration zones in the study area could serve to validate some of the findings presented here, assuming the data is available.

There are some uncertainties concerning the interpretation and use of the principal components. Vegetation seems to be identified in PC3, which is assigned to the blue channel, so vegetation should be enhanced in blue? The composite image uses $R = PC1$, $G = PC2$, and $B = PC3$, yet the text mentions that PC1 is excluded from further analysis, raising questions about why PC1 is included in the color composite image as the red channel where alteration is enhanced. This apparent contradiction may stem from phrasing issues.

Please specify which supervised classification algorithm (e.g., Parallelepiped Classification, Maximum Likelihood Classification, Mahalanobis Distance Classification, etc.) was employed and explain the rationale for its selection. Details about the training sites' locations, the results used for their delineation (color composite of the three components?) and the criteria for their selection, including the number of pixels involved, would clarify the methodology.

The four predefined classes appear to represent a range from low to high alteration. It would be important to know whether there is any field verification of the samples to support this interpretation. Typically, greater bleaching suggests a higher degree of alteration and consequently weaker rocks. Please, clarify if whether the lithology undergoing alteration remains consistent throughout the study area.

I recommend taking into consideration the following article, which includes many insights that could be useful for your work: Pereira, M. L., Zanon, V., Fernandes, I., Pappalardo, L., & Viveiros, F. (2024). Hydrothermal alteration and physical and mechanical properties of rocks in a volcanic environment: A review. *Earth-Science Reviews*, 104754.

I recommend reviewing the writing of the text to improve its consistency. Many sentences are very long.

The corrections are detailed in the attached PDF.

I hope my comments help to improve your work.

Communications Earth & Environment is committed to improving transparency in authorship. As part of our efforts in this direction, we are now requesting that all authors identified as 'corresponding author' create and link their Open Researcher and Contributor Identifier (ORCID) with their account on the Manuscript Tracking System prior to acceptance. ORCID helps the scientific community achieve unambiguous attribution of all scholarly contributions. You can create and link your ORCID

from the home page of the Manuscript Tracking System by clicking on 'Modify my Springer Nature account' and following the instructions in the link below. Please also inform all co-authors that they can add their ORCID to their accounts and that they must do so prior to acceptance.

Version 1:

Decision Letter:

Dear Mr De Jarnatt,

Your manuscript titled "Hydrothermal weakening and slope instability at Vulcano (Italy) analyzed using drones and in-situ strength measurements" has now been seen by our reviewers, whose comments appear below. In light of their advice we are delighted to say that we are happy, in principle, to publish a suitably revised version in Communications Earth & Environment.

We therefore invite you to revise your paper one last time to address the remaining concerns of our reviewers. At the same time we ask that you edit your manuscript to comply with our format requirements and to maximise the accessibility and therefore the impact of your work.

EDITORIAL REQUESTS:

*****Please take care to match our formatting and policy requirements. We will check revised manuscript and return manuscripts that do not comply. Such requests will lead to delays. *****

SUBMISSION INFORMATION:

OPEN ACCESS:

Communications Earth & Environment is a fully open access journal. Articles are made freely accessible on publication. For further information about article processing charges, open access funding, and advice and support from Nature Portfolio, please visit <https://www.nature.com/commsenv/open-access>

Link Redacted

Best regards,

Teng Wang, PhD

Editorial Board Member
Communications Earth & Environment

Martina Grecequet, PhD
Senior Editor,
Communications Earth & Environment

REVIEWERS' COMMENTS:

Reviewer #1 (Remarks to the Author):

Manuscript number: COMMSENV-25-1704

Hydrothermal weakening and slope instability at Vulcano (Italy) analyzed using drones and in-situ strength measurements

In order to determine the correlations with alteration types in the Vulcano Islands and assess the degree of stability, this research uses an integrated strategy using in situ Schmidt hammer and UAV-based photogrammetry. These findings support earlier research on possible failure. For hydrothermalized massifs, a new classification is suggested. The subsurface conditions and limitations have been considered, implying future studies.

From a methodological perspective, this is an impressive correction that involves several clarifications.

The introduction (lines 100–107) and the discussion with the help of the bibliography should, however, make clear the features of the destabilization of volcanic islands in subduction zones and associated trenches, including hydrothermal processes, particularly Aeolian islands.

Despite being mentioned in the introduction (lines 89–95), the abstract doesn't seem to address the connections between hydrothermal alteration and the destabilizing of a volcanic island.

A thorough explanation of these should be given using these results and the discussion that goes along with them.

Moreover, sulfur and vegetations are clearly detected in relation to hydrothermal changes (Fig. 2).

But don't be afraid to use the many bibliographic references that discuss the relationships between spectral classification zonation and hydrothermal alteration halos (Fig. 2, line 428: "other altered region in red-to-pink colors").

Indeed, the conduit exhibits an inner leaching and amorphous silica zone, and is surrounded by less acidic alteration zones related to the formation of alunite and kaolinite (see also Müller et al., 2021; Robb, 2005).

Another kind of destabilization interactions (Fig. 7) is implied by this kind of structural zoning, such as a collapse depletion by conduit collapse. The theoretical structural relationships based on hydrothermal alteration gradients (lines 813-814: faulting not shown) would be preferable. In actuality, the circular fault hypothesis must be taken into consideration when analyzing the features of volcanic collapses.

Finally, the rewriting contains grammatical errors, particularly in lines 378, 380, and 385. "Gray" is an American English word, but "grey" is used in British English.

In both upper and lower case letters, the word "Unaltered" appears frequently (lines 518, 699, 700, 708, etc.). Mathematical notation typically does not observe spaces (lines 524, 581, 701, 702, etc.).

For this reason, I recommend a minor revision.

Best regards
Kbernard

Reviewer #2 (Remarks to the Author):

I am satisfied with the authors revisions and responses to my comments. The revised manuscript has substantially improved in clarity, methodological justification, and analytical depth. The authors have explicitly addressed the concerns I raised, including clarification of the research gap, better integration of comparative literature, quantitative strengthening of the results with appropriate statistical analyses, and a more critical and balanced discussion that acknowledges limitations and operational boundaries. The conclusions are now more precise and well-grounded in the data, and the methodological choices are clearly justified.

Reviewer #3 (Remarks to the Author):

Review- Report to the authors

Title: Hydrothermal weakening and slope instability at Vulcano (Italy) analyzed using drones and in-situ strength measurements

Authors: Benjamin F. De Jarnatt; Thomas R. Walter; Michael J. Heap; Daniel Mueller; Antonino Fabio Pisciotta

After carefully analyzing the rebuttal letter, it is clear that the authors provided a point-by-point response to all the reviewers' comments. I find their replies to be highly satisfactory, as they significantly contribute to making the manuscript clearer and more concise. I sincerely appreciate the effort devoted to addressing each request in detail. This revised version fully incorporates all my previous comments. In my opinion, it represents an excellent piece of research, proposing a sound methodology to study volcanic slope instability with direct application to hazard monitoring and early warning systems. The selected case study, La Fossa cone (Vulcano, Italy), is particularly well suited for this type of analysis, combining drone

photogrammetry with in situ Schmidt hammer measurements, and generating an alteration map through Principal Component Analysis. The results presented are highly relevant, showing an approximate 50% reduction in relative rock strength in areas affected by degassing and hydrothermal activity. These findings are consistent with evidence from past mass-wasting events.

** Visit Nature Portfolio's author and referees' website at www.nature.com/authors for information about policies, services and author benefits**

**Hydrothermal weakening and slope instability at Vulcano**
**(Italy) analyzed using drones and in-situ strength**
**measurements**

Benjamin F. De Jarnatt^{1,2}; Thomas R. Walter^{1,2}; Michael J. Heap^{3,4}; Daniel Mueller¹;
Antonino Fabio Pisciotta⁵

¹ GFZ Helmholtz Centre for Geosciences, Telegrafenberg, 14473 Potsdam, Germany

² Institute of Geosciences, University of Potsdam, Karl-Liebknecht-Str. 24-25, 14476
Potsdam, Germany

³ Université de Strasbourg, CNRS, Institut Terre et Environnement de Strasbourg, UMR
7063, 5 rue Descartes, Strasbourg 67084, France

⁴ Institut Universitaire de France (IUF), 1 rue Descartes, Paris 75231, France

⁵ Istituto Nazionale di Geofisica e Vulcanologia (INGV), Via Ugo la Malfa 153, 90146
Palermo, Italy

**Abstract**

Instability at volcanic edifices poses significant hazards, yet the processes driving rock
weakening remain poorly understood, partly due to the inaccessibility of steep, eroding
flanks. Hydrothermal alteration is a key factor in weakening volcanic rock and
contributing to flank instability. La Fossa cone (Vulcano, Italy) has not only alteration
observed at the crater rim and outer flanks but also a history of mass wasting. Here, we
combined drone photogrammetry with in situ Schmidt hammer testing to derive an
empirical alteration-to-strength relationship for the crater rim and applied this knowledge
to alteration sites on inaccessible outer flanks. We classified alteration types and
transposed over 1,000 relative rock strength measurements onto our thematic map.
Results indicate a ~50% reduction in relative rock strength that correlates with areas of
degassing and hydrothermal activity, coinciding with past mass-wasting events. This
integrated approach offers a practical workflow for assessing volcanic slope instability
and hazard assessment.

Introduction

Volcanic flank instability and subsequent flank collapse pose significant hazards at both
active and dormant volcanoes worldwide. Collapse events can have far-reaching
consequences and potentially devastating effects on local communities and
infrastructure; however, such collapses are relatively rare, with observations limited to
only a few documented cases¹. ~~One important factor is duration:~~ collapses of a volcano
flank may occur suddenly and catastrophically, generating far-traveling deposits and
even tsunami hazards ~~if located near bodies of water~~², or gradually, as a slow-moving
process³ that can be closely monitored over an extended period⁴. Another key factor is
the scale of flank instability, ranging from small mass-wasting events (a few m³) to giant
landslides (>1000 km³), with the scale inversely proportional to the occurrence rate⁵.
The process of a volcanic flank becoming unstable can manifest from a range of
processes, including flank over-steepening⁶⁻⁸, magma emplacement⁹, stress changes¹⁰,
weakened basement rock^{11,12}, shifts in pore pressure^{9,13-15}, heightened seismic
activity^{6,16}, basement spreading¹⁷, and hydrothermal alteration¹⁸⁻²⁰. While a single factor
may trigger instability, it often results from a combination of these processes acting
together^{6,21}.

Hydrothermal alteration is gaining increasing attention for its role in weakening volcanic
edifices, offering insight into both the timing and location of collapses while also
contributing to a persistent weakening that is difficult to detect and monitor²². There is
evidence that collapsing volcanic flanks contain an abundance of hydrothermally altered
material. Prominent examples have been documented at La Soufrière de Guadeloupe
(France), Las Casitas (Nicaragua), Mount Hood and Mount Rainier (USA), Santiaguito
(Guatemala), Merapi (Indonesia), Volcán de Colima (Mexico), and elsewhere^{19,23-25}. For
example, Mount Rainier's edifice contains hydrothermally altered material that has
contributed to past major collapse events, notably the Osceola Mudflow, which
generated a ~3.8 km² lahar that traveled over ~113 km more than 5,000 years ago^{24,26}.
In contrast, Merapi exhibits smaller-scale, more frequent collapses, where zones of
hydrothermal alteration ~~have been shown to~~ reduce dome stability and facilitate
gravitational dome collapse and pyroclastic flows^{23,27}. Deposits from both Mt. Rainier

and Merapi showed evidence of acid-sulfate alteration, highlighting the pervasive role of
hydrothermal systems in driving instability. These examples underscore the broad
spectrum of hazards associated with hydrothermal alteration, from catastrophic sector
collapses to recurring, smaller-scale instabilities, emphasizing the need to identify
altered regions for effective hazard assessment and mitigation.

The remaining issue is that the link between hydrothermal alteration and flank instability
is often established after an event, based on altered material found in landslides or
debris avalanche deposits. Consequently, there is a pressing need to develop methods
to identify areas susceptible to alteration-driven flank failure before a collapse occurs.
With its active hydrothermal system and frequent small-scale landslides, the La Fossa
cone serves as an ideal natural laboratory for developing and testing these methods,
ultimately improving forecasts for larger, more hazardous volcanic collapses. Although
such landslides are orders of magnitude smaller than sector collapses, their frequency
and accessibility offer an opportunity to develop a workflow that can later be scaled to
assess volcanic flanks of greater size and destructive potential.

Hydrothermal alteration changes the composition, mineralogy, and color of rocks, which
is why it can, in part, be recognized by the naked eye²⁸. The effects of hydrothermal
alteration in volcanic settings are widely acknowledged for their impact on the
physicochemical and petrophysical properties of the associated rocks^{14,23,29-31}.
Hydrothermal alteration in volcanic environments typically involves the circulation of
aggressive, acidic, and hot (>200°C) hydrothermal fluids, facilitating fluid-rock
interactions that lead to mineral dissolution, replacement, and precipitation,
consequently altering the chemical and physical composition of the rock from its original
state^{32,33}, creating "rotten rocks". Previous studies have demonstrated that hydrothermal
alteration can significantly affect the porosity, permeability, and strength of volcanic
rocks, thereby compromising the gravitational stability of the volcanic structure^{14,23,29,31}.
Specifically, the relationship between hydrothermal alteration and mechanical
weakening has been well established, with strength closely linked to increased porosity
and alteration intensity. Heap et al.¹⁹ and Darmawan et al.²³ observed a decrease in

uniaxial compressive strength from fresh, unaltered rocks to altered rocks as a function
of alteration intensity. Heap et al.¹⁹ observed a decrease from 60–140 MPa to less than
10 MPa as alteration increases, quantified by the proportion of secondary minerals,
increases from ~16–18% to 75%. Similarly, Darmawan et al.²³ reported that fresh dome
rock samples averaged 132 MPa in strength, whereas the most altered samples
averaged 11 MPa, emphasizing the influence of both porosity and the degree of
alteration (quantified via oxygen isotope ratios) on rock strength. **Consequently, these**
**results underline the importance of monitoring volcanic edifices undergoing**
**hydrothermal alteration before their potential collapse; however, poor and hazardous**
**accessibility to at-risk regions poses a continuous challenge in volcanic hazard**
**assessment.**

Typically, regions undergoing **hydrothermal alteration are found near craters** and along
the flanks of volcanic edifices. More specifically, alterations and hydrothermal deposits
are found at locations that are hazardous to reach or even inaccessible. As a result,
assessing these regions, which often contain steep and unstable slopes, degassing
systems, and challenging terrain ~~while also~~ evaluating the mechanical behavior of rocks
and identifying potential areas of failure, remains a significant challenge. Previous
works, therefore, applied remote sensing technologies, including multispectral and
hyperspectral imaging³⁴, optical and infrared drone remote sensing²⁸, and airborne
electromagnetic and magnetic measurements³⁵ to map areas containing hydrothermal
deposits. While these methods remain valuable for monitoring volcanoes and identifying
hydrothermal alteration, they rarely provide insight into the mechanical behavior of rocks
on hydrothermally altered flanks. **Herein, we used high-resolution drone data and**
**collected a large quantity of field strength measurements to further explore the**
**possibility of accessing the inaccessible. We turn to the La Fossa cone of Vulcano**
**Island (Italy), the southernmost exposure of the Aeolian archipelago (Fig. 1).**

This study integrates thousands of close-range drone images to create a high-resolution
map, enabling a pixel-wise analysis of color variations linked to hydrothermal alteration
and facilitating surface classification. Additionally, over 1,000 in situ relative strength
measurements were collected using a Schmidt hammer, providing a second dataset for

assessing rock mechanical properties. Combining these datasets ~~establishes an~~
empirical relationship between increasing alteration and decreasing rock strength at the
La Fossa cone, allowing us to understand instability hazards at the outer flanks,
particularly in the northern and northeastern regions. This approach overcomes
previous limitations in traditional rock mechanical assessments, which often rely on
laboratory experiments, and provides further insights into how hydrothermal alteration
influences volcanic flank instability.

*Study Area*

The La Fossa volcano, located on Vulcano Island within the Aeolian Archipelago,
represents the southernmost volcanic structure of this volcanic arc situated in the
Tyrrhenian Sea, off the northern coast of Sicily, Italy (Fig. 1B). The archipelago lies
above a subduction zone related to the convergence of the African and European
plates, characterized by the north-westward subduction of the Ionian lithospheric slab
beneath the Tyrrhenian lithosphere. A complex NNW–SSE fault system significantly
influences the structural and volcanic development of the region and plays a significant
role in the recent tectonic activity and deformation processes observed specifically at
Vulcano^{36,37}.

The eruptive history of Vulcano is subdivided into eight eruptive epochs dating back to
approximately 127 ka³⁸⁻⁴⁰, with prolonged quiescent periods in between. Volcano-
tectonic collapses contributed to the morphological evolution visible today^{38,39,41}. Since
1.1 ka, volcanic activity has been active at the La Fossa cone⁴². The recent eruptive
history of the La Fossa volcano includes several significant volcanic episodes, ranging
from effusive and explosive Vulcanian-type eruptions to phreatic events. Historical
eruptions have varied from moderate to high intensity, characterized by explosive,
violent Strombolian to Vulcanian type eruptions and accompanied by lava flow
emissions, widespread tephra deposits, and episodes of phreatic explosions⁴². The
most recent eruption occurred between 1888 and 1890, becoming the archetype of
"Vulcanian" eruptions, characterized by dense lapilli-tuffs and volcanic bombs⁴³. Since
then, La Fossa has undergone frequent periods of anomalous fumarolic degassing,

notably in 1979–1981, 1985–1986, 1988–1991, 1996, 2004–2005, 2009, and most
recently starting in September 2021^{44,45}.

[revised manuscript text omitted]

Further geomorphological changes associated with landslides can be identified at the
 northern flank. The Forgia Vecchia region (Fig. 1A, C) exhibits the most evident signs
 of active deformation, as indicated by geodetic surveys⁴⁹ and intense surface

alteration⁵³. This region is continuously **monitored** due to its ongoing morphological
slope changes⁴⁹, sub-parallel tension cracks near the slope edge⁵⁴, and proximity to the
**village of Vulcano** and the nearby harbor. Additionally, in 2010, another collapse
occurred along the northeastern external rim of La Fossa cone, resulting in a small
debris flow not far from the other landslide areas (Fig. 1A). This event occurred during
**calm conditions**, exposing hydrothermally altered pyroclastics at the surface⁵².
Therefore, a more systematic analysis of flank stability and hydrothermal alteration
occurrence is needed. Here, we combine high-resolution drone imagery with in situ
strength testing (see Methods) in accessible regions to infer relative rock strength
across the otherwise inaccessible steep flanks.

**Results**

*Orthomosaic Analysis and Hydrothermal Alteration Mapping*

True-color orthomosaic maps generated from drone (unmanned aerial vehicle - UAV)
imagery cover an area of approximately 3.74 km² with a resolution of 7.6 cm x 7.6 cm,
which provides us with a comprehensive view of the La Fossa cone (Fig. 2A). The **maps**
include less accessible and less explored regions, such as the 1988 landslide on the
northeasternmost edge of the island and the steepest areas along the north flank and
Forgia Vecchia (Fig. 1A, C, E). The true-color orthomosaic (Fig. 2A) **reveals regions**
**indicative of hydrothermal alteration**, characterized by a heterogeneous distribution of
pale gray to yellowish and white colors, as suggested by previous studies⁴⁸. Most of the
typical alteration discolorization is found along the upper rim of the Gran Cratere and
outer, older crater rims. Some alteration sites are extensive, stretching over 425 m long
and 145 m wide. These altered regions also contain **interspersed sulfur deposit patches**,
primarily concentrated at the CFZ (Fig. 1D, 2A). The alteration spreads inward from the
CFZ on the northern crater crest and the inner crater wall, terminating to the south at
the crater floor. **In all other directions, the brightly-colored altered material sharply**
**transitions into reddish-brown colored material in all other directions, marking the**
**contact between altered and unaltered regions.**

The drone survey also highlighted another alteration zone on the northern outer flank, in
the Forgia Vecchia region, where the altered material fans westward and descends
toward the village of Vulcano (Fig. 1C, 2A). Altered areas are also observed on the
southern flanks and within the boundaries of the 1988 landslide areas. These regions,
characterized by hydrothermal deposits and alteration, are often challenging to map
accurately using true-color data alone, highlighting the need for a more effective feature
extraction method. Consequently, we applied a statistical technique (PCA, see
methods) to reduce data dimensionality and isolate spectral variations related to
hydrothermal deposits, thereby enhancing our mapping efforts.

Our PCA resulted in three principal components (PC1, PC2, and PC3), which account
for approximately 95.6%, 2.7%, and 1.6% of the total variance in the data, respectively
(Supplementary Table 1). While PC1 captures brightness differences well, it does not
include the necessary color change detection for alteration mapping and is therefore
excluded from further analysis (Fig. 2B). PC2 highlights the CFZ as dark gray,
representing altered material, and clearly distinguishes it from the surrounding lighter
gray, unaltered surfaces based on color transitions. PC3 accounts for the smallest
portion of the total variance and highlights vegetated regions along the lower outer
flanks and sulfur deposits within the main fumarole field (Fig. 2D).

By recombining these components, we develop a false-color composite raster, created
by stacking the three individual components, PC1, PC2, and PC3, into the red, green,
and blue channels of the composite, respectively (Fig. 2E). The red-to-pink hues
effectively delineate the CFZ and highlight features that are less discernible in the true-
color orthomosaic. The red-to-pink regions include the CFZ and OFZ regions (Fig. 2E),
the Forgia Vecchia collapse site, the 1988 landslide, and the sections affected by the
2010 landslide. The blue to blue-purple tones correspond to unaltered regions, while
green hues emphasize varying vegetation in the lower-flank areas. This differentiation
between altered, unaltered, and vegetated surfaces results from the PCA's ability to
enhance spectral variations associated with surface alteration and bleaching, thereby
maximizing variance between surface types, and allowing hydrothermally altered areas

to be separated from other materials⁵⁵. Consequently, regions highlighted in red-to-pink
within the composite raster correspond to hydrothermally altered surfaces, making the
PC composite raster a suitable candidate for feature extraction.

**Figure 2: Orthomosaic and Principal Components (PC).** **A)** True-color orthomosaic of the study area,
providing a high-resolution reference. **B)** PC1 highlights the most variance within the dataset; **C)** PC2
emphasizes the second most variation in the data and highlights color variance, notably at the CFZ; **D)**
PC3 contains the least amount of variance in the data and highlights the vegetation and sulfur deposits in
dark grey; **E)** Composite PCA image enhancing visualization between all principal components, and
effectively highlights the CFZ and OFZ and other altered region in red-to-pink colors.

*Alteration classes*

[revised manuscript text omitted]

The main north-south transect proved valuable, capturing relative rock strength across
some of the flank's steepest and **most altered regions**. For instance, the southern inner
crater wall exhibited a distinct gradient in Schmidt rebound values along its transect
(Fig. 5E). Rocks at the unaltered southern rim were among the strongest measured in
the field; however, values declined substantially at the upper portion of the southern
inner crater wall as the transect approached the Very High alteration contact (Fig. 5E).
Further downslope, rock strength increased again as the material transitioned back into
unaltered material (Fig. 5E). A similar pattern was observed on the outer northern flank,
where the weakest rocks were concentrated in the steepest sections (Fig. 5D). At lower
elevations, however, Schmidt rebound values exhibited greater heterogeneity, likely due
to increased variation in alteration intensity and localized geological influences,
particularly mass wasting, which transports rocks with differing alteration types
downslope (Fig. 5D).

Rocks classified as High and Very High alteration grades are primarily located near the
**CFZ**, where many of the weakest rocks documented during the campaign were found.

These alteration grades were the dominant surface type on the northern and southern
flanks; however, their spatial extent was more limited compared to the CFZ, resulting in

fewer measurements from those areas. Nonetheless, measurements taken near the
previously collapsed sections of the flanks consistently recorded low Schmidt rebound

values, indicating a concentration of some of the softest and weakest rocks in our field
area.

**Figure 5. Schmidt hammer measurement locations and corresponding Schmidt hammer rebound**
**values are color coded to indicate relative rock strength.** A) Distribution of all Schmidt hammer

[revised manuscript text omitted]

**Data availability**

The data (drone products and Schmidt hammer measurements) will be published on
GFZ Dataservices (<https://dataservices-cms.gfz.de/>) under a CC by 4.0 license. We are
currently in the process of providing the data and will change this statement once it is
completed.

Reference List

- Ramalho, R. S. *et al.* Hazard potential of volcanic flank collapses raised by new
megatsunami evidence. *Science Advances* **1**, e1500456 (2015).
<https://doi.org/10.1126/sciadv.1500456>
- Moore, J. G. & Moore, G. W. Deposit from a giant wave on the island of Lanai, Hawaii.
*Science* **226**, 1312-1315 (1984). <https://doi.org/10.1126/science.226.4680.1312>
- Watt, S. F. L. *et al.* Combinations of volcanic-flank and seafloor-sediment failure offshore
Montserrat, and their implications for tsunami generation. *Earth and Planetary Science*
*Letters* **319-320**, 228-240 (2012). <https://doi.org/10.1016/j.epsl.2011.11.032>
- Kim, Y. C., Wauthier, C. & Walter, T. R. Satellite Geodesy Uncovers 15 m of Slip on a
Detachment Fault Prior to the 2018 Collapse at Anak Krakatau, Indonesia. *Geophysical*
*Research Letters* **51**, e2024GL112296 (2024). <https://doi.org/10.1029/2024GL112296>
- Siebert, L. & Reid, M. E. Lateral edifice collapse and volcanic debris avalanches: a post-
1980 Mount St. Helens perspective. *Bulletin of Volcanology* **85**, 61 (2023).
<https://doi.org/10.1007/s00445-023-01662-z>
- 6 Acocella, V. *Volcano-Tectonic Processes*. Ch.6 pp 568-577 (Springer, 2021).
<http://dx.doi.org/10.1007/978-3-030-65968-4>
- 7 Day, S. J., Heleno da Silva, S. I. N. & Fonseca, J. F. B. D. A past giant lateral collapse
and present-day flank instability of Fogo, Cape Verde Islands. *Journal of Volcanology*

*and Geothermal Research* **94**, 191-218 (1999). [https://doi.org/10.1016/S0377-](https://doi.org/10.1016/S0377-0273(99)00103-1)
[0273\(99\)00103-1](https://doi.org/10.1016/S0377-0273(99)00103-1)

8 Hunt, J. E., Cassidy, M. & Talling, P. J. Multi-stage volcanic island flank collapses with
coeval explosive caldera-forming eruptions. *Scientific Reports* **8**, 1146 (2018).
<https://doi.org/10.1038/s41598-018-19285-2>

9 Elsworth, D. & Voight, B. Evaluation of volcano flank instability triggered by dyke
intrusion. *Geological Society, London, Special Publications* **110**, 45-53 (1996).
<https://doi.org/10.1144/GSL.SP.1996.110.01.03>

10 Alparone, S., Bonaccorso, A., Bonforte, A. & Currenti, G. Long-term stress-strain
analysis of volcano flank instability: The eastern sector of Etna from 1980 to 2012.
*Journal of Geophysical Research: Solid Earth* **118**, 5098-5108 (2013).
<https://doi.org/10.1002/jgrb.50364>

Borgia, A., Ferrari, L. & Pasquarè, G. Importance of gravitational spreading in the
tectonic and volcanic evolution of Mount Etna. *Nature* **357**, 231-235 (1992).
<https://doi.org/10.1038/357231a0>

Goto, Y., Danhara, T. & Tomiya, A. Catastrophic sector collapse at Usu volcano,
Hokkaido, Japan: failure of a young edifice built on soft substratum. *Bulletin of*
*Volcanology* **81**, 37 (2019). <https://doi.org/10.1007/s00445-019-1293-x>

Day, S. J. Hydrothermal pore fluid pressure and the stability of porous, permeable
volcanoes. *Geological Society, London, Special Publications* **110**, 77-93 (1996).
<https://doi.org/10.1144/GSL.SP.1996.110.01.06>

Heap, M. J. *et al.* Hydrothermal alteration can result in pore pressurization and volcano
instability. *Geology* **49**, 1348-1352 (2021). <https://doi.org/10.1130/G49063.1>

Reid, M. E. Massive collapse of volcano edifices triggered by hydrothermal
pressurization. *Geology* **32**, 373-376 (2004). <https://doi.org/10.1130/G20300.1>

Voight, B. & Elsworth, D. Failure of volcano slopes. *Geotechnique* **47**, 1-31 (1997).
<https://doi.org/10.1680/geot.1997.47.1.1>

Van Wyk de Vries, B. & Francis, P. W. Catastrophic collapse at stratovolcanoes induced
by gradual volcano spreading. *Nature* **387**, 387-390 (1997).
<https://doi.org/10.1038/387387a0>

Ball, J. L., Calder, E. S., Hubbard, B. E. & Bernstein, M. L. An assessment of
hydrothermal alteration in the Santiaguito lava dome complex, Guatemala: implications
for dome collapse hazards. *Bulletin of Volcanology* **75**, 676 (2013).
<https://doi.org/10.1007/s00445-012-0676-z>

19 Heap, M. J. *et al.* Alteration-Induced Volcano Instability at La Soufrière de Guadeloupe
(Eastern Caribbean). *Journal of Geophysical Research: Solid Earth* **126**, (2021).
<https://doi.org/10.1029/2021JB022514>

20 López, D. L. & Williams, S. N. Catastrophic Volcanic Collapse: Relation to Hydrothermal
Processes. *Science* **260**, 1794-1796 (1993).
<https://doi.org/10.1126/science.260.5115.1794>

21 Poland, M. P., Peltier, A., Bonforte, A. & Puglisi, G. The spectrum of persistent volcanic
flank instability: A review and proposed framework based on Kīlauea, Piton de la
Fournaise, and Etna. *Journal of Volcanology and Geothermal Research* **339**, 63-80
(2017). <https://doi.org/10.1016/j.jvolgeores.2017.05.004>

- Cecchi, E., van Wyk de Vries, B. & Lavest, J.-M. Flank spreading and collapse of weak-
cored volcanoes. *Bulletin of Volcanology* **67**, 72-91 (2004).
<https://doi.org/10.1007/s00445-004-0369-3>
- Darmawan, H. *et al.* Hidden mechanical weaknesses within lava domes provided by
buried high-porosity hydrothermal alteration zones. *Scientific Reports* **12**, 3202 (2022).
<https://doi.org/10.1038/s41598-022-06765-9>
- Reid, M. E., Sisson, T. W. & Brien, D. L. Volcano collapse promoted by hydrothermal
alteration and edifice shape, Mount Rainier, Washington. *Geology* **29**, 779-782 (2001).
[https://doi.org/10.1130/0091-7613\(2001\)029<0779:VCPBHA>2.0.CO;2](https://doi.org/10.1130/0091-7613(2001)029<0779:VCPBHA>2.0.CO;2)
- Heap, M. J. *et al.* Imaging strain localisation in porous andesite using digital volume
correlation. *Journal of Volcanology and Geothermal Research* **404**, 107038 (2020).
<https://doi.org/10.1016/j.jvolgeores.2020.107038>
- Crowley, J. K. & Zimbelman, D. R. Mapping hydrothermally altered rocks on Mount
Rainier, Washington, with airborne visible/infrared imaging spectrometer (AVIRIS) data.
*Geology* **25**, 559-562 (1997).
[https://doi.org/10.1130/0091-7613\(1997\)025%3C0559:MHAROM%3E2.3.CO;2](https://doi.org/10.1130/0091-7613(1997)025%3C0559:MHAROM%3E2.3.CO;2)
- Darmawan, H., Walter, T. R., Troll, V. R. & Budi-Santoso, A. Structural weakening of the
Merapi dome identified by drone photogrammetry after the 2010 eruption. *Nat. Hazards*
*Earth Syst. Sci.* **18**, 3267-3281 (2018). <https://doi.org/10.5194/nhess-18-3267-2018>
- Müller, D. *et al.* Anatomy of a fumarole field: drone remote-sensing and petrological
approaches reveal the degassing and alteration structure at La Fossa cone, Vulcano,
Italy. *Solid Earth* **15**, 1155-1184 (2024). <https://doi.org/10.5194/se-15-1155-2024>
- Heap, M. J. *et al.* The tensile strength of hydrothermally altered volcanic rocks. *Journal*
*of Volcanology and Geothermal Research* **428**, 107576 (2022).
<https://doi.org/10.1016/j.jvolgeores.2022.107576>
- Pola, A. *et al.* Geomechanical characterization of the Miocene Cuitzeo ignimbrites,
Michoacán, Central Mexico. *Engineering Geology* **214**, 79-93 (2016).
<https://doi.org/https://doi.org/10.1016/j.enggeo.2016.10.003>
- Weydt, L. M. *et al.* The impact of hydrothermal alteration on the physiochemical
characteristics of reservoir rocks: the case of the Los Humeros geothermal field
(Mexico). *Geothermal Energy* **10**, 20 (2022).
<https://doi.org/10.1186/s40517-022-00231-5>
- Browne, L. R. P. Hydrothermal Alteration in Active Geothermal Fields. *Annual Review of*
*Earth and Planetary Sciences* **6**, 229-248 (1978).
<https://doi.org/10.1146/annurev.ea.06.050178.001305>
- Wyring, L. *et al.* Mechanical and physical properties of hydrothermally altered rocks,
Taupo Volcanic Zone, New Zealand. *Journal of Volcanology and Geothermal Research*
**288**, 76-93 (2014). <https://doi.org/10.1016/j.jvolgeores.2014.10.008>
- Kereszturi, G., Schaefer, L. N., Miller, C. & Mead, S. Hydrothermal Alteration on
Composite Volcanoes: Mineralogy, Hyperspectral Imaging, and Aeromagnetic Study of
898 Mt Ruapehu, New Zealand. *Geochemistry, Geophysics, Geosystems* **21**, (2020).
<https://doi.org/10.1029/2020GC009270>
- Finn, C. A., Deszcz-Pan, M., Ball, J. L., Bloss, B. J. & Minsley, B. J. Three-dimensional
geophysical mapping of shallow water saturated altered rocks at Mount Baker,

Washington: Implications for slope stability. *Journal of Volcanology and Geothermal*
*Research* **357**, 261-275 (2018). <https://doi.org/10.1016/j.jvolgeores.2018.04.013>

Selva, J. *et al.* Multiple hazards and paths to eruptions: A review of the volcanic system
of Vulcano (Aeolian Islands, Italy). *Earth-Science Reviews* **207**, 103186 (2020).
<https://doi.org/10.1016/j.earscirev.2020.103186>

Totaro, C. *et al.* New insights on the active degassing system of the Lipari–Vulcano
complex (South Italy) inferred from Local Earthquake Tomography. *Scientific Reports*
**12**, 18867 (2022). <https://doi.org/10.1038/s41598-022-21921-x>

De Astis, G., La Volpe, L., Peccerillo, A. & Civetta, L. Volcanological and petrological
evolution of Vulcano island (Aeolian Arc, southern Tyrrhenian Sea). *Journal of*
*Geophysical Research: Solid Earth* **102**, 8021-8050 (1997).
<https://doi.org/10.1029/96JB03735>

De Astis, G., Doronzo, D. M. & Di Vito, M. A. A review of the tectonic, volcanological and
hazard history of Vulcano (Aeolian Islands, Italy). *Terra Nova* (2023).
<https://doi.org/10.1111/ter.12678>

Madonia, P., Cangemi, M., Costa, M. & Madonia, I. Mapping fumarolic fields in volcanic
areas: A methodological approach based on the case study of La Fossa cone, Vulcano
island (Italy). *Journal of Volcanology and Geothermal Research* **324** (2016).
<https://doi.org/10.1016/j.jvolgeores.2016.05.014>

Keller, J. THE ISLAND OF VULCANO. *Rendic Soc Ital Mineral Petro* **36**, 369-414
(1980).

Di Traglia, F., Pistolesi, M., Bonadonna, C. & Rosi, M. The last 1100 years of activity of
La Fossa caldera, Vulcano Island (Italy): new insights into stratigraphy, chronology, and
landscape evolution. *Bulletin of Volcanology* **86**, 47 (2024).
<https://doi.org/10.1007/s00445-024-01738-4>

Mercalli, G. & Silvestri, O. *Le eruzioni dell'isola di Vulcano incominciate il 3 agosto 1888*
*e terminate il 22 marzo 1890: relazione scientifica della Commissione incaricata degli*
*studi dal R. Governo.* (Stab. Bontempelli, 1891).

Aiuppa, A. *et al.* Mafic magma feeds degassing unrest at Vulcano Island, Italy.
*Communications Earth & Environment* **3**, 255 (2022). [https://doi.org/10.1038/s43247-](https://doi.org/10.1038/s43247-022-00589-1)
[022-00589-1](https://doi.org/10.1038/s43247-022-00589-1)

Federico, C. *et al.* Inferences on the 2021 Ongoing Volcanic Unrest at Vulcano Island
(Italy) through a Comprehensive Multidisciplinary Surveillance Network. *Remote Sensing*
**15** (2023). <https://doi.org/10.3390/rs15051405>

Graziani, A., Tommasi, P., Rotonda, T. & Bevivino, C. Preliminary analysis of instability
phenomena at Vulcano Island, Italy 147-154 (2007).
<http://dx.doi.org/10.1201/NOE0415451406.ch19>

Mannini, S., Harris, A. J. L., Jessop, D. E., Chevrel, M. O. & Ramsey, M. S. Combining
ground-and ASTER-based thermal Measurements to Constrain fumarole field heat
budgets: The case of Vulcano Fossa 2000–2019. *Geophysical Research Letters* **46**,
11868-11877 (2019). <https://doi.org/10.1029/2019GL084013>

Müller, D., Bredemeyer, S., Zorn, E., De Paolo, E. & Walter, T. R. Surveying fumarole
sites and hydrothermal alteration by unoccupied aircraft systems (UAS) at the La Fossa

cone, Vulcano Island (Italy). *Journal of Volcanology and Geothermal Research* **413**,
107208 (2021). <http://dx.doi.org/10.1016/j.jvolgeores.2021.107208>

Pesci, A., Teza, G., Casula, G., Fabris, M. & Bonforte, A. Remote Sensing and Geodetic
Measurements for Volcanic Slope Monitoring: Surface Variations Measured at Northern
Flank of La Fossa Cone (Vulcano Island, Italy). *Remote Sensing* **5**, 2238-2256 (2013).
<http://dx.doi.org/10.3390/rs5052238>

Chiodini, G., Cioni, R. & Marini, L. Reactions governing the chemistry of crater fumaroles
from Vulcano Island, Italy, and implications for volcanic surveillance. *Applied*
*Geochemistry* **8**, 357-371 (1993). [https://doi.org/10.1016/0883-2927\(93\)90004-Z](https://doi.org/10.1016/0883-2927(93)90004-Z)

Italiano, F., Pecoraino, G. & Nuccio, P. M. Steam output from fumaroles of an active
volcano: Tectonic and magmatic-hydrothermal controls on the degassing system at
Vulcano (Aeolian arc). *Journal of Geophysical Research: Solid Earth* **103**, (1998).
<https://doi.org/https://doi.org/10.1029/98JB02237>

Madonia, P. *et al.* Shallow landslide generation at La Fossa cone, Vulcano island (Italy):
a multidisciplinary perspective. *Landslides* **16**, 921-935 (2019).
<http://dx.doi.org/10.1007/s10346-019-01149-z>

Barde-Cabusson, S. *et al.* New geological insights and structural control on fluid
circulation in La Fossa cone (Vulcano, Aeolian Islands, Italy). *Journal of Volcanology*
*and Geothermal Research* **185**, 231-245 (2009).
<https://doi.org/10.1016/j.jvolgeores.2009.06.002>

Marsella, M., *et al.*, Stability Conditions and Evaluation of the Runout of a Potential
Landslide at the Northern Flank of La Fossa Active Volcano, Italy, in *Landslide Science*
*and Practice*. 2013. p. 309-314. <http://dx.doi.org/10.1007/978-3-642-31310-3-41>

Crósta, P. A. & Moore, M. J. Geological mapping using Landsat Thematic Mapper
imagery in Almeria Province, south-east Spain. *International Journal of Remote Sensing*
**10**, 505-514 (1989). <https://doi.org/10.1080/01431168908903888>

Revil, A. *et al.* Inner structure of La Fossa di Vulcano (Vulcano Island, southern
Tyrrhenian Sea, Italy) revealed by high-resolution electric resistivity tomography coupled
with self-potential, temperature, and CO₂ diffuse degassing measurements. *Journal of*
*Geophysical Research: Solid Earth* **113** (2008).
<https://doi.org/https://doi.org/10.1029/2007JB005394>

Aydin, A. & Basu, A. The Schmidt hammer in rock material characterization. *Engineering*
*geology* **81**, 1-14 (2005). <https://doi.org/10.1016/j.enggeo.2005.06.006>

Basu, A. & Aydin, A. A method for normalization of Schmidt hammer rebound values.
*International journal of rock mechanics and mining sciences* **41**, 1211-1214 (2004).
<http://dx.doi.org/10.1016/j.ijrmms.2004.05.001>

Harnett, C. E. *et al.* Evolution of Mechanical Properties of Lava Dome Rocks Across the
1995–2010 Eruption of Soufrière Hills Volcano, Montserrat. *Frontiers in Earth Science* **7**
(2019). <https://doi.org/10.3389/feart.2019.00007>

Katz, O., Reches, Z. e. & Roegiers, J.-C. Evaluation of mechanical rock properties using
a Schmidt Hammer. *International Journal of Rock Mechanics and Mining Sciences* **37**,
723-728 (2000). [http://dx.doi.org/10.1016/S1365-1609\(00\)00004-6](http://dx.doi.org/10.1016/S1365-1609(00)00004-6)

- Samaniego, P. *et al.* The historical (218 ± 14 aBP) explosive eruption of Tutupaca
volcano (Southern Peru). *Bulletin of Volcanology* **77**, 51 (2015).
<https://doi.org/10.1007/s00445-015-0937-8>
- Voight, B. *et al.* The 26 December (Boxing Day) 1997 sector collapse and debris
avalanche at Soufriere Hills Volcano, Montserrat. *Geological Society, London, Memoirs*
**21** (2002). <https://doi.org/10.1144/GSL.MEM.2002.021.01.17>
- de Vries, B. v. W., Kerle, N. & Petley, D. Sector collapse forming at Casita volcano,
Nicaragua. *Geology* **28**, 167-170 (2000). [https://doi.org/10.1130/0091-7613\(2000\)28<167:SCFACV>2.0.CO;2](https://doi.org/10.1130/0091-7613(2000)28<167:SCFACV>2.0.CO;2)
- Watters, R. J., Zimbelman, D. R., Bowman, S. D. & Crowley, J. K. Rock mass strength
assessment and significance to edifice stability, Mount Rainier and Mount Hood,
Cascade Range volcanoes. *Pure and Applied Geophysics* **157**, 957-976 (2000).
<http://dx.doi.org/10.1007/s000240050012>
- Mathieu, L. Quantifying Hydrothermal Alteration: A Review of Methods. *Geosciences* **8**,
no. 7 (2018). <https://doi.org/10.3390/geosciences8070245>.
- Poganj, A., Heap, M. J. & Baud, P. Spatial distribution of alteration and strength in a lava
dome: Implications for large-scale volcano stability modelling. *Journal of Volcanology*
*and Geothermal Research* (2025).
<https://doi.org/https://doi.org/10.1016/j.jvolgeores.2025.108344>
- Heap, M. *et al.* The influence of heterogeneity on the strength of volcanic rocks and the
stability of lava domes. *Bulletin of Volcanology* **85** (2023).
<https://doi.org/10.1007/s00445-023-01669-6>
- Matthews, J. A. & Winkler, S. Schmidt-hammer exposure-age dating: a review of
principles and practice. *Earth-Science Reviews* **230**, 104038 (2022).
<https://doi.org/10.1016/j.earscirev.2022.104038>
- Schöpa, A., Pantaleo, M. & Walter, T. R. Scale-dependent location of hydrothermal
vents: Stress field models and infrared field observations on the Fossa Cone, Vulcano
Island, Italy. *Journal of Volcanology and Geothermal Research* **203**, 133-145 (2011).
<https://doi.org/10.1016/j.jvolgeores.2011.03.008>
- Heap, M. J. *et al.* Hydrothermal alteration of andesitic lava domes can lead to explosive
volcanic behaviour. *Nature Communications* **10**, 5063 (2019).
<https://doi.org/10.1038/s41467-019-13102-8>
- Harnett, C. E., Heap, M. J., Troll, V. R., Deegan, F. M. & Walter, T. R. Large-scale lava
dome fracturing as a result of concealed weakened zones. *Geology* **50**, 1346-1350
(2022). <https://doi.org/10.1130/G50396.1>
- Granados-Bolaños, S., Quesada-Román, A. & Alvarado, G. E. Low-cost UAV
applications in dynamic tropical volcanic landforms. *Journal of Volcanology and*
*Geothermal Research* **410**, 107143 (2021).
<https://doi.org/10.1016/j.jvolgeores.2020.107143>
- James, M. R. *et al.* Volcanological applications of unoccupied aircraft systems (UAS):
Developments, strategies, and future challenges. *Volcanica* **3**, 67-114 (2020).
<https://doi.org/10.30909/vol.03.01.67114>
- Westoby, M. J., Brasington, J., Glasser, N. F., Hambrey, M. J. & Reynolds, J. M.
'Structure-from-Motion' photogrammetry: A low-cost, effective tool for geoscience

- applications. *Geomorphology* **179**, 300-314 (2012).
<https://doi.org/10.1016/j.geomorph.2012.08.021>
- Jiang, S., Jiang, C. & Jiang, W. Efficient structure from motion for large-scale UAV
images: A review and a comparison of SfM tools. *ISPRS Journal of Photogrammetry and*
*Remote Sensing* **167**, 230-251 (2020). <http://dx.doi.org/10.1016/j.isprsjprs.2020.04.016>
- Snavely, N., Seitz, S. M. & Szeliski, R. Modeling the world from internet photo
collections. *International Journal of Computer Vision* **80**, 189-210 (2008).
<http://dx.doi.org/10.1007/s11263-007-0107-3>
- Lillesand, T. M., Kiefer, R. W. & Chipman, J. W. *Remote sensing and image*
*interpretation*. 6th edn, pp. 527-529; 545-550 (John Wiley & Sons, 2008).
- Lillesand, T., Kiefer, R. W. & Chipman, J. *Remote Sensing and Image Interpretation*.
pp. 524-529; 556-560 (Wiley, 2015).
- Crósta, A. P. & Moore, J. M. Geological mapping using Landsat Thematic Mapper
imagery in Almeria Province, south-east Spain. *International Journal of Remote Sensing*
**10**, 505-514 (1989). <https://doi.org/10.1080/01431168908903888>
- Marzban, P. *et al.* Hydrothermally altered deposits of 2014 Askja landslide, Iceland,
identified by remote sensing imaging. *Frontiers in Earth Science* **11** (2023).
<https://doi.org/10.3389/feart.2023.1083043>
- Schaefer, L. N., Kereszturi, G., Villeneuve, M. & Kennedy, B. Determining physical and
mechanical volcanic rock properties via reflectance spectroscopy. *Journal of*
*Volcanology and Geothermal Research* **420**, 107393 (2021).
<https://doi.org/https://doi.org/10.1016/j.jvolgeores.2021.107393>
- Goudie, A. S. The Schmidt Hammer in geomorphological research. *Progress in Physical*
*Geography* **30**, 703-718 (2006). <https://doi.org/10.1177/0309133306071954>
- Schmidt, E. A non-destructive concrete tester. *Concrete* **59**, 34-35 (1951).
- Matthews, J. A. *et al.* A rock-surface microweathering index from Schmidt hammer R-
values and its preliminary application to some common rock types in southern Norway.
*CATENA* **143**, 35-44 (2016). <https://doi.org/10.1016/j.catena.2016.03.018>
- Rosas-Carbajal, M., Komorowski, J.-C., Nicollin, F. & Gibert, D. Volcano electrical
tomography unveils edifice collapse hazard linked to hydrothermal system structure and
dynamics. *Scientific Reports* **6**, 29899 (2016). <https://doi.org/10.1038/srep29899>
- Kim, H., Hyun, C.-U., Park, H.-D. & Cha, J. Image Mapping Accuracy Evaluation Using
UAV with Standalone, Differential (RTK), and PPP GNSS Positioning Techniques in an
Abandoned Mine Site. *Sensors* **23** (2023). <https://doi.org/10.3390/s23135858>
- Harnett, C. E. & Heap, M. J. Mechanical and topographic factors influencing lava dome
growth and collapse. *Journal of Volcanology and Geothermal Research* **420**, 107398
(2021). <https://doi.org/10.1016/j.jvolgeores.2021.107398>

Acknowledgments

B.F. De Jarnatt thanks GFZ-Potsdam for financial support. We thank INGV Palermo for
resources and logistical support in the field and Section 4.6 at GFZ-Potsdam for using

their Schmidt hammer(s). We also thank Julia Nikutta for her assistance in the field.
This work was supported by a European Research Council Synergy Grant (ERC-
ROTTnROCK-101118491). M.J. Heap is also supported by ANR grant MYGALE
("Modelling the physical and chemical Gradients of hydrothermal alteration for warning
systems of flank collapse at explosive volcanoes"; ANR-21-CE49-0010) and
acknowledges support from the Institut Universitaire de France (IUF).

**Author contributions**

BDJ led this paper's work, field campaign, analysis, and writing of this paper. TRW
supervised the project and contributed to the fieldwork and analysis. MJH was the co-
supervisor and contributed to the analysis and co-supervision. DM contributed to the
field works, data, analysis, and data. AFP contributed to the fieldwork. All co-authors
contributed to the writing of the manuscript.

**Competing interests**

The authors declare no competing interests.

**Supplementary information (In separate word doc)**

**Table 1: PCA eigen value statistics**

**Table 2: Schmidt hammer summary statistics**

**Table 3: All Schmidt hammer values and GPS coordinates**

Manuscript Title: Hydrothermal weakening and slope instability at Vulcano (Italy)
analyzed using drones and in-situ strength measurements

Manuscript Number: COMMSENV-25-1704

Our replies are found **in red**.

Response to Editor

***Present novel and fully supported insight into the assessment of volcanic slope instability using drones and in situ measurements.

Response: Agreed and changes made. In the revised version, we now even further highlight the novelty of this study by clearly outlining how our integrated approach of combining UAV-based photogrammetry, PCA-based spectral analysis, and Schmidt hammer measurements offers a high-resolution, spatially continuous assessment of hydrothermal alteration and related mechanical weakening. This method is particularly effective in steep, hazardous, and otherwise inaccessible areas, addressing a key gap in volcano monitoring. The Introduction and Discussion sections now explicitly highlight this.

***Clarify the research gap, outline your method and all parameters used, justify the choice of the supervised classification algorithm, resolve statistical interpretation, and quantitatively compare your approach, referencing similar applications.

Response: Agreed and changes made. In response, we have made the following improvements: The research gap is now more clearly defined in the Introduction, emphasizing the need for remote approaches to assess mechanical weakening in areas where direct sampling is not possible. The Methods section has been expanded to include all relevant parameters and to justify our use of the Maximum Likelihood classifier for supervised classification, supported by appropriate citations. Our statistical interpretation has been strengthened with the addition of a Kruskal–Wallis test, post-hoc pairwise comparisons, and bootstrapped 95% confidence intervals. These are discussed in the revised Results, Discussion, and Methods sections. We now include quantitative comparisons with previous studies (e.g., Merapi, La Soufrière de Guadeloupe), demonstrating consistency between our findings and data from other studies that show that the compressive strength of volcanic rocks is reduced as a function of increasing hydrothermal alteration.

***Expand the discussion of your findings, addressing generality and uniqueness, and demonstrate that all your data and analysis fully support your claims.

Response: Agreed and changes made. We have substantially revised the Discussion section to distinguish site-specific insights from findings with broader relevance. To support the generality of our approach, we compare our results to similar studies at other volcanic systems, including Merapi, Colima, and Ruapehu. Additionally, we reference previous work at La Fossa that employed hyperspectral, drone-based, and thermal remote sensing to map hydrothermal alteration. These studies support the validity of our spectral classification approach, while our integration of in situ strength measurements provides a novel contribution. We also explicitly link the results of our statistical analyses, including Kruskal–Wallis testing, post-hoc comparisons, and bootstrapped confidence intervals, to the observed mechanical weakening, ensuring our conclusions are well supported by the data.

***meet our editorial thresholds as outlined below

Response: Done. We made sure our manuscript complies with our editorial policies, and we completed and uploaded the checklist as a Related Manuscript file type with the revised article.

REVIEWER COMMENTS:

Reviewer #1 (Remarks to the Author):

Manuscript number: COMMSENV-25-1704

Hydrothermal weakening and slope instability at Vulcano (Italy) analyzed using drones and in-situ strength measurements.

In order to determine the correlations with alteration types in the Vulcano Islands and assess the degree of stability, this research uses an integrated strategy using in situ Schmidt hammer and UAV-based photogrammetry.

These findings support earlier research on possible failure. For hydrothermalized massifs, a new classification is suggested.

I've made an effort to evaluate this manuscript and value its scientific content. The subsurface conditions and limitations have been considered, implying future studies. It gives me great pleasure to suggest that the manuscript be published.

Response: We thank the reviewer for the supportive evaluation and are pleased that the scientific merit of the work is recognized. We appreciate the recommendation for publication.

Reviewer #2 (Remarks to the Author):

Dear authors

I have evaluated the manuscript title: "Hydrothermal weakening and slope instability at Vulcano (Italy) analyzed using drones and in-situ strength measurements" prepared for publication in Communications Earth & Environment (COMMSENV-25-1704). The manuscript presents a contribution to the assessment of volcanic slope instability using drone-based photogrammetry and in-situ Schmidt hammer measurements. While the study is methodologically solid and potentially impactful, several key aspects require clarification and deeper analysis. In particular, the discussion lacks sufficient critical depth and comparative evaluation with existing literature. The authors should better justify methodological choices (e.g., classification parameters) and quantitatively compare their approach, ideally using benchmark metrics, or referencing similar applications at other volcanic sites. Furthermore, the conclusions are too general and need to be better grounded in the data, with clearer articulation of limitations and applicability conditions.

Kind regards
Antonio Pola

Response: We thank reviewer #2 for the thoughtful and constructive assessment. We appreciate the recognition of the methodological strength and potential impact of our study. In response to the points raised, we have revised the manuscript to improve the clarity and critical depth of the discussion, strengthened the justification for our methodological choices, and added comparisons to relevant literature where appropriate. Additionally, the conclusions have been refined to more directly reflect the data and acknowledge the limitations and context of our findings. Detailed responses to these general and each specific comment are provided below **in red**, following your original statement.

The following are my main observations and recommendations:

1. Abstract: The abstract is generally well structured, but its clarity could be improved by more explicitly stating the research gap and the novelty of the proposed approach.

Response: We appreciate this helpful suggestion and have made appropriate changes. The revised abstract now more clearly states the difficulty of assessing rock weakening on steep, inaccessible volcanic flanks, emphasizing the lack of current understanding or capabilities as the research gap. We have also clarified the novelty of our integrated approach, which combines drone photogrammetry and in situ Schmidt hammer testing,

to demonstrate its value in mapping hydrothermal alteration weakening and slope instability in hazardous volcanic environments.

To improve the abstract, we revised the following sentences:

“Instability at volcanic edifices poses significant hazards, yet the processes driving rock weakening, particularly on steep, eroding flanks, remain poorly understood due to limited accessibility”. (Tracked Change: p. 1, lines: 20-22)

“Here, we developed an integrated method combining drone photogrammetry with in situ Schmidt hammer testing to derive an empirical alteration-to-strength relationship for the crater rim and applied this knowledge to alteration sites on inaccessible flanks”. (Tracked Change: p. 1, lines 28-31)

While the methodology is briefly mentioned, it omits the crucial use of PCA-based remote sensing, which underpins the classification of hydrothermal alteration.

Response: We agree with this observation and have revised the abstract to clearly state the role of our principal component analysis (PCA) in our method. The PCA was used as an aid to highlight spectral variance within the data, in the drone-derived RGB imagery, serving as the basis for both the supervised and unsupervised classifications of hydrothermal alteration. This clarification emphasizes PCA's critical function in isolating altered zones and enabling quantitative surface mapping.

To improve the abstract, we refined the following sentence:

“An alteration was mapped using Principal Component Analysis (PCA) to aid our classifications.” (Tracked change: p. 1, lines 32-34)

The reported ~50% reduction in rock strength is compelling, but would be more informative if contextualized with specific Schmidt rebound values or strength ranges.

Response: Thank you for this suggestion. We have included the full range of Schmidt hammer rebound values in the abstract (10.5 to 82) to provide the requested context for the strength reductions.

To improve the abstract, we refined the following sentence:

“This map was used to transpose over 1,000 Schmidt hammer measurements (R-values ranging from 10.5 to 82) were transposed onto our thematic strength-alteration map.” (Tracked Change: p. 1, lines 34-38)

Finally, the abstract would benefit from a stronger concluding sentence that highlights the broader implications of the method for volcanic hazard monitoring and early-warning systems.

Response: We appreciate this suggestion and have revised the final sentence of the abstract to better emphasize the broader applicability of our integrated approach. Specifically, we now highlight the broader applicability of our approach to hazard monitoring and early-warning systems in active-volcanic environments where traditional monitoring is limited.

We refined the following sentence:

“This integrated approach offers a transferable workflow for assessing volcanic slope instability, with direct applications to hazard monitoring and early warning systems.”
(Tracked Change: p. 1, lines 39-41)

2. Introduction: The introduction is informative but lacks focus and a clear statement of the research gap.

Response: We thank the reviewer for this helpful comment. In the revised Introduction, we clarify the study’s focus early on and explicitly define the research gap. The revised text frames the work in the context of volcanic flank collapse and highlights hydrothermal alteration as an under-recognized yet critical factor in assessing related hazards.

We thus refined the following sentences:

“However, despite their high impact, such large-scale events are relatively rare, and our ability to forecast them remains limited. Documented observations are confined to only a few well-studied cases^{1,2}. “
(Tracked Change: p. 2, lines 59-62)

“As a result, the link between hydrothermal alteration and mass wasting is usually established in the aftermath, based on hydrothermally altered material within debris deposits¹⁷.”
(Tracked Change: p. 3, lines 89-91)

“These studies demonstrate not only the global relevance of hydrothermal weakening but also the persistent challenge of identifying and quantifying altered material before failure occurs. “
(Tracked Change: p. 4, lines 108-111)

It reviews many contributing factors to volcanic flank collapse, yet delays in identifying what is specifically missing in the literature. The introduction cites numerous case studies, but their integration feels list-like. While these examples illustrate the global relevance of hydrothermal weakening, they are not clearly tied to the central research question.

Response: We appreciate the reviewer's thoughtful feedback. In response, we revised the introduction to provide a more concise overview of contributing processes, followed immediately by a focused discussion of hydrothermal alteration. We now more clearly identify the lack of established methods to assess its role in flank collapse and better connect the cited case studies to the central research question.

We thus refined the following sentences:

“Hydrothermal alteration provides insight into both the timing and location of collapses, yet it often affects steep, hazardous, or otherwise inaccessible areas, making it challenging to monitor²⁴. As a result, the link between hydrothermal alteration and mass-wasting is usually established in the aftermath, based on hydrothermally altered material in debris deposits²¹. Evidence of this relationship has been observed in past collapse events, including those at La Soufrière de Guadeloupe (France), Las Casitas (Nicaragua), Mount Hood and Mount Rainier (USA), Santiaguito (Guatemala), Merapi (Indonesia), Volcán de Colima (Mexico), and elsewhere^{21,25-27}. “
(Tracked Change p. 3, lines 84-95)

“These studies demonstrate not only the global relevance of hydrothermal weakening but also the persistent challenge it poses for volcanic hazard assessment, specifically, the difficulty of identifying altered, unstable material before failure occurs. “
(Tracked Change: p. 4, lines 108-111)

Additionally, while several relevant examples of volcanoes affected by hydrothermal weakening are cited, the transition to the case study at La Fossa is abrupt and would benefit from a stronger rationale for its selection.

Response: We appreciate the reviewer's observation. While the rationale for selecting La Fossa was briefly noted in the original version, we agree that the transition was abrupt. In response, we have revised the introduction to more clearly link the broader challenge of identifying hydrothermal weakening prior to collapse with the unique advantage of La Fossa as a study site. Specifically, we improved the transition from relevant examples to La Fossa and the benefits that the region has for developing and testing our approach.

We thus refined the following sentences:

“Consequently, there is a pressing need to develop methods for identifying areas susceptible to alteration-driven flank failure. With its active hydrothermal system and frequent landslides, the La Fossa cone of Vulcano Island (Italy) (Fig. 1) serves as an ideal natural laboratory for developing and testing these methods, ultimately improving the forecasting of larger, more hazardous volcanic collapses. Although the landslides at La Fossa are orders of magnitude smaller than sector collapses, their frequency and accessibility offer an opportunity to develop a workflow that can later be scaled to assess volcanic flanks of greater size and destructive potential. These flank failures at La Fossa are thought to be driven, in part, by hydrothermal alteration, making it an ideal site for identifying altered regions prone to future flank instability. “

(Tracked Change: p.3, Lines 133-144)

3. Study area: The section is detailed and provides solid geologic, tectonic, and eruptive context. However, it reads more like an extended descriptive summary than a targeted justification for the research. While the tectonic and eruptive background is relevant, the discussion could be condensed to emphasize the aspects most pertinent to instability and hydrothermal alterations, specifically geomorphological changes, landslide history, and degassing patterns.

Response: We thank the reviewer for this observation. In response, we have reorganized and revised the Study Area section to shift the emphasis from general background toward aspects directly relevant to hydrothermal alteration and slope instability. Specifically, we have condensed general tectonic and eruptive history to improve the focus on current morphological processes and current hazards, and reorganized the section to give more prominence to fumarolic activity, degassing patterns, and their links to hydrothermal alteration and mechanical weakening. We also clearly highlighted the 1988 and 2010 landslide events and their relevance for the influence of hydrothermal alteration on flank instability, and restructured the text to better frame why La Fossa is an ideal site to investigate alteration-driven instability.

We thus refined the following sentences:

“Past works at La Fossa have also mapped zones undergoing hydrothermal alteration using geochemical³¹, thermal⁵⁶, and remote sensing approaches^{31,57}, establishing La Fossa's suitability for examining how hydrothermal processes contribute to volcanic flank instability. “

(Tracked Change: p.9, lines 277-280)

“The recent 1988 and 2010 collapses at La Fossa, combined with persistent degassing and widespread hydrothermal alteration, offer a unique opportunity to investigate the controls on volcanic slope stability and alteration-related weakening.”

(Tracked Change p. 12, lines 318-323)

“Further signs of instability are evident along the northern flank, particularly in the Forgia Vecchia regions (Fig. 1A, C), which exhibit active deformation, as indicated by geodetic surveys⁵⁴.”

(Tracked Change: p.11, lines 305-307)

The 1988 and 2010 mass-wasting events are important, but their connection to hydrothermal weakening should be more explicitly framed to reinforce the rationale for site selection.

Response: We agree with the reviewer and with the revision of the Study Area section; we have better framed the text, highlighting the significance of the 1988 and 2010 events to reinforce the rationale for the La Fossa cone site selection.

We thus revised the following sentences:

“The recent 1988 and 2010 collapses at La Fossa, combined with persistent degassing and widespread hydrothermal alteration, offer a unique opportunity to investigate the controls on volcanic slope stability and alteration-related weakening. Here, we combine high-resolution drone imagery with in situ strength testing (see Methods) in accessible regions to infer relative rock strength across the otherwise inaccessible steep flanks.”

(Tracked Change: p. 12, lines 318-323)

4. Results: The Results section presents a rich and well-organized dataset. However, the section remains descriptive and would benefit from more quantitative depth.

Response: We thank the reviewer for this helpful observation. We agree that the original manuscript was more qualitative rather than quantitative. In response, we have significantly added more quantitative detail in the manuscript and have integrated the results in several areas of the revised Results section. Specifically, we have summarized field-measured rock strength, presenting statistical summaries for ~1,100 Schmidt rebound measurements, including mean, median, quartiles, 95th percentiles, and added bootstrapped 95% confidence intervals for each alteration grade. Also, we assess statistical significance, reporting results of a Kruskal–Wallis test ($p < 0.0001$) with Bonferroni-corrected post hoc comparisons, showing Unaltered significantly differs from all altered categories, and Low differs from High and Very High alteration ($p < 0.001$). We present transect-based quantitative patterns, describing rebound value

changes along specific profiles, including sharp declines in strength when crossing into altered zones and recovery in unaltered areas. For example, along the C–C' profile, rebound values drop from Very Strong (>65) and Strong (≤ 65) at the unaltered rim to Weak (≤ 35) and Very Weak (≤ 20) within mapped Very High alteration zones, before increasing again downslope in unaltered material.

We thus refined the following sentences:

“Unaltered material... exhibits the highest rebound values, averaging 60.1 with a median of 63 [95% CI: 62.00–65.00]... Very High alteration exhibits the lowest values, with a median of 27.0 [95% CI: 23.75–30], representing the weakest material observed in the field.”

(Tracked Changes: pg 27, lines 603–613)

“Our Kruskal-Wallis test confirms that differences in Schmidt hammer values across alteration classes are statistically significant ($p < 0.0001$). Post-hoc pairwise comparisons with Bonferroni correction reveal that Unaltered rocks significantly differ from all alteration categories, and that Low alteration is also significantly different from both High and Very High classes ($p < 0.001$) (see Methods: *In-situ Rock Hardness Assessment*). However, Medium, High, and Very High groups are not statistically separable, consistent with increasing overlap in rebound distributions at greater alteration intensities (Fig. 6).“

(Tracked Change: p.28, lines 623-631)

“Rocks at the unaltered southern rim were among the strongest measured in the field, with Schmidt rebound values in the Very Strong ($R > 65$) to Strong ($R \leq 65$) range. As the transect progressed northward toward the steepest section of the inner crater wall, it intersected surfaces mapped as Very High and High. This area coincided with a decline in relative rock strength, showing Weak ($R \leq 35$) and Very Weak ($R \leq 20$) Schmidt hammer values (Fig. 5E).“

(Tracked Change: p 24, lines 544-550)

While the observed ~50% reduction in Schmidt hammer values across alteration classes is compelling, statistical analyses (e.g., confidence intervals, significance tests, or effect size metrics) are missing and would strengthen the interpretation.

Response: We thank the reviewer for this insightful comment. In response, we have incorporated additional statistical analyses into the Results, Discussion, and Methods sections of the revised manuscript. We have now included a Kruskal-Wallis test to evaluate overall differences in rock strength across alteration classes, followed by Bonferroni-corrected post-hoc comparisons to identify statistically distinct groupings. In

addition, we calculated bootstrapped 95% confidence intervals for the median rebound values of each alteration class, which is particularly well-suited for our non-normal distribution of data. These results are now clearly presented in the text, Figure 6, Table 2, and in the Supplementary Materials. These data support our initial claim and conclusion that rock strength decreases with increasing alteration.

We thus refined the following sentences:

“Despite the widespread variability in Schmidt rebound values, our summary statistics (mean, quartiles, and 95th percentile, and bootstrapped 95% confidence intervals) reveal a distinct pattern: Schmidt rebound values decrease systematically with increasing alteration intensity.” (Tracked Change: p. 27, lines 600-603)

“Our Kruskal-Wallis test confirms that differences in Schmidt hammer values across alteration classes are statistically significant ($p < 0.0001$). Post-hoc pairwise comparisons with Bonferroni correction reveal that Unaltered rocks significantly differ from all alteration categories, and that Low alteration is also significantly different from both High and Very High classes ($p < 0.001$). However, Medium, High, and Very High groups are not statistically separable, consistent with increasing overlap in rebound distributions at greater alteration intensities.” (Tracked Change: p.28, lines 623-631)

Additionally, the results rely heavily on visual interpretation of figures, with limited engagement with the supplemental tables, particularly Table 2, which contains key summary statistics. Explicitly referencing these tables and integrating their content into the narrative would improve data transparency and traceability.

Response: We appreciate the reviewer's feedback regarding the integration of supplementary data. In response, we have incorporated Tables 1 and 2 from the supplementary materials directly into the main text. These now appear on p.11, line 288, and p. 21, line 497, respectively, in the revised manuscript. We now explicitly reference these tables within the Results section and have integrated the summary statistics into the narrative to enhance data transparency and support the interpretation of the figures.

(Tracked Change: p. 14, line 399 and p. 28, line 633)

Finally, the spatial patterns of weakening are discussed in qualitative terms but could be enhanced by linking them more systematically to mapped alteration zones and collapse-prone sectors, reinforcing the hazard implications of the study.

Response: We thank the reviewer for this suggestion. We agree and have revised the Results section to include quantified Schmidt rebound values across the transects,

specifically along the northern flank (B-B') and the southern crater wall (C-C'). These values are now directly linked to mapped alteration (e.g., Very High, High), showing consistent correspondence between low rebound values (Weak to Very Weak) and highly altered regions. Additionally, we clarify spatial trends along these transects and the role of mass wasting in distributing altered material down slope. Together, these revisions provide a more systematic and spatially resolved interpretation of weakening patterns and reinforce the hazard implications by connecting low-strength regions and collapse-prone sectors. Excerpts were added in several places in the results section to discuss the results in more quantitative terms.

We thus revised the following sentences:

“Along the B-B' elevation profile, near the edge of the collapse scar, Schmidt rebound values predominantly fell within the Moderate ($R \leq 50$) to Very Weak ($R \leq 20$) range, indicating decreased relative rock strength at the scarp crest. As the B-B' profile progresses northward toward the lower portion of Forgia Vecchia and, within the mapped alteration, most rocks fell into the Very Weak (≤ 20) to Weak (≤ 35) categories.” (Tracked Changes: pg 24 Lines 557-563)

“The Very Weak rebound values are spatially associated with High to Very High alteration, including both regions of past mass-wasting and intact slopes showing signs of surface weakening. This spatial alignment corresponds to a localized zone of reduced strength consistent with mapped alteration intensity.” (Tracked Changes: pg 30 Lines 661-665)

5. Discussions: The discussion lacks the critical comparative depth and analytical rigor expected for a study addressing volcanic slope instability. While it reinforces the link between hydrothermal alteration and mechanical weakening, it does so primarily through site-specific correlations.

Response: We thank the reviewer for this observation. In response, we have significantly restructured and expanded the comparative context of our findings. Specifically, we have referenced similar laboratory-based studies from Merapi (Darmawan et al. 2022) and La Soufrière de Guadeloupe (Heap et al., 2021, Poganj et al., 2025) that documented a similar decreasing trend with increasing hydrothermal alteration. These studies are used to demonstrate that the observed ~50% reduction in Schmidt hammer values aligns with decreasing strength values also reported at other volcanoes. In addition, this information was initially in the Introduction; however, we saw fit to relocate it to the Discussion section for a more logical flow. This cross-site consistency reinforces the broader relevance of our results beyond Vulcano Island.

We thus revised the following sentences:

“Although these values provide relative rock strength rather than absolute strength, the observed ~50% reduction in strength aligns well with past laboratory-based studies, which observed a similar trend in rock strength weakening of hydrothermally altered volcanic rocks. For example, at La Soufrière de Guadeloupe (Eastern Caribbean), two independent studies reported a decrease in rock strength with increasing alteration: Heap et al.¹⁸ measured UCS values decreasing from 140–60 MPa to less than 10 MPa, and Poganj et al.⁷⁰ found average values of ~82 MPa for pristine rock and ~6.5 MPa for extremely altered rock, showing that rock strength decreases systematically with increasing Alteration Grade Index (AGI), a visual metric incorporating discoloration. Similarly, at Merapi (Indonesia), Darmawan et al.²⁶ reported a decrease in UCS values from ~132 MPa in fresh dome rocks to ~11 MPa in fully altered rocks, correlated with an increase in porosity, an increase in $\delta^{18}\text{O}$, and, most relevant for this study, an increase in rock discoloration. These parallels support the use of the Schmidt hammer rebound values as a practical, field-based proxy for assessing alteration-driven rock strength weakening.”

(Tracked Change: p. 32-33, lines 718-734)

The ~50% reduction in Schmidt hammer values is not compared with similar findings at other volcanoes, limiting the reader’s ability to assess the generality or uniqueness of the observations.

Response: We thank the reviewer for pointing this out, and this concern has been directly addressed by comparing our field-based results with published UCS data from other volcanoes, as stated in the point above. We explicitly cite studies that report decreasing UCS values ~132 MPa in fresh rocks to ~11 MPa in altered rocks, with documented links to porosity, $\delta^{18}\text{O}$, and discoloration. This comparison supports the general applicability of our approach and validates Schmidt hammer measurements as a field-scale proxy.

We thus revised the following sentences:

“Although these values provide relative rock strength rather than absolute strength, the observed ~50% reduction in strength aligns well with past laboratory-based studies, which observed a similar trend in rock strength weakening of hydrothermally altered volcanic rocks. For example, at La Soufrière de Guadeloupe (Eastern Caribbean), two independent studies reported a decrease in rock strength with increasing alteration: Heap et al.¹⁸ measured UCS values decreasing from 140–60 MPa to less than 10 MPa, and Poganj et al.⁷⁰ found average values of ~82 MPa for pristine rock and ~6.5 MPa for extremely altered rock, showing that rock strength decreases systematically with increasing Alteration Grade Index (AGI), a visual metric incorporating discoloration.

Similarly, at Merapi (Indonesia), Darmawan et al.²⁶ reported a decrease in UCS values from ~132 MPa in fresh dome rocks to ~11 MPa in fully altered rocks, correlated with an increase in porosity, an increase in $\delta^{18}\text{O}$, and, most relevant for this study, an increase in rock discoloration. These parallels support the use of the Schmidt hammer rebound values as a practical, field-based proxy for assessing alteration-driven rock strength weakening.“

(Tracked Change: p. 32-33, lines 718-734)

Moreover, the inference that low rebound values reflect pre-collapse weakening is not critically examined, considering alternative explanations, such as post-failure exposure or clast transport. The discussion would also benefit from the inclusion of additional physical properties (such as porosity), which are often reported alongside Schmidt hammer values and would provide a more complete mechanical characterization of altered rocks.

Response: We thank the reviewer for this suggestion and agree that alternative explanations for low rebound values should be considered. We have revised the Discussion to acknowledge the potential influence of post-failure exposure and clast transport, while reinforcing our interpretation of pre-collapse weakening based on spatial correspondence with mapped alteration and in situ conditions. While porosity data were beyond the scope of this study, we now note that our observed strength trends align with previous work linking porosity and discoloration to mechanical weakening, supporting the robustness of our findings.

We thus revised the following sentences:

“While our interpretation assumes that low Schmidt hammer rebound values reflect pre-collapse weakening, we acknowledge other alternative mechanisms, such as post-failure weathering or clast transport, that could influence our measurements. Although we are confident that the reduced Schmidt rebound values result from hydrothermal alteration, we cannot entirely rule out the contribution of weathering in rocks exposed at the surface for prolonged periods.“

(Tracked Change: p. 32, lines 736-742)

“While porosity and mineralogical data are often used to complement mechanical interpretations, such measurements were beyond the scope of this field-based study. Indeed, measuring porosity would have significantly decreased our number of Schmidt hammer rebound values, which was the goal of this study. Nonetheless, the observed trend weakening correlates with visible discoloration and mapped alteration zones, which closely align with works at other volcanoes that included porosity data, providing

independent support for the interpretation of hydrothermal-induced rock weakening³⁰.“
(Tracked Change: p. 33, lines 770-777)

Lastly, while the workflow is presented as broadly applicable, the discussion fails to clearly define its operational limits, decision thresholds, or how it could be integrated into real-time hazard monitoring. A more systematic and comparative approach would significantly enhance the interpretive power and relevance of the study.

Response: We thank the reviewer for this valuable comment. In the revised Discussion and *Implications* sections, we have clarified the operational limits of our workflow, noting its dependence on surface visibility, image resolution, and spectral separability of alteration classes. We also explicitly state that, while not a real-time tool per se, the workflow enables high-frequency monitoring and can act as a near-real-time proxy for mechanical weakening. Additionally, we now describe how establishing relationships between alteration class and rebound strength allows for the generation of time series data using drones, an approach particularly suited to inaccessible or hazardous terrain.

We thus revised the following sentences:

“While this workflow is broadly applicable, it has operational limits. The workflow effectiveness depends on surface exposure, image resolution, and the spectral distinguishability of alteration intensity. These limitations may reduce accuracy in the presence of shadowing or vegetated areas.”

(Tracked Change: p 35, lines 785-793)

“While our workflow is not a real-time monitoring tool in the strictest sense, it enables high-frequency, repeated assessments of alteration and flank conditions. More broadly, it lays the foundation for a workflow that, once refined, could serve as a near-real-time proxy for mechanical weakening in evolving volcanic settings—offering a valuable, low-risk complement to traditional monitoring techniques, particularly where access is restricted.”

(Tracked Change: p. 37, lines 842-848)

6. Conclusions: The conclusions need to be more direct and grounded in the evidence presented. Key statements (such as the method being scalable or transferable) are too broad and lack support from comparative benchmarks or clearly defined conditions.

Response: We thank the reviewer for this constructive feedback. We have revised the language around the scalability and transferability to reflect more defined conditions under which our method is applicable. Specifically, we now state its use is constrained by factors such as surface exposure, image resolution, and spectral distinctiveness.

This change grounds the method's scalability in real-world field conditions rather than generalizing its applicability.

We thus revised the following sentences:

“While the applicability of our approach is constrained by operational factors such as surface exposure, image resolution, and spectral distinctiveness within the mapped alteration, the correlation between mapped alteration classes and in situ rock strength underscores the robustness of this method as a proxy for identifying mechanically weakened regions. The absence of subsurface constraints (e.g., geophysical or petrological data) limits our ability to assess depth-dependent weakening; however, this limitation reflects a common challenge in many volcanic settings, where only surface outcrops are available. Within this context, our protocol remains well-suited for field-based hazard evaluation.”

(Tracked Change: p. 39, lines 902-911)

The section also overlooks important limitations discussed earlier, such as the lack of subsurface data.

Response: We directly addressed the reviewer's concern by acknowledging the lack of subsurface data in the conclusion. We address this limitation as a common constraint in volcanic terrains, especially those that are hazardous or difficult to access. This addition helps clarify the scope and practical applicability of our field-based protocol.

We thus added the following sentences:

“The absence of subsurface constraints (e.g., geophysical or petrological data) limits our ability to assess depth-dependent weakening; however, this limitation reflects a common challenge in many volcanic settings, where only surface outcrops are available. Within this context, our protocol remains well-suited for field-based hazard evaluation.”

(Tracked Change: p. 39, lines 907-911)

For a study aiming at high-impact relevance, the conclusions must convey precise, evidence-based takeaways and clearly define the scope and limits of applicability.

Response: We revised the conclusions to convey the key takeaways, while defining the scope and limits of the applicability of our approach. The findings from our data have highlighted known areas of weakness. We discuss the correlation with past slope failures, providing additional comments that clarify the operational scope and limits of our method, as well as its relevance for a field-based assessment. These revisions build

upon the earlier changes to ensure that the conclusions are consistent, comprehensive, and aligned with the reviewer's request.

We thus added the following sentences:

“While the applicability of our approach is constrained by operational factors such as surface exposure, image resolution, and spectral distinctiveness within the mapped alteration, the correlation between mapped alteration classes and in situ rock strength underscores the robustness of this method as a proxy for identifying mechanically weakened regions. The absence of subsurface constraints (e.g., geophysical or petrological data) limits our ability to assess depth-dependent weakening; however, this limitation reflects a common challenge in many volcanic settings, where only surface outcrops are available. Within this context, our protocol remains well-suited for field-based hazard evaluation. By enabling the direct detection of structurally weak zones, this framework overcomes many limitations of traditional geotechnical methods and can be applied to other volcanic systems worldwide, providing critical insights into the role of hydrothermal alteration in the long-term stability of volcanic edifices.”

(Tracked Change: p. 39, lines 902-911)

7. Methods: The Methods section is detailed and technically solid, but its placement after the Results disrupts the logical flow and hinders reproducibility.

Response: We appreciate the reviewer's concern and recognize the importance of accessibility and reproducibility. However, the current structure follows the required formatting guidelines of *Nature Communications Earth & Environment*, which places the Methods section after the Results and Discussion.

To address the reviewer's concern, we have improved the manuscript by adding explicit cross-references to the Methods and relevant subsections throughout the text. These changes ensure that readers can easily locate and understand the methodological context while following the narrative flow of the results.

Moreover, while the PCA and classification workflows are described in depth, the rationale behind parameter choices (e.g., number of classes, selection of training zones) could be better justified with references or validation metrics.

Response: We appreciate the reviewer's thoughtful feedback. In fact, a nearly similar concern was raised by another reviewer; therefore, this prompted us to significantly expand and clarify the rationale behind both our supervised classification approach in the revised manuscript, specifically in the Supervised and *Unsupervised Classification of Alteration Zones* subsection of the Methods.

We have now added clarification for the number of classes used in the unsupervised classification. The selection of a four-class alteration intensity class (Very High, High, Moderate, and Low) was based on capturing spectral variability, while avoiding over-segmentation and complicating interpretability. Thus, using a four-class system allowed for more straightforward interpretation, while still preserving relevant surface spectral variation. This rationale was further supported in the text, citing Cihlar et al. (2014), which expands on the rationale of using a smaller number of meaningful classes to reduce complexity.

In addition, we have now provided and expanded justification for our training area selection. The region we refer to as the Central Fumarole zone (CFZ) was chosen for training site selection because it has been a well-documented, well-recognized, and thoroughly investigated site of persistent fumarolic activity and hydrothermal alteration. Multiple independent studies have confirmed the extent and type of alteration in this area. Here, we used three different studies to support our reasoning.

- 1) Azzarini et al. (2001) mapped silicic and advanced argillic alteration zones using MIVIS hyperspectral data, in which the authors highlighted advanced alteration within the CFZ.
- 2) Harris et al. (2009) used thermal remote sensing methods to identify persistent heap anomalies associated with degassing in the CFZ. Regardless of their method of classification (e.g., Lower, Middle, and Upper zones), they clearly outlined the region and made a correlation of surface temperature with discoloration, and outlined the region of interest.
- 3) Müller et al. (2024) integrated high-resolution drone imagery, PCA-based feature extraction, and gas flux measurements to map zones of surface alteration and fumarolic degassing within the CFZ. Their use of PCA to enhance discoloration patterns closely parallels the approach employed in our study, and their findings confirmed a correlation between degassing and visibly bleached surfaces, further validating the CFZ as a site of well-documented alteration.

We thus revised the following sentences:

“These training areas served as spectral reference inputs for the Maximum Likelihood classifier, allowing each pixel to be assigned to the most spectrally similar category^{100,101}. Our justification for training area selection is based on the CFZ, a well-documented region of persistent fumarolic activity and hydrothermal alteration. Previous studies have mapped this region as an area of silicic and advanced argillic alteration using hyperspectral data⁶⁹, identified thermal anomalies associated with persistent fumarolic activity⁶⁸, and similar mapping methods as ours using drone-PCA methods, but combined with gas flux measurements³³. This cross-validation from independent

datasets supports the CFZ as a reliable and validated spectral endmember for a supervised classification. Therefore, this region is well-suited for our training area site selection for our supervised classification of hydrothermal alteration.”

(Track Change: p. 43, lines: 1013-1025)

The use of both supervised and unsupervised classification is appropriate, but the transition between them is abrupt and would benefit from a clearer explanation of their complementary roles.

Response: We thank the reviewer for this helpful observation. In response, we have revised the transition between the supervised and unsupervised classification sections to clarify their complementary roles. Specifically, we added a brief sentence that explains how the unsupervised classification builds upon the previous supervised method, capturing the internal variability within the previously defined altered zone.

Thus we added the following sentence:

“To investigate variations within the altered zone itself, we applied an unsupervised classification, complementing the binary output of the supervised approach.”

(Tracked Change: p. 43, lines 1028-1029)

The Schmidt hammer protocol is well-detailed, including limitations and the decision to use single-shot measurements. Nonetheless, the absence of cross-validation with UCS laboratory data, although acknowledged, remains a limitation for quantitative modeling.

Response: We appreciate the reviewer’s recognition of our detailed Schmidt hammer protocol and acknowledgment of its limitations. We agree that the absence of UCS laboratory data limits direct calibrations and constrains the use of our results for quantitative modeling that requires UCS as an input parameter. However, given that our study prioritized identifying spatial patterns of mechanical weakening rather than deriving absolute strength values, the Schmidt hammer provided a practical and effective proxy.

This approach allowed us to efficiently detect zones of relative weakness across altered volcanic terrains. As now clarified in the Discussion, these relative strength patterns can be used to streamline future sampling strategies by highlighting mechanically weak sarees that warrant more targeted UCS testing or detailed geotechnical investigation.

We thus revised the following sentences:

“Although our Schmidt hammer measurements are not UCS-calibrated, they effectively capture spatial patterns of mechanical weakening. This strengthens our approach and

highlights the Schmidt hammer as a powerful tool to streamline future fieldwork by guiding sampling strategies and prioritizing zones for a detailed mechanical analysis.” (Tracked Changes: p. 34, lines 753-757)

The authors should clarify the logic and validation behind classification choices and discuss whether calibration or comparison with UCS data is possible to support the strength estimates.

Response: We thank the reviewer for this valuable comment. We have clarified and revised in the Methods that UCS calibration or conversions were not applied, as existing empirical relationships are typically developed for unaltered lithologies and would not accurately reflect the altered rocks at Vulcano. In the absence of site-specific UCS testing, we adopted a relative approach using unconverted Schmidt hammer R-values. This method is widely accepted for field-based strength assessments and has been successfully applied in similar contexts (e.g., Matthews et al., 2022), making it particularly suited for the simple workflow developed in this study.

We also clarified the logic behind our classification choices. Rebound values were grouped into five strength categories (very weak to very strong) to support spatial comparisons. Given the range of alteration intensity, this framework provides a consistent basis for internal spatial comparison.

In addition, the Schmidt hammer itself was calibrated with the producer-supplied calibration anvil. This was performed before the device was used in the field.

We thus added the following sentences:

“This approach allowed us to explore the spatial relationship between surface alteration intensity and in situ rock strength. To facilitate interpretation and comparison, we grouped Schmidt hammer rebound values into five relative strength categories based on R-values: very weak (≤ 20), weak (≤ 35), moderate (≤ 50), strong (≤ 65), and very strong (≥ 65). These categories are not UCS-calibrated but reflect rebound value ranges commonly used for relative surface strength assessments⁷³. Given the range of alterations, intensity, and absence of site-specific UCS testing, this classification provides a consistent framework for internal comparison for spatial trends.”

(Tracked Change: p. 47, lines 1138-1146)

Figures:

Please, take care of the following aspects. 1) Labels in a composite figure should not differ in order, size, style, and position.

Response: We appreciate the reviewer's feedback. We have adjusted all of the panels in Figure 2 (A, B, C, D, E) and ensured that all images are in the correct order, of similar size, with a similar font style, and with the label placement consistent across all figures. To clarify, the labels in the composite image (Fig. 2E) are included to emphasize the highlighted features that were not present in the other image panels. With these suggestions, the figure on p.12, line 311 of the revised manuscript, now maintains consistency and avoids any visual confusion. In addition, we reviewed all figures in the manuscript to maintain uniform formatting, thereby improving visual consistency and avoiding any potential confusions.

(Tracked Change: p. 17, line 419)

Include the direction in which the photo was taken; if possible, include a graphic scale.

Response: Thank you for this helpful suggestion. We have updated the figure caption (Figures 1C, D, and E) to include the camera viewing direction for all photos. This change can be found on p. 8, line 211 of the revised manuscript. In additions we have added a scale to Figures 1C, D, and E).

(Tracked Change: p. 11, line 288)

The range and scale of the composite figures should be the same to avoid any confusion; a different scale is presented in Figure 2; consequently, visual comparison of the graphs could be difficult.

Response: We agree with the reviewer's concern. To improve visual comparability, we have revised Figure 2 to ensure that all figure panels contain the same range and scale. This enables direct visual comparison between panels without distortion or misinterpretation. The revised figure can be found below, and also on p. 11, line 305 of the revised manuscript.

(Tracked Change: p. 17, line 419)

3. Reviewer Comment

It must be ensured that when composite figures are scaled, the labels are not deformed on any of their axes.

Response: Thank you for bringing this to our attention. We have ensured that our modifications to the previous suggestions have not distorted the labels or scales across all panels in Figure 2. All revisions of Figure 2 can be found on p.11, line 305 of the revised manuscript. In addition, we reviewed all other figures in the manuscript to verify that scaling did not affect label formatting (e.g., latitude/longitude labels, panel titles). This ensures uniform text style and scale across all figures and avoids any deformation or visual inconsistencies.

(Tracked Change: p. 17, lines 419)

Reviewer #3 (Remarks to the Author):

This study addresses a very interesting topic focused on monitoring hydrothermal alteration zones to identify potential collapses. The study area is La Fossa volcano, on Vulcano Island (Aeolian Archipelago, Italy). The work is based on the generation of orthomosaics using unmanned aerial vehicles and the Schmidt hammer method to evaluate rock strength in relation to hydrothermal alteration, with the goal of producing thematic maps. The authors performed an interpolation of the results, as most of the hydrothermal alteration zones are largely inaccessible. They found that the results indicate a 50% reduction in rock strength, which they associate with areas of degassing and hydrothermal activity, and which also coincide with zones affected by mass-wasting events.

I believe this is a very novel and interesting research; however, I think there are some methodological issues that the authors could consider to improve their work.

Response: We appreciate the constructive comments regarding improvements to the Methods section. In the revised manuscript, we have carefully addressed the reviewer's concerns by further clarifying the limitations of our method, particularly regarding surface-based measurements, the interpolation of inaccessible regions, and the absence of UCS calibration. We have strengthened the rationale for our classification choices and field strategy, and emphasized the exploratory nature of our approach in contexts where direct access is limited. These clarifications are detailed in the revised Methods, Discussion, and Limitations sections, and we believe they enhance both the robustness and transparency of our study.

It would be important to highlight somewhere in the text that hydrothermal alteration can also lead to a strengthening of the rocks, for example, through silicification.

Response: We thank the reviewer for this relevant suggestion. We have revised the text in the Introduction to explicitly acknowledge that hydrothermal alteration can also lead to rock strengthening. Specifically, we now include information that silicification is a suggested process that can increase cohesion and stiffness by reducing porosity and permeability.

We thus revised the following sentences:

“While hydrothermal alteration is most often associated with rock weakening, it can, in some cases, increase rock strength. Silicification, for example, can reduce porosity and permeability by filling fractures and pore spaces with silica-rich minerals, leading to increased cohesion and stiffness. Nevertheless, strengthening as a result of alteration

has also been found to contribute to instability by facilitating pore pressure buildup and reducing permeability, ultimately promoting deformation or failure^{16,30}.”
(Tracked Change: p. 4, lines 115-124)

Please move the Methods section before the Results. Many questions related to the methodology arise while reading the results.

Response: Thank you for this suggestion. We agree that having immediate access to methodological details can help with the interpretation of the Results. However, the structure of the manuscript follows the required format of *Nature Communications Earth & Environment*, which places the Methods section after the Results and Discussion.

To address the concern, we have revised the manuscript to include additional cross-references to relevant sections of the Methods through the Results and Discussion. This ensures that methodological details are accessible at the point of need and enhances the logical flow within the constraints of the journal's formatting.

Previous works related to hydrothermal alteration zones in the study area could serve to validate some of the findings presented here, assuming the data is available.

Response: We thank the reviewer for this thoughtful suggestion. In response, we have incorporated a sentence into the *Study Area* highlighting previous work at La Fossa that mapped hydrothermal alteration zones using geochemical, thermal, and remote sensing approaches. This highlights the relevance of the field site and supports its suitability for investigating the links between hydrothermal alteration and volcanic instability.

We thus added the following sentences:

“Past works at La Fossa have also mapped zones undergoing hydrothermal alteration using geochemical³³, thermal⁶⁸, and remote sensing approaches^{33,69}, establishing the location's suitability for examining how hydrothermal processes contribute to volcanic flank instability.” (Tracked Change: pg. 9, lines 277-280)

There are some uncertainties concerning the interpretation and use of the principal components. Vegetation is identified in PC3, which is assigned to the blue channel, so vegetation should be enhanced in blue?

Response: We thank the reviewer for this comment. We agree that this should be further expanded to clarify the results; therefore, we have revised the text in the Results section. We have explicitly explained that the principal components each highlight different surface features, and these are expressed at varying grayscale values, with

some as higher (brighter) values, and others as lower (darker) ones. When combined into the RGB composite, each component contributes proportionally to its assigned color channel based on its grayscale intensity. As a result, areas of alteration appear in the red-to-pink color hues, when multiple PCs contribute simultaneously, even if some features are emphasized at lower grayscale values. We have updated the text to explain this behavior more clearly, without introducing additional technical complexity. Again, we appreciate this being pointed out, and the text should provide a clearer explanation of the interpretation.

We thus added the following sentences:

“These features highlighted in each principal component are expressed at different greyscale values. Some regions are emphasized by higher (brighter) values (e.g., PC1), while others, such as the CFZ in PC2 and vegetation in PC3, are more prominent at lower (darker) values. While each component captures useful surface characteristics, none were sufficient for alteration mapping across the entire region when used individually.” (Tracked Change: p. 14, lines 384-391)

The composite image uses $R = PC1$, $G = PC2$, and $B = PC3$, yet the text mentions that PC1 is excluded from further analysis, raising questions about why PC1 is included in the color composite image as the red channel, where alteration is enhanced. This apparent contradiction may stem from phrasing issues.

Response: We appreciate the reviewer’s comment. We agree that the phrasing in the original text may have led to confusion. To clarify: PC1, PC2, and PC3 were each evaluated individually for their suitability in alteration mapping. While PC1 captures brightness variation, it lacks spectral contrast needed to identify alteration zones on its own, and therefore is not used as a standalone extraction method. However, PC1 was still included in the composite image, as well as PC2 and PC3, creating our final PC composite image. We have revised the text to clarify that individual components were excluded only from single-band analysis, and that the combined composite was used for both interpretation and classification. This issue has now been addressed in the revised Results section.

We thus revised the following sentences:

“While each component captures useful surface characteristics, none were sufficient for alteration mapping across the entire region when used individually. Therefore, we excluded the individual PCs from single-component analysis and instead used a composite of all three to support classification and interpretation.”

(Tracked Change: p. 14, lines 388-391)

Please specify which supervised classification algorithm (e.g., Parallelepiped Classification, Maximum Likelihood Classification, Mahalanobis Distance Classification, etc.) was employed and explain the rationale for its selection.

Response: We thank the reviewer for this helpful suggestion. In response, we have clarified in the revised manuscript that we used the Maximum Likelihood Classification algorithm for the supervised classification of altered versus unaltered surfaces. This algorithm was chosen for its strong statistical basis, as it considers both the mean and variance of spectral signatures, making it especially effective at distinguishing classes with overlapping spectral features, an important factor in hydrothermal alteration mapping.

We also expanded the explanation of our training area selection, referencing prior studies that identify the Central Fumarole Zone (CFZ) as a well-documented and spectrally distinct hydrothermal alteration region. These updates have been incorporated into the revised Results and Methods sections to improve clarity, transparency, and reproducibility of the classification process.

We thus revised the following sentence (Results):

“Based on our PCA composite (Fig. 2E), we applied a classification using the Maximum Likelihood algorithm in ArcGIS Pro to delineate and highlight all regions of hydrothermal alteration.” (Tracked Changes: p. 17, lines 432-435)

We thus added the following sentence (Methods):

“These training areas served as spectral reference inputs for the Maximum Likelihood classifier, allowing each pixel to be assigned to the most spectrally similar category^{100,101}. Our justification for training area selection is based on the CFZ, a well-documented region of persistent fumarolic activity and hydrothermal alteration. Previous studies have mapped this region as an area of silicic and advanced argillic alteration using hyperspectral data⁶⁹, identified thermal anomalies associated with persistent fumarolic activity⁶⁸, and similar mapping methods as ours using drone-PCA methods, combined with gas flux measurements³³. This cross-validation from independent datasets supports the CFZ as a reliable and validated spectral endmember for a supervised classification. Therefore, this region is well-suited for our training area site selection for our supervised classification of hydrothermal alteration.” (Tracked Changes p. 43, lines 1013-1025)

Details about the training sites' locations, the results used for their delineation (color composite of the three components?), and the criteria for their selection, including the number of pixels involved, would clarify the methodology.

Response: We thank the reviewers for their feedback on this point. We have clarified the supervised classification method and training site selection in the revised manuscript. Specifically, we now state that a Maximum Likelihood classifier was used in ArcGIS Pro to classify altered and unaltered surfaces based on the composite PCA raster. Training areas were manually drawn as polygons of varying size and shape, placed both inside and outside the CFZ. These areas were selected based on surface variability visible in the PCA composite, particularly differences in color and texture. A total of approximately 100 training areas were used, 50 within the PCA highlighted regions, and 50 in the areas interpreted as unaltered. The number of pixels per training polygon was not standardized, as the goal was to capture spectral diversity rather than apply uniform sampling. These clarifications have been incorporated into the revised Methods section.

We thus added the following sentences:

“These training areas were manually drawn as polygons of varying size and shape, placed both inside and outside of the CFZ. Polygons were selected to capture the range of surface variability visible in the PCA composite, particularly the differences in color. The number of pixels per training site was not standardized, as the focus was on representing spectral diversity rather than a uniform size.” (Tracked Change: p. 43, lines 1008-1012)

“These training areas served as spectral reference inputs for the Maximum Likelihood classifier, allowing each pixel to be assigned to the most spectrally similar category^{83,84}. Our justification for training area selection is based on the CFZ, a well-documented region of persistent fumarolic activity and hydrothermal alteration. Previous studies have identified and mapped this region to the site of silicic and advanced argillic alteration using hyperspectral data⁵¹, thermal anomalies associated with persistent fumarolic activity⁵⁰, and similar mapping methods as ours using drone-PCA methods, combined with gas flux measurements²⁶.” (Tracked Change: p. 43, lines 1013-1021)

The four predefined classes appear to represent a range from low to high alteration. It would be important to know whether there is any field verification of the samples to support this interpretation.

Response: We thank the reviewer for raising this important point regarding the interpretation of our alteration classes. While we did not conduct geochemical or mineralogical analyses as part of this study, we noted that our classification results closely parallel those of Müller et al. (2024), who performed a detailed geochemical and mineralogical assessment of hydrothermal alteration at La Fossa. The spatial

distribution of alteration in our subclasses strongly aligns with their validated alteration zones, providing support that our approach captures meaningful variations in alteration intensity.

We thus revised the following sentences:

“These spatial patterns align with findings from Müller et al.³⁰, who mapped hydrothermal alteration at La Fossa and validated their classifications using geochemical and mineralogical analyses. Their study showed a systematic increase in sulfur content and a decrease in Fe₂O₃ across alteration gradients, further supporting the interpretation that our subclasses reflect meaningful variations in alteration intensity.”

(Tracked Changes p. 19, lines 474-478)

Typically, greater bleaching suggests a higher degree of alteration and consequently weaker rocks. Please clarify whether the lithology undergoing alteration remains consistent throughout the study area.

Response: We thank the reviewer for their thoughtful and constructive comment. To address these concerns, we have incorporated clarifications at appropriate locations in the manuscript.

We now clarify in the Results and Methods section that the four predefined classes are interpreted as representing increasing degrees of hydrothermal alteration, consistent with previous studies that link surface bleaching and mineralogical changes to reduced rock strength (e.g., Heap et al., 2021; Darmawan et al., 2022; Poganj et al., 2025).

Regarding lithology, while a comprehensive lithological analysis at each individual measurement site was not pursued, as it would have significantly reduced the number of measurements collected, we have explicitly addressed this point in the Limitations section. We now state that we reference existing geological maps (Piochi et al., 2009; Di Traglia et al., 2013; De Astis et al., 2013; Costa et al., 2020), which show that the surface of the La Fossa cone is predominantly composed of pyroclastic deposits. To minimize lithological variability, Schmidt hammer measurements were restricted to intact well-exposed pyroclastic surfaces that met the criteria for reliable Schmidt hammer testing (e.g., sufficient size and intact exposure). We also now note that hydrothermal overprinting can, in some cases, obscure the original rock type, and this limitation has also been explicitly acknowledged in the revised text.

We thus revised the following sentences:

“Although we did not perform a comprehensive lithological analysis at each measurement site, an effort that would have significantly reduced, rather than

maximised, the number of measurements collected, we mitigated this limitation by relying on established geological maps, which show that La Fossa cone edifice is predominantly composed of pyroclastic deposits^{36,95-98}. In an attempt to limit lithological variability, Schmidt hammer measurements were conducted, where possible, on intact and well-exposed pyroclastic rock surfaces. This approach helped to ensure that the observed strength variations primarily reflect differences in alteration intensity, while acknowledging that hydrothermal overprinting can often obscure the original rock type.” (Tracked Change: p. 49, lines 1196-1209)

9. Reviewer Comment

I recommend considering the following article, which offers many valuable insights that could be useful for your work: Pereira, M. L., Zanon, V., Fernandes, I., Pappalardo, L., & Viveiros, F. (2024). Hydrothermal alteration and physical and mechanical properties of rocks in a volcanic environment: A review. *Earth-Science Reviews*, 104754.

Revision: We thank the reviewer for this excellent suggestion. We agree that the article by Pereira et al. (2024) provides insights into the relationship between hydrothermal alteration and physical and mechanical properties of volcanic rocks. We have now incorporated this reference into the revised manuscript where relevant, and appreciate the opportunity to strengthen our discussion with this relevant contribution.

10. Reviewer Comment

I recommend reviewing the writing of the text to improve its consistency. Many sentences are very long.

Response: We thank the reviewer for this helpful comment. We have carefully reviewed the manuscript for sentence length and overall writing consistency. Several long or complex sentences have been revised to improve clarity and readability, particularly in the Introduction, Results, and Discussion sections. These edits ensure that the narrative is more concise and accessible to a broader readership while maintaining technical accuracy.

11. Reviewer Comment

The corrections are detailed in the attached PDF.
I hope my comments help to improve your work.

Response: We thank the reviewer for their reading and detailed corrections. We have reviewed the annotated PDF and incorporated the suggested changes where appropriate.

Annotated manuscript:

1. Introduction L40: Please review the terminology used, are you referring to all types of mass-wasting processes? The collapse events may be associated with volcanic debris avalanches. I suggest verifying the definition and including the appropriate citation to support this.
Response: Accepted and changes made. Yes, we agree with the suggestion. We used the appropriate terminology for the types of mass-wasting processes. Here, we used the Varnes method of classification.
(Tracked Change: p. 2 Line 39, 40, 83, and throughout the entire manuscript)
2. L43: Strikethrough text.
Response:: Accepted and changes made.
(Tracked Change: p. 2 Line 111)
3. L45: Strikethrough text
Response: Accepted and changes made.
(Tracked Change: p. 2, line 113)
4. L56-58: Hydrothermal alteration, particularly when associated with silicification, can also enhance rock strength. I agree with the weakening effect, especially due to clay formation in many epithermal and porphyry systems. However, I believe it is important to note that, in some cases, hydrothermal fluids may contribute to increasing rock competence.
Response: Accepted and changes made: We included information in certain instances that hydrothermal alteration can indeed increase rock competence. For this, we used Pereira et al. (2024) and Heap et al. (2021) as our supporting references.

We thus revised the following sentences:

“While hydrothermal alteration is most often associated with rock weakening, certain processes, such as silicification, can instead increase rock strength by reducing porosity and permeability. This occurs through the infilling of fractures and pore spaces with silica-rich minerals, leading to increased cohesion and stiffness²⁷.”

(Tracked Change: p. 4, lines 115-120)

5. L67: Strikethrough text
Response:: Accepted and changes made.
(Tracked Change: p. 4, line 152)
6. L75: Re-work Sentence L67: A key remaining issue is that the link between hydrothermal alteration and flank instability is often inferred after a collapse event based on the presence of altered materials in landslide or debris avalanche deposits.

Response: Accepted and changes made; however, after revision, the sentence was deleted and moved up in the text to satisfy other reviewer suggestions.

We thus revised the following sentences:

“As a result, the link between hydrothermal alteration and mass wasting is usually established in the aftermath, based on hydrothermally altered material within debris deposits¹⁸.”

(Tracked Change: p. 3, line 89-91)

7. L82: Please check the terminology.

Response: The 1988 and 2010 events are indeed considered as landslides, as confirmed by Madonia et al. (2019).

8. L107-110: These results highlight the need to monitor hydrothermally altered volcanic edifices, though limited and hazardous access to high-risk areas remains a major challenge for hazard assessment.

Response: Suggestion is accepted.

We thus revised the following sentences:

“Consequently, these findings highlight the need to monitor hydrothermally altered edifices; however, limited and hazardous access to high-risk areas remains a major challenge for effective hazard assessment.”

(Tracked Change: p. 6, lines 172-175)

9. L112: Discoloration is common in breached volcanoes; however, in unbreached ones, the hydrothermally altered core may not be visible at the surface.

Response: Accepted and changes made.

We thus revised the following sentences:

“However, it is not always visible at the surface, particularly in unbreached volcanoes, where hydrothermal alteration remains confined at depth. In such cases, detecting subsurface alteration is especially challenging due to limited surface expression and potential coverage by younger deposits.”

(Tracked Change: p. 6, lines 181-185)

10. L116: Change wording:

Response: Accepted and changes made, “while also” is deleted from the text.

(Tracked Change: p. 7, line 205)

11. L123-126: Sentence re-work: In this study, we used high-resolution drone data and collected extensive field strength measurements to quantify hydrothermally altered rocks and further explore the possibility of accessing otherwise inaccessible areas.

We focused on the La Fossa cone of Vulcano Island (Italy), the southernmost volcanic center of the Aeolian archipelago (Fig. 1), and extrapolated our data to regions of the volcano that are not directly accessible.

Response: Accepted and changes made.

We thus revised the following sentence:

“In this study, we address this limitation by combining high-resolution drone photogrammetry and collecting extensive field strength measurements to quantify hydrothermally altered rocks and further explore the possibility of accessing otherwise inaccessible areas.”

(Tracked Change: p. 7, line 204-207)

12. L131: change wording:

Response: Accepted and changes made, the phrasing “we established” has been added.

(Tracked Change: p. 7, line 232)

13. L188: Please include relevant citations.

L188: What previous studies have addressed hydrothermal alteration zones?

Response: We have now incorporated the relevant citations and more thoroughly considered previous studies. Specifically, we have added *Madonia et al. (2019)* as the appropriate reference for this sentence. In addition, we have included the works of *Müller et al. (2024)*, *Harris et al. (2009)*, and *Azzarini et al. (2001)* in the following sentence to further strengthen and support our writing.

(Tracked Change: p. 9, line 277)

L188: Is it possible to use existing data (particularly from more accessible areas) to validate some of the findings presented in this work?

Response: We thank the reviewer for these thoughtful suggestions. We have explained this issue in detail in the summary suggestions. In short, to support our mapping efforts, we have now incorporated a sentence that highlights previous studies that map hydrothermal alteration.

We thus added the following sentence:

“Past works at La Fossa have also mapped zones undergoing hydrothermal alteration using geochemical³⁵, thermal⁶², and remote sensing approaches^{35,63}, establishing the location's suitability for examining how hydrothermal processes contribute to volcanic flank instability.”

(Tracked Change: p. 9, line: 277-280)

14. L195: Ensure consistent coordinate formatting across all maps. For example, in this case, "38° 24N" is shown, but the minute symbol is missing after "24".

Response: Accepted and changes made, the revised Figure 1 contains the corrected formatting.

(Tracked Change: p.11, line 288)

15. L195: Add a color-coded legend.

Response: Accepted and changes made, a legend explaining the colored areas of interest has been added to Figure 1.

(Tracked Change: p.11, line 288)

16. L195: Add reference to the scars.

Response: Accepted and changes made. References Pesci et al. (2013) and Madonia et al. (2019) have been added to the figure captions.

(Tracked Change: p.11, line 288)

17. L200: lower-case “triangles”

Response: Accepted and changes made, the word “Triangles” has been corrected to “triangles”.

(Tracked Change: p. 11, line 297)

18. L200: Please specify the corresponding figure for each triangle.

Response: Accepted and changes made. To reduce figure clutter and confusion, we incorporated this change into the figure caption, explaining the direction from which Figures 1C, D, and E were taken.

We thus revised the following:

“**C)** Photograph of the northern flank, taken from the base of La Fossa facing S-SE. **D)** Photograph of the fumarole field, taken from the southern crater rim facing the N-NE. **E)** Photograph of the 1988 landslide scar, taken offshore facing S-SW.”

(Tracked Change: p.11, lines 300-302)

19. L204: lower-case “the”

Response: Accepted and changes made. “The” has been deleted from the revised figure caption.

(Tracked Change: p. 12, line 346)

20. L209: What does the monitoring consist of?

Response: Accepted and changes made. We found that GPS and seismic monitoring have been the most persistent in continued monitoring. There have been several other types of monitoring; however, they were not as permanent as the other monitoring methods.

We thus revised the following sentence:

“This region is continuously monitored using geophysical methods (e.g., GPS and seismic stations) due to its ongoing morphological slope changes⁵⁴, sub-parallel tension cracks near the slope edge⁵⁵, and proximity to the village of Vulcano and the nearby harbor (Fig. 1A).”

(Tracked Change: p. 12, lines 311-316)

21. L211: Please indicate their location on the map.

Response: Thank you for this suggestion. We have added red dashed lines to Figure 1 representing the tension fractures. Location of fractures was adapted from Marsella et al. (2013).

(Tracked Change: p.11, line 288)

22. L214: What do you mean by "calm conditions"? Does this refer to a period without eruption or fumarolic activity? Please specify the conditions, or the absence of which conditions, that prevailed when the collapse event occurred.

Response: Accepted and changes made, thank you for noticing this. The calm conditions mentioned were those described in the Madonia et al. (2013) study, referring to periods of stable seismic and fumarolic activity.

We thus revised the following sentence:

“This event occurred during calm conditions, as indicated by stable and seismic monitoring data⁵⁷, and exposed hydrothermally altered pyroclastics at the surface⁵¹.”

(Tracked Change: p. 12, line 316-318)

23. L219: Please move the Methods section before the Results. Many questions related to the methodology arise while reading the results.

Response: We recognize the reviewer's concern that methodological questions arise during the Results. However, in accordance with *Nature Communications Earth & Environment's* formatting requirements, the Methods section must follow the Discussion. To address this, we have added clear cross-references to the Methods section and its subsections throughout the Results and Discussion, ensuring readers can easily locate the relevant methodological context. This improves accessibility and logical flow without disrupting the narrative structure.

(Tracked Change: p. 14, lines 410-411, p.18, lines 475-476, and throughout the text)

24. L225: Which ones?

Response: We thank the reviewer for bringing this to our attention. We agree that the original was misleading for the reader. The word “maps” was plural, which likely led to confusion. The word has been changed to “map”, which reduces the confusion, because the previous text is specifically referring to the true-color orthomosaic map.

(Tracked Change: p. 12, line 336)

25. L228: They correspond to the typical discolored areas found on the summits of volcanoes.

Response: We thank the reviewer for their helpful observation. We have clarified in the Results section that the observed discoloration at La Fossa is consistent with the typical summit hydrothermal alteration pattern at other active volcanoes.

This text revision helps explain that the observations seen reflect a broader volcanological framework.

We thus revised the following sentence:

“This discoloration, particularly along the summit and crater rims, resembles typical hydrothermal alteration patterns observed at the summits of other active volcanoes.”

(Tracked Change: p. 13, lines 342-343)”

26. L233: How were the patches of sulfide deposits identified? Bibliography?

Response: We appreciate the reviewer's comment and have clarified the basis for identifying the sulfur deposits in the revised manuscript. The sulfur was identified by its distinctive yellow coloration in the true-color orthomosaic, a commonly accepted visual indicator of sulfur, and this interpretation was confirmed through direct field observations. This identification is consistent with previous studies as cited in the text.

We thus revised the following sentence:

“Interspersed sulfur deposits concentrated within the CFZ were identified based on yellow coloring in the true-color orthomosaic, confirmed through field observations and consistent with previous studies²⁶ (Fig. 1D, 2A, B).”

(Tracked Change: p. 13, lines 346-349)

27. L236-238: Which figure are you referring to?

Please indicate this clearly in the figure to improve the flow of the text.

It would be helpful to add a zoomed-in view of the alteration zone, highlighting the various features described in the text.

Response: We thank the reviewer for this valuable suggestion. In response, we have revised Figure 2 to include a zoomed-in view of the Central Fumarole Zone (CFZ), now presented as Fig. 2B. In this updated figure, we have:

- The boundary between the altered and unaltered regions is clearly delineated
- A dashed line has been added to outline the CFZ and to highlight the contact between the two surfaces.
- An arrow was added to highlight the sulfur deposit within the CFZ
This update now highlights the features described in the text.
In addition, figure references have been added within the text for clarification and flow.

(Tracked Change: p. 17, lines 419)

28. L240: Please indicate the alteration zone in the figure.

Response: We thank the reviewer for pointing out the potential ambiguity. Our intent was not to introduce a new alteration zone, but to describe additional

bleach (hydrothermally altered) material that can be seen in the true-color orthomosaic. To clarify this, we have revised the text to the phrase “another alteration zone” and instead state:

We thus revised the following sentence:

“Additionally, bleached material is observed along the northern outer flank, in the Forgia Vecchia region, where the altered material fans westward and descends toward the village of Vulcano (Fig. 1C, 2A).”

(Tracked Change: p. 13, lines 360-363)

29. L243-248: Please move to the "Methods" section.

Response: We appreciate the reviewer’s suggestion and acknowledge that methodological context is important for accessibility. To address this, we have added explicit cross-references to the Methods section and its relevant subsections throughout the Results and Discussion. This allows readers to readily locate methodological details while preserving the narrative flow. These revisions comply with *Nature Communications Earth & Environment* formatting requirements and improve overall clarity.

30. L252: change wording:

Response: Accepted and changes made. We deleted the original wording to improve sentence structure and clarity.

We thus revised the following sentence:

“PC1 captures overall surface brightness and highlights the OFZ and also alteration on the outer flanks; however, it lacks the necessary color change detection for outlining the CFZ (Fig. 2C).”

(Tracked Changes p.14, line 374-376)

31. L256: add “(Fig. 2C)”

Response: Accepted and changes made. Please see the revised excerpt.

(Tracked Changes p. 14, line 378)

32. L267-270:

I have some doubts regarding the interpretation and use of the principal components. Please clarify the following:

Is vegetation identified in PC3? If PC3 is assigned to the blue channel, vegetation should be enhanced in blue.

The composite is $R = PC1$, $G = PC2$, $B = PC3$, but in the text, it states that PC1 is excluded from further analysis, so why is it included in the color composite image? PC1 is assigned to the red channel, and alteration is enhanced in red.

Please explain the rationale behind this exclusion; I suspect it may be a phrasing issue.

To summarize: PC2 highlights altered material, PC3 highlights vegetated regions and sulfur deposits. That's right?

Response: We thank the reviewer for these insights. We have also responded to these issues in the reviewer summary above. To further clarify, PC1, PC2, and PC3 were each evaluated individually for their suitability for mapping hydrothermal alteration. While PC1 captures brightness variation, it lacks the spectral contrast necessary for effective standalone identification of altered areas and was therefore excluded from single-band analysis. However, it was still included in the final RGB composite (assigned to red), where it contributes structural and brightness context.

PC2 (green channel) highlights altered ground surfaces due to its spectral sensitivity to mineralogical changes, while PC3 (blue channel) enhances vegetated areas and also helps distinguish sulfur-rich zones. Importantly, these features highlighted by each component occur at different gray scale values: for instance, the CFZ in PC1 has higher values, while the PC2 has darker values that outline the CFZ boundary more clearly than the other PCs. Thus, when creating the composite, altered regions are highlighted in red (PC1 channel).

We have revised the text in the Results section to clearly state that individual principal components were excluded only from single-band classification, and that the composite (R=PC1, G=PC2, B=PC3) was used for visual interpretation and classification.

We thus added the following sentence:

“These features highlighted in each principal component are expressed at different greyscale values. Some regions are emphasized by higher (brighter) values (e.g., PC1), while others, such as the CFZ in PC2 and vegetation in PC3, are more prominent at lower (darker) values. While each component captures useful surface characteristics, none were sufficient for alteration mapping across the entire region when used individually. Therefore, we excluded the individual PCs from single-component analysis and instead used a composite of all three to support classification and interpretation.”

(Tracked Changes p. 14, lines 384-391)

“By combining these components, we developed a false-color composite raster by stacking PC1, PC2, and PC3 into the red, green, and blue channels, respectively (Fig. 2F). Each principal component contributed distinct visual information to the composite, and their combination highlighted distinct features associated with alteration.”

(Tracked Changes p. 15, lines 401-405)

33. L275: Please add E and N to the coordinates.

Response: Accepted and changes made, the revised Figure 2 contains the corrected formatting, with E and N added to the coordinates.

(Tracked Change: p. 17, line 419)

34. L284: Please specify which supervised classification algorithm (e.g., Parallelepiped Classification, Maximum Likelihood Classification, Mahalanobis Distance Classification, etc.) was used and explain the rationale for its selection.

Response: Accepted and changes made: Here, we used a supervised Maximum Likelihood classifier, which serves as a spectral input for the classifier, which assigns each pixel to a category based on its most spectrally similar correlate.

(Tracked Change: p. 17, line 433)

35. L286:

Where are the training sites located?

In which result were the training sites delineated?

Was it in the color composite of the three components?

How were the training sites selected?

What criteria were used?

How many pixels were included in the training sites?

Response: We thank the reviewers for their feedback on this point. We addressed a similar issue in the Reviewer summary at the start of our rebuttal letter. We have clarified the supervised classification method and training site selection in the revised Results and Method sections. Specifically, a Maximum Likelihood classifier was applied in ArcGIS Pro using a PCA composite raster. Approximately 100 manually digitized training areas, 50 in altered regions and 50 in unaltered areas, were selected based on color and texture variability in the PCA composite. These polygons, drawn both inside and outside the CFZ, varied in size and shape to capture spectral diversity rather than adhere to a fixed pixel count.

We thus revised the following sentence in our Results section:

“Based on our PCA composite (Fig. 2E), we applied a supervised classification using the Maximum Likelihood algorithm in ArcGIS Pro to delineate and highlight all regions of hydrothermal alteration(see Methods: *Supervised and Unsupervised Classification of Alteration Zones*). We selected 50 representative training sites within the red-to-pink hues of the PCA composite that coincided with the CFZ, a well-documented area of hydrothermal activity where elevated acid gas fluxes promote chemical leaching and surface bleaching^{28,59}.”

(Tracked Change: p. 17, lines 432-435)

We thus revised the following sentences in our Methods section:

“Following our PCA, we applied a supervised classification using the Maximum Likelihood algorithm in ArcGIS Pro to differentiate between altered and unaltered surfaces. Approximately 100 training areas were selected, with 50 placed within PCA-highlighted red-to-pink regions representing altered surfaces and 50 outside of the region representing unaltered surfaces. These training areas were manually drawn as polygons of varying size and shape, placed both inside and outside of the CFZ. Polygons were selected to capture the range of surface variability visible in the PCA composite, particularly the differences in color. The number of pixels per training site was not standardized, as the focus was on representing spectral diversity rather than a uniform size.”

(Tracked Change: p. 42, lines 1003-1012)

“Our justification for training area selection is based on the CFZ, a well-documented region of persistent fumarolic activity and hydrothermal alteration. Previous studies have mapped this region as an area of silicic and advanced argillic alteration using hyperspectral data⁵³, identified thermal anomalies associated with persistent fumarolic activity⁵², and similar mapping methods as ours using drone-PCA methods, combined with gas flux measurements²⁸. This cross-validation from independent datasets supports the CFZ as a reliable and validated spectral endmember for a supervised classification.”

(Tracked Change: p. 43, lines 1015-1022)

36. L295: Please unify the font style in both figures.

Response: Accepted and changes made, the revised Figure 3 has unified font styles. In addition, the text in the figure contains two sizes (12 and 10).

(Tracked Change: p. 18, line 448)

L295: Indicate the direction of the coordinate in Figure B, and also include the northern coordinates, which are currently missing.

Response: Accepted and changes made, the revised Figure 3 contains the corrected formatting, with E added to the coordinates.

(Tracked Change: p. 18, line 448)

37. L298: The caption could be: Hydrothermal alteration maps. A) Supervised classification. A) Unsupervised classification.

Response: We agree that this wording is more concise. The proposed suggestion for the caption of Figure 3 has been accepted.

We thus added the following sentence:

“**Figure 3.** Hydrothermal alteration maps. **A)** Supervised classification, **B)** Unsupervised classification, with black squares representing areas shown in detail in **Figure 4 A, B, C, and D.**”

(Tracked Change: p. 19, line 447-449)

38. L300: Methods section

Response: We thank the reviewer for this helpful comment and recognize the importance of accessibility. To address this, we have added explicit cross-references to the Methods section and its relevant subsections throughout the Results and Discussion. These changes provide a clear methodological context while maintaining the manuscript's narrative flow and adhering to *Nature Communications Earth & Environment* formatting requirements.

39. L302: Based on these statements in the Results section, I believe it is very important that the Methods section be presented before the Results.

Response: We thank the reviewer for this helpful comment and appreciate the emphasis on presenting methodological details early. However, *Communications Earth & Environment*, in accordance with *Nature's* formatting standards, requires the Methods section to follow the Results and Discussion. To improve accessibility within this structure, we have added explicit cross-references to the relevant subsections of the Methods throughout the Results. This ensures that readers can easily locate the methodological context while preserving the journal's mandated narrative flow.

40. L303: Please clearly specify: What do these classes represent? Greater alteration? Different amounts of minerals? Different minerals in each class? Do they correspond to different types of hydrothermal alteration (e.g., intermediate argillic, advanced argillic)?

Response: We thank the reviewer for this important comment. We have clarified what the four subclasses represent in the revised Results section. Specifically, we note that the region outlined by the supervised classification corresponds to zones of advanced argillic to acid-sulfate alteration, as previously documented at La Fossa (Azzarini et al., 2001). The four subclasses—Low, Medium, High, and Very High—reflect a progressive degree of surface discoloration or bleaching, which has been widely used in remote sensing studies as a visual proxy for the intensity of hydrothermal alteration.

We thus added the following sentence:

“This highlighted region corresponds to areas of advanced argillic to acid-sulfate alteration, as previously identified at La Fossa⁵³. Our unsupervised classification was performed in ArcGIS Pro (see Methods: Supervised and Unsupervised Classification of Alteration Zones), which segmented our alteration zone into four subclasses: Low, Medium, High, and Very High, based on the intensity of discoloration or bleaching. These subclasses reflect the increasing levels of hydrothermal alteration intensity, as inferred from the progressively brighter bleached surfaces across the region.”

(Tracked Change: p. 19, line 455-462)

41. L306: Is there any form of validation for this comparison? The text mentions previous studies on hydrothermal alteration. Would it be possible to use some test sites to correlate the type of alteration, quantity, and mineralogy with the four subclasses mentioned?

Response: We appreciate the reviewer's suggestion regarding the validation of alteration subclasses. While direct geochemical sampling was beyond the scope of this study, we compared our classification with previous work at La Fossa that included detailed geochemical and mineralogical analysis. In particular, Müller et al. (2024) identified systematic compositional changes across gradients that strongly resemble the spatial structure and intensity of our subclasses. This close correlation supports the validity of our classification and enhances confidence that it reflects meaningful hydrothermal alteration patterns. This revision has been added to the revised Results section.

We thus added the following sentence:

"These spatial patterns align with findings from Müller et al.²⁸, who mapped hydrothermal alteration at La Fossa and validated their classifications using geochemical and mineralogical analyses. Their study showed a systematic increase in sulfur content and a decrease in Fe₂O₃ across alteration gradients, further supporting the interpretation that our subclasses reflect meaningful variations in alteration intensity."

(Tracked Change: p. 19-20, lines 474-478)

42. L328: Please mark the scars.

Response: Accepted and changes made: Scars have been marked for all images. Landslide scarps have been adapted from Pesci et al. (2013) and Madonia et al. (2019).

(Tracked Change: p. 22, line 498)

43. L328: Add the legend (color)

Response: Accepted and changes made. A legend has been added to Figure 4B to explain the surface classifications.

(Tracked Change, p. 22, line 498)

44. L330: It is clearly visible that the zone of low alteration corresponds to the semicircular outline of the scarp. The areas of high alteration appear to extend in a NNW direction, which seems to coincide with a gully, and therefore with a low-lying area where material is transported.

Response: We appreciate the reviewer's close observation and that the Very High alteration zone in the 1988 landslide region closely follows a gully, which may reflect the downslope transport of altered material. This was a very good observation that was previously not identified. However, we note that similarly

altered surfaces also extend beyond the gully's bounds. This distribution suggests that the observed alteration is not solely a result of material transport, but could likely represent both in situ and remobilized hydrothermal deposits. We have revised the text in the results section to explain this feature.

We thus added the following sentence:

"In the 1988 landslide region, a distinct region of Very High alteration extends in a north-northwest direction, closely following a gully. This may, in part, reflect the downslope transport of altered material from higher elevations. However, similarly altered surfaces also extend beyond the gully, suggesting that this feature likely reflects both in situ and remobilized altered material."

(Tracked Changes: p. 20, line 485-489)

45. L336-241: This entire first part belongs in the Methods section. Only the results should remain here; please move the rest to the Methods section.

Response: We thank the reviewer for this observation, and we understand the concern regarding the placement of some of the methodological details. However, we respectfully believe that the brief explanation included at the beginning of the section is essential for clarity and proper interpretation of the data. Since the Methods section is placed at the end of the manuscript in the *Nature* format, a minimal amount of context is necessary in the Results to ensure the readers unfamiliar with the Schmidt hammer, and its explanation of the measurements acting as a proxy for UCS measurements is essential in the text so the reader can immediately understand the meaning and relevance of the rebound values presented throughout.

Additionally, the explanation provides critical information about how the Schmidt rebound values relate to rock strength and why no UCS conversion was applied, and how the values were used as a relative strength proxy. Removing this information from the Results would, in our view, disrupt the narrative and hinder the reader's ability to follow the interpretation of the findings. Therefore, we have retained this concise contextual explanation in the Results while ensuring that all detailed procedures are fully described in the Methods section. We hope you understand this decision, given the journal's formatting style and our emphasis on maintaining clarity and readability for the reader.

46. L357: Please revise the alteration-related comments.

Response: We thank the reviewer for this comment regarding the previous concerns, which we understand to reflect a broader request for improved clarity

and consistency in the description, classification, and spatial representation of hydrothermal alteration throughout the manuscript and figures.

Therefore, to summarize the revisions, please see below:

- The four alteration subclasses (Low, Medium, High, Very High) are now clearly defined in the revised Results section, with an explanation that they represent a gradient of visible surface discoloration consistent with advanced argillic to acid-sulfate alteration at La Fossa, as documented by Azzarini et al. (2001).

We thus added the following sentence:

“Narrowing the study area to a defined alteration zone enabled a subclass refinement using an unsupervised clustering approach. This highlighted region corresponds to areas of advanced argillic to acid-sulfate alteration, as previously identified at La Fossa⁵³.”

(Tracked Change: p. 19, line 454-457)

- To support the classification's validity, we reference mineralogical trends reported by Müller et al. (2024) that show strong spatial correspondence with our alteration map.

We thus added the following sentence:

“These spatial patterns align with findings from Müller et al.²⁸, who mapped hydrothermal alteration at La Fossa and validated their classifications using geochemical and mineralogical analyses. Their study showed a systematic increase in sulfur content and a decrease in Fe₂O₃ across alteration gradients, further supporting the interpretation that our subclasses reflect meaningful variations in alteration intensity.”

(Tracked Change: p. 19-20, lines 474-478)

- In areas such as the 1988 landslide region, we have clarified how alteration patterns relate to both in situ processes and possible downslope transport.

We thus added the following sentence:

“In the 1988 landslide region, a distinct region of Very High alteration extends in a north-northwest direction, closely following a gully. This may, in part, reflect the downslope transport of altered material from higher elevations. However, similarly altered surfaces also extend beyond the gully, suggesting that this feature likely reflects both in situ and remobilized altered material.”

(Tracked Change p. 20, line 485-489)

- The CFZ is now clearly labeled in the relevant zoomed-in panel (Figures 5A, B), and an additional Alteration Class legend has been added to aid interpretation.

These updates match the visual style already used in Figure 4 for the northern flank and landslide insets.

(Tracked Change: p. 22, line 502)

- We have also clarified the interpretation of sulfur deposits and revised ambiguous phrasing around mapped alteration features.

We thus added the following sentence:

“Interspersed sulfur deposits concentrated within the CFZ were identified based on yellow coloring in the true-color orthomosaic, confirmed through field observations and consistent with previous studies²⁶ (Fig. 1D, 2A, B).”

(Tracked Change: p. 13, lines 346-349)

- Finally, we added explicit figure references in the text to strengthen the link between visual data and the discussion of alteration zones.

(Tracked Change: p. 19 lines 457, 476)

These revisions, taken together, aim to improve the clarity, consistency, and geological interpretation of alteration-related content across the manuscript and figures. We appreciate the reviewer’s feedback, which helped guide these improvements.

47. L371: Please indicate the location of the different zones in the figures.

Response: Thank you for the helpful comment. To improve clarity regarding the distribution of alteration zones—particularly at the Central Fumarole Zone (CFZ), we have revised the figures as follows:

- Figures 5A and 5B now explicitly label the CFZ to clarify its location relative to areas of the alteration grade.
- An additional Alteration Class legend has been included to facilitate the interpretation of alteration intensity.
- These updates follow the same visual style as Figure 4, which presents other zoomed-in regions discussed in the text, ensuring consistency across all panels.

(Tracked Changes: p. 25, line 578)

We have also updated the main text to reference the revised figures, enhancing the connection between the visual data and the discussion of alteration patterns.

(Tracked Changes: p. 24, 25, line 573 and line 610)

48. L377: Beyond the details shown in the two topographic profiles (corresponding to the collapsed areas), it would be worthwhile to include additional profiles from non-collapsed zones. For example, in the W–E profiles, there appears to be a variation from very strong to very weak, but this cannot be clearly observed due to the current scale.

Response: We thank the reviewer for this helpful observation and suggestion. We have added a panel to the figure (Fig. 5B), which contains an additional A-A' transect from a non-collapsed zone, which clearly shows a drop in Schmidt rebound values across the CFZ. While this sharp transition is specific to the A-A', other W-E transects display more heterogeneous strength distributions. In addition, the A-A' profile is included in the supplementary material to illustrate the sharp transition in relative rock strength from unaltered to altered surfaces. (Tracked Change: Figure 5B: p. 26, line 585 and Supplementary Figure 1)

49. L379: Please adjust the coordinates to ensure a consistent format across all figures. In this case, the indicators "E" and "N" are missing.

Response: Accepted and changes made, the revised Figure 5 contains the corrected formatting, with E and N added to the coordinates.

(Tracked Change: p. 26, line 585)

50. L379: Strikethrough text

Response: Accepted and changes made, Figure 5 caption has been revised.

(Tracked Change: p. 26, line 585)

51. L395: Where?

Response: Accepted and changes made. We added "Table 2, Fig. 6" to clarify the location the text is referring to.

We thus revised the following sentence:

"Unaltered material, marked in grey (Table 2, Fig. 6), accounts for 461 locations (n = 461), or 44% of our dataset, and consistently exhibits the highest rebound values, averaging 60.1 with a median of 63 [95% CI: 62.00-65.00], representing the strongest rocks in our field area."

(Tracked Change: p. 27, line 603-606)

52. L444: Considering the existence of previous studies on hydrothermal alteration, would it be possible to present one or more validation sites? Alternatively, this could be proposed as a potential direction for future research.

Response: We thank the reviewer for this suggestion. While direct sampling was beyond the scope of this study, we address validation through comparison with published mineralogical and geochemical data at La Fossa. Throughout the revised text, we have included references such as Müller et al. (2024), who identified compositional gradients that closely align with our mapped alteration classes. We have also justified the CFZ as a training site for our classification approach, which is supported again by Müller et al. (2024) and additional prior studies that used hyperspectral and thermal methods (Azzarini et al., 2001; Harris et al., 2009). In addition, we now note in the Discussion that future field-based validation, including targeted sampling and mineralogical analysis, would be a valuable next step to refine the classification scheme further.

We thus added the following sentence:

“Future work could build on this foundation by incorporating targeted sampling at mapped alteration sites to validate the classification scheme and further constrain the links between mineralogy, porosity, and mechanical weakening.”
(Tracked Change: p. 34, line 757-760)

53. L549-555: I believe it would be much better—and in line with common practice—for the Methods section to be placed before the Results section in the article's structure. Prior to line 549, there are numerous references to the Methods, which disrupt the flow and make it difficult for the reader to follow the logic of the work. For example, see lines 217, 247.

Response: We thank the reviewer for this helpful comment. As *Communications Earth & Environment* follows *Nature*-style formatting, the Methods section is placed after the Results and Discussion. In response to earlier reviewer feedback requesting clearer links to methodological details, we included a small number of targeted cross-references to the Methods section, where clarification may be helpful to the reader.

We recognize the concern that such references may disrupt the narrative flow. To address this, we have carefully reviewed and reduced cross-referencing to only essential cases and, where possible, integrated brief methodological summaries directly within the Results. We hope this strikes an effective balance between readability and transparency, within the constraints of the required article structure.

54. L606: Which is the original RGB data? The true color orthomosaic maps?

Response: Accepted and changes made, we added “from our true-color orthomosaic”.

We thus revised the following sentence:

“PCA transforms the original RGB data from our true-color orthomosaic into eigenvectors, known as principal components (PCs) (Fig. 8B), which are linear combinations of the original RGB data weighted by transformation coefficients⁸¹.”
(Tracked Change: p. 41, line 972-973)

55. L617: Please move to line 611.

Response: Accepted and changes made, the entire text from lines 617-625 has been moved up in the text to improve flow.

(Changes pg 42, line 978-985)

56. L628 & L630: Please revise the comments in the result section.

Response: Accepted and changes made. The comment addresses the previous issue by clarifying the supervised classification algorithm employed and explaining the rationale behind these decisions. We updated the text to address this issue in the Results, Discussion, and Methods sections. In the revised manuscript, we clarify that the supervised classification distinguishing altered from unaltered surfaces was carried out using the Maximum Likelihood Classification approach. This technique was chosen because it accounts for both the average and variability of spectral signatures, allowing more reliable separation of classes with similar spectral responses, an important consideration in mapping hydrothermal alteration.

We have also provided further detail on how training areas were selected, drawing on prior work that identifies the Central Fumarole Zone (CFZ) as a well-characterized and spectrally distinctive hydrothermal alteration site. These revisions have been incorporated into the Methods and Results to strengthen methodological transparency and reproducibility. Specifically, at this highlighted location within the Methods section, we revised the text to specifically address the previous concerns for clarity and justification behind our decisions.

We thus added the following sentence:

“Following our PCA, we applied a supervised classification using the Maximum Likelihood algorithm in ArcGIS Pro to differentiate between altered and unaltered surfaces. Approximately 100 training areas were selected, with 50 placed within PCA-highlighted red-to-pink regions representing altered surfaces and 50 outside of the region representing unaltered surfaces. These training areas were manually drawn as polygons of varying size and shape, placed both inside and outside of the CFZ. Polygons were selected to capture the range of surface variability visible in the PCA composite, particularly the differences in color. The number of pixels per training site was not standardized, as the focus was on representing spectral diversity rather than a uniform size.

These training areas served as spectral reference inputs for the Maximum Likelihood classifier, allowing each pixel to be assigned to the most spectrally similar category^{80,81}. Our justification for training area selection is based on the CFZ, a well-documented region of persistent fumarolic activity and hydrothermal alteration. Previous studies have mapped this region as an area of silicic and advanced argillic alteration using hyperspectral data⁵³, identified thermal anomalies associated with persistent fumarolic activity⁵², and similar mapping methods as ours using drone-PCA methods, combined with gas flux measurements²⁸. This cross-validation from independent datasets supports the CFZ as a reliable and validated spectral endmember for a supervised

classification. Therefore, this region is well-suited for our training area site selection for our supervised classification of hydrothermal alteration.”
(Tracked Change: p.43, lines 1003-1025)

57. L644: Clarify that the area not classified as a hydrothermal alteration zone is masked in the supervised classification.

Response: We thank the reviewer for this helpful observation. In response, we have revised the text to clarify that the unaltered material was masked out using the results of our supervised classification before applying the unsupervised classification. This allowed us to focus only on altered material during the unsupervised classification.

We thus added the following sentence:

“To better capture these variations, we applied an unsupervised classification to the true-color (RGB) orthomosaic, confining the analysis to the previously classified altered region by masking out the unaltered regions and focusing only on altered material.”

(Tracked Change: p. 44, line 1034-1037)

58. L645: The four predefined classes are interpreted as ranging from low to high, correct? Is there any field verification of the samples? Does greater bleaching always indicate a higher degree of alteration, meaning weaker rocks? Is the lithology being altered always the same throughout the study area? Is there any lithological map that confirms or refutes whether the same rock type is consistently being altered?

Response:

We thank the reviewer for their thoughtful and constructive questions. To address these concerns, we have incorporated the revised texts in locations where they best fit for logical flow.

Here, we have clarified in the revised Methods section that the four predefined classes are interpreted as increasing degrees of hydrothermal alteration, consistent with prior studies that link surface bleaching to mineralogical changes in volcanic settings.

We thus added the following sentence:

“To maintain interpretability, we used a four-class system, consistent with common practice in unsupervised image classification, where large numbers of spectral clusters are often grouped into a smaller number of meaningful classes to reduce complexity⁸⁴. As a result, the four classes reflect inherent spectral

similarities associated with varying degrees of hydrothermal alteration, as expressed through surface color variations such as bleaching intensity.”
(Tracked Change: p. 44, lines 1048-1053)

“We interpret these classes as representing increasing degrees of hydrothermal alteration, consistent with previous studies linking bleaching to mineralogical alteration developed in volcanic settings²¹.”
(Tracked Change: p. 44, lines 1058-1060)

Response: In the same section, we now note that this classification approach allowed us to test and verify rock weakening with increasing alteration using the Schmidt hammer measurements, with results showing a consistent trend of decreasing rebound values with increasing intensity.

We thus added the following sentence:

“To test whether surface bleaching intensity corresponds to rock strength, Schmidt hammer measurements were compared across the four alteration classes defined by our unsupervised classification. This approach allowed us to explore the spatial relationship between surface alteration intensity and in situ rock strength.”

(Tracked Change: p. 47, lines 1136-1139)

Response: Regarding lithology, while a comprehensive lithological analysis of each measurement site is beyond the scope of the study, we address this point in the Limitations section. We cite existing geological maps (Piochi et al. 2009; Di Traglia et al., 2013; De Astis et al., 2013, Costa et al., 2020) that identify the surface of La Fossa cone as predominantly composed of pyroclastic deposits, and we explicitly limited Schmidt hammer sampling to exposed pyroclastic surfaces to reduce lithological variability.

We thus revised the following sentence:

“Although we did not perform a comprehensive lithological analysis at each measurement site, an effort that would significantly reduce, rather than maximize, the number of measurements collected, we mitigated this limitation by relying on established geological maps, which indicate that the La Fossa cone edifice is predominantly composed of pyroclastic deposits^{38,92-95}. In an attempt to limit lithological variability, Schmidt hammer measurements were conducted, where possible, on exposed pyroclastic rock surfaces. This approach helped to ensure that the observed strength variations primarily reflect differences in alteration

intensity, while acknowledging that hydrothermal overprinting can often obscure the original rock type.”

(Tracked Change: p.49, lines 1196-1209)

These revisions aim to clarify our classification assumptions and reinforce the robustness of the field validation strategy.

59. L654: So the digital numbers (0 to 255) were divided into four to define the four classes? 64 for each one?

Response: These classes were not divided equally, such as equal-interval binning. The algorithm used separates naturally occurring clusters within the RGB color space; therefore, the classes reflect inherent spectral similarities within the varying degrees of hydrothermal alteration expressed through surface color variations. The clarification of this has been provided in the text.

We thus added the following sentence:

“Rather than applying an evenly spaced division of digital values (e.g., equal-interval binning), the classification process detects and separates naturally occurring clusters within the RGB color space. To maintain interpretability, we used a four-class system, consistent with common practice in unsupervised image classification, where large numbers of spectral clusters are often grouped into a smaller number of meaningful classes to reduce complexity⁸⁴. As a result, the four classes reflect inherent spectral similarities associated with varying degrees of hydrothermal alteration, as expressed through surface color variations such as bleaching intensity.”

(Tracked Change: p. 44, lines 1046-1053)

60. L732: This constitutes a significant limitation in the study, so I believe it is very important to clearly establish the depth limit of the hydrothermal alteration zones.

Response: We appreciate the reviewer’s comment and agree that the depth extent of hydrothermal alteration is an important consideration, and one that should be further explored, if possible. As noted, our analyses, based on Schmidt hammer rebound values and drone-based mapping, are inherently limited to surface exposures and do not constrain the depth of hydrothermal alteration. We have clarified that the extent of alteration at depth remains an inference in the absence of borehole or geophysical data, which are not available at this site.

However, we stress that this is a strength of our approach: it is intentionally designed as a rapid, surface-based tool for assessing hydrothermal weakening and slope instability in hazardous volcanic settings where subsurface

investigations are often unfeasible. By focusing on accessible outcrops, our method contributes to practical hazard mapping and can support monitoring and mitigation efforts in similar environments.

We thus added the following sentence:

“While this limitation is acknowledged, it also reflects a practical reality at many active volcanic systems where borehole sampling is not feasible. Our surface-based approach is therefore intentionally designed to utilize accessible outcrops to develop a rapid, non-invasive approach for assessing volcanic instability. Future studies that integrate remote sensing, rock property measurements, and subsurface geophysical data, such as electrical resistivity tomography⁸⁹, would provide a more holistic assessment approach, but we stress that geophysical measurements are not possible at particularly hazardous volcanoes. This limitation reinforces the utility of our surface-based method, which is designed as a rapid, field-applicable tool for assessing hydrothermal weakening and slope instability where subsurface data cannot be readily acquired.”

(Tracked Change: p.48, lines 1160-1172)

61. L739: add “are”.

Response: Accepted and changes made, in the revised sentence, the word “are” is added for better sentence flow and grammar.

We thus revised the following sentence:

“Additional limitations are related to the spatial accuracy and environmental constraints of the data.”

(Track Change: p. 48, line 1173)

62. L766: add “ with past work results”

Response: Accepted and changes made, in the revised sentence, we added “past work” to the text.

We thus revised the following sentence:

“Consistent with past works^{26,28,48}, PC1 correlated with surface brightness, PC2 highlighted the CFZ, and PC3 emphasized sulfur deposits, suggesting that the classifications are not artifacts and can be replicated.”

(Track Change: p. 50, line: 1214)

Benjamin F. De Jarnatt
Section 2.1, Physics of Earthquakes and Volcanoes
Helmholtzstr. 7, 14467 Potsdam, Germany
www.gfz.de

Potsdam, 17 Oct. 2025

Dear Reviewers,

We would like to thank you for your thoughtful and constructive feedback on our manuscript titled: "**Hydrothermal weakening and slope instability at Vulcano (Italy) analyzed using drones and in-situ strength measurements**" (Paper # COMMSENV-25-1704B), which has been deemed acceptable "in principle" for publication in *Nature Communications Earth & Environment*. Your recommendations and feedback have been invaluable throughout this process, and your comments have significantly improved the clarity and presentation of our work. In this final revision, we have carefully addressed the minor comments from Reviewer 1 and incorporated the requested edits accordingly. These refinements have further strengthened the work, clarified any open issues within the text, and ensured the manuscript is in full alignment with the journal's standards.

We also greatly appreciate that Reviewers 2 and 3 expressed satisfaction with our previous revisions and have endorsed the manuscript for publication. Their assessment, together with their earlier constructive feedback, has been vital in improving the clarity, methodological justification, and analytical depth of the study.

Below, we provide a final point-by-point response to the remaining comments. For transparency, the Reviewers' comments are shown in **black**, and our responses are in **red** with corresponding page and line number for convenience.

Sincerely,

Benjamin F. De Jarnatt and co-authors

REVIEWER COMMENTS:

Reviewer #1 (Remarks to the Author):

Manuscript number: COMMSENV-25-1704

Hydrothermal weakening and slope instability at Vulcano (Italy) analyzed using drones and in-situ strength measurements.

In order to determine the correlations with alteration types in the Vulcano Islands and assess the degree of stability, this research uses an integrated strategy using in situ Schmidt hammer and UAV-based photogrammetry. These findings support earlier research on possible failure. For hydrothermalized massifs, a new classification is suggested. The subsurface conditions and limitations have been considered, implying future studies. From a methodological perspective, this is an impressive correction that involves several clarifications.

Response:

Thank you for your thoughtful and positive feedback. We are glad that the methodological integration and clarification were well received. We have revised the manuscript based on your suggestions below regarding the subsurface constraints to further enhance the study's contribution to volcanic stability assessments.

From a methodological perspective, this is an impressive correction that involves several clarifications. The introduction (lines 100–107) and the discussion with the help of the bibliography should, however, make clear the features of the destabilization of volcanic islands in subduction zones and associated trenches, including hydrothermal processes, particularly Aeolian islands.

Response:

We appreciate the reviewer for this insight. In response, we have clarified the tectonic and geodynamic framework relevant to the instability mechanisms, specifically in the Aeolian Archipelago. Specifically, we now emphasize that the subduction-related stresses and hydrothermal activity have contributed to past mass-wasting events in Aeolian Arc volcanoes (e.g., Stromboli, Vulcano), referencing Tibaldi et al, 2001 and Romagnoli et al., 2009. This addition was best placed in the *Study Area* for textual flow.

The text now reads:

"The La Fossa volcano, located on Vulcano Island, within the Aeolian Archipelago, represents the southernmost volcanic structure of this volcanic arc situated in the Tyrrhenian Sea, off the northern coast of Sicily, Italy (Fig. 1B). The archipelago lies above a subduction zone related to the convergence of the African and European plates, characterized by the north-westward subduction of the Ionian lithospheric slab beneath the Tyrrhenian lithosphere, where subduction-related stresses and

hydrothermal activity have contributed to flank instability across Aeolian Arc volcanoes^{38,39}. A complex NNW–SSE fault system further influences the structural and volcanic development of the region and plays a significant role in the recent tectonic activity and deformation processes observed specifically at Vulcano^{40,41}." (Tracked Change: p. 5-6; lines 159-168)

Despite being mentioned in the introduction (lines 89–95), the abstract doesn't seem to address the connections between hydrothermal alteration and the destabilizing of a volcanic island.

Response:

We thank the reviewer for this helpful observation. In response, we have slightly revised the text to clarify the effects of hydrothermal alteration in a broader context, while remaining within the word limit.

The revised text now reads:

"Instability at volcanic edifices poses significant hazards, yet the processes driving rock weakening, particularly on steep, eroding flanks, remain poorly understood due to limited accessibility. Hydrothermal alteration is a key factor in weakening volcanic rocks and contributing to edifice destabilization and flank instability." (Tracked Change: p. 1, lines 20-23)

Moreover, sulfur and vegetations are clearly detected in relation to hydrothermal changes (Fig. 2).

Response:

We thank the reviewer for this observation. We agree that both vegetation and sulfur are visible in the principal component analysis; however, these features are not directly indicative of hydrothermal alteration. Their appearance reflects the spectral variance captured by the PCA, which emphasizes surfaces with distinct reflectance characteristics. Specifically, PC3 accounts for minimal variance within the data and highlights localized sulfur and vegetation. While the PCA composite highlights hydrothermal alteration in red-to-pink hues, sulfur deposition in yellow hues within the CFZ, and vegetation in green hues. To improve clarity, we have revised the text in the Results section (pp. 10-11, lines 286, 287, 290, and 308) to note that PCA measures variance within the data, and that the highlighted sulfur and vegetation signatures are distinct from the alteration signal.

But don't be afraid to use the many bibliographic references that discuss the relationships between spectral classification zonation and hydrothermal alteration halos (Fig. 2, line 428: "other altered region in red-to-pink colors"). Indeed, the conduit exhibits an inner leaching and amorphous silica zone, and is surrounded by less acidic alteration zones related to the formation of alunite and kaolinite (see also Müller et al., 2021; Robb, 2005).

Response:

We thank the reviewer for this important suggestion. In response, we have added references Fulignati et al. (1999) and Müller et al. (2021) in the Results section to acknowledge the observed mineralogical relationships between silicic and advanced argillic alteration at Vulcano. In addition, we incorporated Robb (2005) in the Discussion section to provide a broader theoretical context for hydrothermal alteration zonation. The revised text now reads:

Results:

These regions reflect silicic to advanced argillic alteration, consistent with previous observed patterns of hydrothermal alteration zonation at Vulcano^{46,58}. (Tracked Change: p. 11, lines: 306-308)

Discussion:

This distribution reflects a well-established hydrothermal zonation pattern, with the CFZ representing the acidic, silica-rich region surrounded by an advanced argillic halo (OFZ)⁸⁴. (Tracked Change: p.33 lines: 830-832)

Another kind of destabilization interactions (Fig. 7) is implied by this kind of structural zoning, such as a collapse depletion by conduit collapse. The theoretical structural relationships based on hydrothermal alteration gradients (lines 813-814: faulting not shown) would be preferable. In actuality, the circular fault hypothesis must be taken into consideration when analyzing the features of volcanic collapses.

Response:

We appreciate the reviewer's concern. We agree that the previous version of Figure 7 could be interpreted as implying deeper structural controls on hydrothermal alteration. In response, we have reworded some text in the Introduction to ensure that the roles of structurally controlled fluid migration and hydrothermal alteration is clearly acknowledged. We have revised Figure 7 to emphasize the shallower structural features and hydrothermal zonation. The revised figure now highlights that shallow processes (e.g., landslides or collapses) can excavate and expose previously hidden

hydrothermally altered zones. Finally, we have a clarifying sentence in the *Implications* subsection to better explain these relationships.

The text now reads:

Introduction

"Typically, regions undergoing hydrothermal alteration are found near craters and along the flanks of volcanic edifices, where fluid ascent is controlled by fracture networks³⁶."

Implications

"In some cases, a landslide or collapse event can excavate and expose deeper, previously buried zones of hydrothermal alteration (Fig.7)."

(Tracked Change: p. 25, lines: 629-631)

Finally, the rewriting contains grammatical errors, particularly in lines 378, 380, and 385. "Gray" is an American English word, but "grey" is used in British English.

Response:

We thank the reviewer for the comment. We had previously used the British English spelling "grey" by oversight. This has now been corrected to the American form of "gray" throughout the manuscript (p. 10, line 288 and p. 12, line 321), maintaining consistency in language style.

In both upper and lower case letters, the word "Unaltered" appears frequently (lines 518, 699, 700, 708, etc.). Mathematical notation typically does not observe spaces (lines 524, 581, 701, 702, etc.).

Response:

We thank the reviewer for pointing out the inconsistent capitalization of the word "unaltered." We have carefully reviewed the manuscript and standardized its usage throughout. The term "Unaltered" is now capitalized when referring to the alteration class defined in our classification scheme (e.g., Table 2 and Figure 6), and written in lowercase "unaltered" when used descriptively to refer to general rock or surface conditions.

Additionally, in response to the comment about grammatical errors, we have reviewed all mathematical notation and made the following corrections for consistency and clarity: removed unnecessary spaces before and after superscripts, standardized the notation "p" from our Kruskal-Wallis test to " p " replaced "x" with " \times " where dimensional expressions are present, and ensured proper spacing around mathematical operators (e.g., $<$, $>$). These revisions align the manuscript with standard mathematical and typographical conventions.

Reviewer #2 (Remarks to the Author):

I am satisfied with the authors revisions and responses to my comments. The revised manuscript has substantially improved in clarity, methodological justification, and analytical depth. The authors have explicitly addressed the concerns I raised, including clarification of the research gap, better integration of comparative literature, quantitative strengthening of the results with appropriate statistical analyses, and a more critical and balanced discussion that acknowledges limitations and operational boundaries. The conclusions are now more precise and well-grounded in the data, and the methodological choices are clearly justified.

Response:

We sincerely appreciate the reviewer's thoughtful and detailed feedback during the initial review round, as well as the positive assessment of our revised manuscript. Your comments and suggestions have significantly enhanced the clarity, focus, and scientific rigor of our work. In particular, your recommendations helped us clarify the research gap and novelty in the abstract and introduction, refine the rationale for the case study, expand our quantitative analyses, improve the **Discussion** and **Conclusion** sections, and ensure figure consistency. We are pleased to hear that the revisions meet your expectations.

Reviewer #3 (Remarks to the Author):

Review- Report to the authors

Title: Hydrothermal weakening and slope instability at Vulcano (Italy) analyzed using drones and in-situ strength measurements

Authors: Benjamin F. De Jarnatt; Thomas R. Walter; Michael J. Heap; Daniel Mueller; Antonino Fabio Pesciotta

After carefully analyzing the rebuttal letter, it is clear that the authors provided a point-by-point response to all the reviewers' comments. I find their replies to be highly satisfactory, as they significantly contribute to making the manuscript clearer and more concise. I sincerely appreciate the effort devoted to addressing each request in detail. This revised version fully incorporates all my previous comments. In my opinion, it represents an excellent piece of research, proposing a sound methodology to study volcanic slope instability with direct application to hazard monitoring and early warning systems.

The selected case study, La Fossa cone (Vulcano, Italy), is particularly well suited for this type of analysis, combining drone photogrammetry with in situ Schmidt hammer measurements and generating an alteration map through Principal Component Analysis. The results presented are highly relevant, showing an approximate 50% reduction in relative rock strength in areas affected by degassing and hydrothermal activity. These findings are consistent with evidence from past mass-wasting events.

Response:

We sincerely thank the reviewer for their positive assessment of our revised manuscript. We appreciate your recognition that the revisions addressed all previous comments and enhanced the clarity, conciseness, and overall quality of the work. Your detailed feedback, especially the annotated PDF highlighting specific issues, was incredibly helpful in refining the manuscript's structure, language, and presentation.

We are grateful for your thoughtful evaluation that the study provides a solid and relevant contribution to understanding volcanic slope instability and its implications for hazard monitoring and early-warning systems. Your constructive comments and supportive feedback were highly motivating and helped us develop a clearer, more impactful paper.